# DPLM-2: A Multimodal Diffusion Protein Language Model

**Xinyou Wang**[*◇♡] **Zaixiang Zheng**[†♡] **Fei Ye**[♡] **Dongyu Xue**[♡] **Shujian Huang**[◇] **Quanquan Gu**[‡♡]
[◇]School of Computer Science, Nanjing University [♡]ByteDance Research
wangxinyou@smail.nju.edu.cn, {zhengzaixiang, quanquan.gu}@bytedance.com
[†]Project Lead [‡]Corresponding Author
Project Page: https://bytedance.github.io/dplm/dplm-2

## ABSTRACT

Proteins are essential macromolecules defined by their amino acid sequences, which determine their three-dimensional structures and, consequently, their functions in all living organisms. Therefore, generative protein modeling necessitates a multimodal approach to simultaneously model, understand, and generate both sequences and structures. However, existing methods typically use separate models for each modality, limiting their ability to capture the intricate relationships between sequence and structure. This results in suboptimal performance in tasks that requires joint understanding and generation of both modalities. In this paper, we introduce DPLM-2, a multimodal protein foundation model that extends discrete diffusion protein language model (DPLM) to accommodate both sequences and structures. To enable structural learning with the language model, 3D coordinates are converted to discrete tokens using a lookup-free quantization-based tokenizer. By training on both experimental and high-quality synthetic structures, DPLM-2 learns the joint distribution of sequence and structure, as well as their marginals and conditionals. We also implement an efficient warm-up strategy to exploit the connection between large-scale evolutionary data and structural inductive biases from pre-trained sequence-based protein language models. Empirical evaluation shows that DPLM-2 can simultaneously generate highly compatible amino acid sequences and their corresponding 3D structures eliminating the need for a two-stage generation approach. Moreover, DPLM-2 demonstrates competitive performance in various conditional generation tasks, including folding, inverse folding, and scaffolding with multimodal motif inputs, as well as providing structure-aware representations for predictive tasks.

## 1 INTRODUCTION

Proteins are macromolecules that execute crucial roles in every living organism. They are characterized by their amino acid sequences and three-dimensional structure, where the sequence determines the structure, which in turn governs the protein's function. Generative modeling for proteins has made significant strides in recent years. Among them, diffusion models (Ho et al., 2020; Song et al., 2020) exhibit great success in protein structure-based generative modeling (Watson et al., 2023; Yim et al., 2023). Meanwhile, large-scale protein language models (Rives et al., 2019; Lin et al., 2022), trained on evolutionary-scale sequence database, have become one of the most important cornerstones in sequence-based foundation models for protein sequence representation learning and generation. Remarkably, DPLM (Wang et al., 2024), a discrete diffusion (Austin et al., 2021) based protein language models, has exhibited the state-of-the-art performance in both sequence generation and understanding, addressing a wide range of sequence-oriented applications.

Many protein engineering applications, *e.g.*, motif-scaffolding (Watson et al., 2023; Yim et al., 2024) and antibody design (Jin et al., 2021; Kong et al., 2022; Zhou et al., 2024), require jointly determine both structure and sequence. However, the aforementioned approaches mostly employ generative models for one modality (either sequence or structure) and resort to separate models (Jumper et al., 2021; Dauparas et al., 2022) for the other. This highlights the pressing need for multimodal protein generative models that can integrate both sequence and structure, enabling a more comprehensive understanding of protein behaviors and functions. This, therefore, raises the following question:

---

[*]This work was done during Xinyou's internship at ByteDance Research.

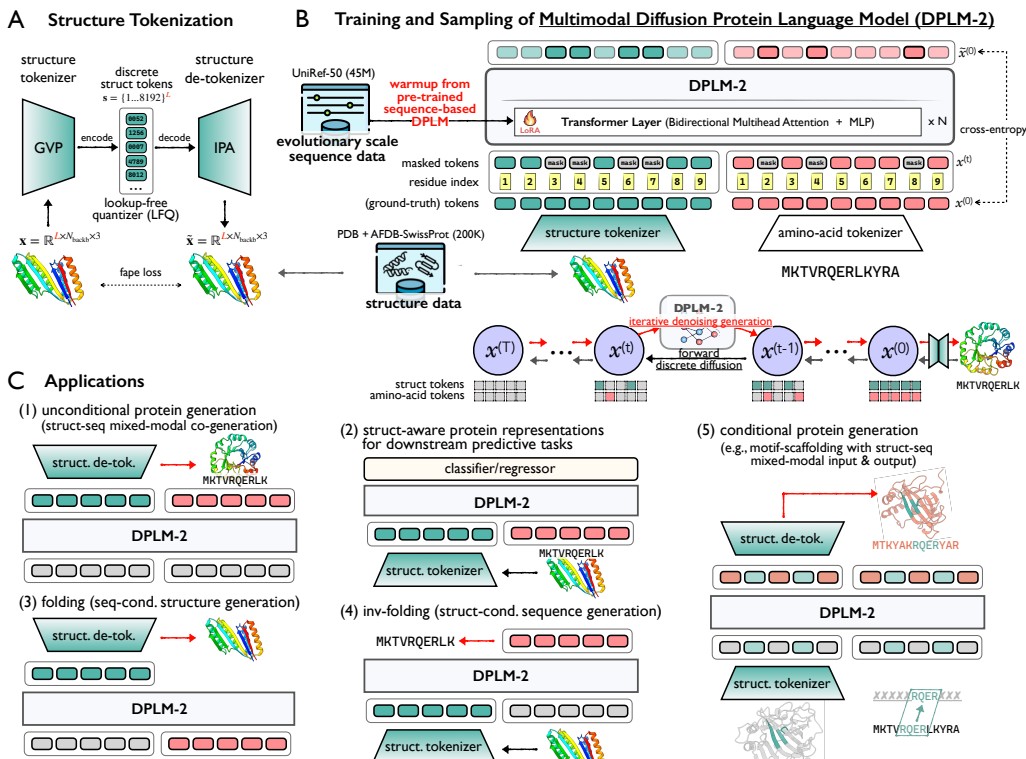

Figure 1: *Overall illustration of* DPLM-2. **(A)** Structure tokenization consists of a GVP-based encoder to yield invariant backbone geometric features, a lookup-free quantizer (LFQ) to discretize encoded structural features into structure tokens within a codebook, and an IPA-based decoder as de-tokenizer to convert structure tokens back to backbone atomic coordinates. **(B)** Multimodal learning and generation of protein structure and sequence with DPLM-2. **(C)** Various applications of DPLM-2 as a protein foundation model: (1) unconditional protein sequence-structure mixed-modal co-generation; (2) protein sequence-structure joint representation for predictive tasks; (3) structure prediction; (4) fixed-backbone sequence generation; (5) conditional protein generation with structure-sequence mixed-modal input and output.

> *Can we build a multimodal protein foundation model to simultaneously model, understand, and generate both sequences and structures?*

To pursue this goal, Multiflow (Campbell et al., 2024) is a recent effort for structure-sequence co-generation that incorporates sequences into structure-based generative models using multimodal flow matching. Despite its impressive structure generation capability, Multiflow exhibits suboptimal performance in co-generating structurally-compatible sequences and consequently resorts to instance-level knowledge distillation from ProteinMPNN (Dauparas et al., 2022). Furthermore, it completely falls short in protein folding for given sequences, showing Mulitflow's inadequacy in sequence understanding. We argue that this bottleneck arises from the absence (co-)evolutionary inductive bias derived from massive pre-training from sequence database, as prior studies have demonstrated that the evolutionarily-informed representations learned by pre-trained protein language models implicitly capture structural information enables direct structure prediction (Lin et al., 2022). As a consequence, the limitation in sequence understanding and generation renders Multiflow inadequate as a multimodal protein generative foundation.

Inspired by the connection between evolutionary knowledge and spatial interactions, we deem that sequence-based generative language models like DPLM, with their strong sequence generation and predictive abilities, hold great promise as a foundation for multimodal learning for proteins. Despite its exciting potential, this approach presents two key challenges: (1) language models cannot directly handle continuous data like structure; and (2) language models heavily necessitate sufficient scale of data and compute resources while structure data is much smaller compared to sequence databases.

In this paper, we address the aforementioned questions by introducing DPLM-2, a multimodal protein foundation model that advances the state-of-the-art discrete diffusion-based protein language model (*i.e.*, DPLM) to accommodate both sequences and structures. By training on both experimental and high-quality synthetic structures, DPLM-2 learns the joint distribution of sequence and structure,

as well as their marginals and conditionals. We present several key recipes to facilitate multimodal learning in DPLM-2: (1) the core difficulty lies in enabling the language model to learn structural information, which is challenging and remains elusive, for which we develop a lookup-free quantization (LFQ, Yu et al., 2023) structure tokenizer to convert 3D coordinates to discrete tokens and vice versa (Fig. 1A, §3.3); (2) we implement an efficient warm-up strategy to exploit the connection between large-scale evolutionary data and structural inductive biases from pre-trained sequence-based DPLM (Fig. 1B, §3.2); and (3) we also address the exposure bias problem in discrete diffusion for sequence learning (Ranzato et al., 2016; Bengio et al., 2015) by a self-mixup training strategy that leads to enhanced generation quality and diversity.

We highlight our main contributions and findings as follows:

(i) We present DPLM-2, a multimodal protein generative language model that aims to simultaneously model, understand and generate protein structure and sequence. We show that it can be fairly efficient and effective to obtain a mulitmodal protein model with moderate amount of high-quality data, a decent structure tokenizer and publicly-accessible sequence-only pre-trained language models.

(ii) As a mulitmodal generative model, DPLM-2 enables unconditional protein co-generation of both structure and sequence, which demonstrates good structure-sequence consistency (Fig. 1C(1)). Our empirical evaluation shows that DPLM-2 attains competitive co-generation performance compared to structure-based generative approaches, while the proteins generated by DPLM-2 have a better alignment with the characteristics of natural proteins in secondary structure statistics (§4.1).

(iii) In addition, DPLM-2 supports various conditional generation tasks by its multimodal nature, ranging from (sequence-conditioned) folding (Fig. 1C(3), §4.2), (structure-conditioned) inverse-folding (Fig. 1C(4), §4.3), to more successful motif-scaffolding given multimodal motif conditioning (Fig. 1C(5), §4.4).

(iv) Last but not least, we demonstrate that the structure-aware protein representation learned by DPLM-2 brings additional benefit for a range of protein predictive tasks (Fig. 1C(2), §4.5).

**Concurrent work.** During the development of DPLM-2, we became aware of the recently proposed multimodal generative protein language model, ESM3 (Hayes et al., 2024), which also jointly models tokenized structure and sequence using a generative masked language model. While both models aim for similar goals, DPLM-2 differs from ESM3 in several key aspects: **(1)** *Multimodal protein generation:* DPLM-2 treats structure and sequence modalities equally by design and emphasizes the simultaneous co-generation of compatible protein sequence and structure, whereas ESM3 is a sequence-first model (other modalities are subject to dropout during training) and generates in cascaded modality-by-modality manner. **(2)** *Data and compute efficiency:* ESM3 seeks to perform mulimodal pre-training from scratch using a huge amount of synthetic data, with modal size ranging from 1.4B to 98B. With strict license and absence of training infrastructure, this prohibits community from replicating for customized purposes. In contrast, DPLM-2 leverages much smaller datasets (PDB + SwissProt) and builds on open-source, pre-trained sequence-based DPLM (150M/650M/3B), which leverages DPLM's learned evolutionary knowledge and inherits strong sequence understanding and generation capabilities. We are also committed to open-source our models, training and inference code to democratize multimodal generative protein LMs to benefit the community. Overall, we believe DPLM-2 provides unique contributions to the community.

## 2 PRELIMINARIES

### 2.1 GENERATIVE MODELING FOR PROTEIN

Table 1: *Generative tasks w.r.t. structure & sequence.*

| task | objective |
|---|---|
| folding | $p_\theta(\mathbf{x}|\mathbf{s})$ |
| inv-folding | $p_\theta(\mathbf{s}|\mathbf{x})$ |
| seq. gen. | $p_\theta(\mathbf{s})$ |
| struct. gen. | $p_\theta(\mathbf{x})$ |
| seq-struct co-gen. | $p_\theta(\mathbf{s}, \mathbf{x})$ |

The aim of generative protein modeling is to estimate the underlying distribution $\text{prot} \sim q(\text{prot})$ of the protein data of our interest by learning a probabilistic model $p_\theta(\text{prot})$. Here $\text{prot} = (r_1, r_2, \ldots, r_L)$ denotes a protein with $L$ residues, where each residue $r_i = (s_i, x_i)$ is represented by two major modalities, *i.e.*, $s_i \in \{0, 1\}^{|\mathcal{S}|}$ is a categorical variable for its amino acid type in $\mathcal{S} = \{1, ..., 20\}$, and $x_i \in \mathbb{R}^{N_{\text{atoms}} \times 3}$ is the real-value Cartesian coordinates of its residue atoms (we only consider backbone atoms herein, *i.e.*, $[N, C_\alpha, C, O]$ with $N_{\text{atoms}} = 4$). Namely,

$$p_\theta(\text{prot}) = p_\theta(s_1, s_2, \ldots, s_L, \ x_1, x_2, \ldots, x_L) = p_\theta(\mathbf{s}, \mathbf{x})$$

As a result, most of protein tasks can be viewed as specifying their input conditioning and output between these two modalities (Tab. 1), including (1) sequence-conditioned structure prediction (fold-

ing, Jumper et al., 2021; Lin et al., 2022; Huguet et al., 2024), (2) structure-conditioned sequence generation (inverse folding or fixed-backbone design, Dauparas et al., 2022; Hsu et al., 2022; Zheng et al., 2023b), (3) sequence learning or generation (Rives et al., 2019; Nijkamp et al., 2022; Alamdari et al., 2023; Wang et al., 2024), (4) structure generation (Yim et al., 2023; Watson et al., 2023; Ingraham et al., 2023), and (5) sequence-structure co-generation (Jin et al., 2021; Shi et al., 2022; Campbell et al., 2024). These further enable various conditional applications by allowing single or mixed-modal conditioning for partial generation, *e.g.*, motif-scaffolding and antibody design.

## 2.2 DIFFUSION PROTEIN LANGUAGE MODEL (DPLM)

Language models (LMs), typically parameterized by Transformers (Vaswani et al., 2017) have become the *de facto* choice dominating different domains with scalable and performing expressiveness (OpenAI, 2023). Among them, protein LMs have been serving as one of the AI foundation for protein sequence learning (Rives et al., 2019; Lin et al., 2022) and generation (Nijkamp et al., 2022; Alamdari et al., 2023).

Diffusion protein language model (DPLM, Wang et al., 2024), in particular, shows excelling performance in both generation and representation learning of protein sequences. DPLM is grounded in *absorbing* discrete diffusion framework (Austin et al., 2021; Zheng et al., 2023a), which is characterized by a forward and backward Markov process. Let $\mathtt{Cat}(\mathbf{x};\mathbf{p})$ be a categorical distribution on protein sequence $\mathbf{y}$ parameterized by a vector $\mathbf{p}$ on $(|\mathcal{V}|-1)$-dimensional probability simplex. The forward process of discrete diffusion defines a Markov process governed by the transition kernel $q(\mathbf{x}^{(t)}|\mathbf{x}^{(t-1)}) = \mathtt{Cat}\big(\mathbf{x}^{(t)};\beta_t\mathbf{x}^{(t-1)}+(1-\beta_t)\mathbf{q}_{\text{noise}}\big)$ that gradually perturb the data $\mathbf{x}^{(0)} \sim q(\mathbf{x}^{(0)})$ into a stationary distribution $\mathbf{x}^{(T)} \sim \mathbf{q}_{\text{noise}}$. For absorbing diffusion, $\mathbf{q}_{\text{noise}}$ is the point mass with all of the probability on the absorbing (mask) state. The learned *backward* process $p_\theta(\mathbf{x}^{(t-1)}|\mathbf{x}^{(t)})$ reversely denoises the $\mathbf{x}^{(T)}$ towards the data distribution $\mathbf{x}^{(0)}$, which is typically optimized by the variational bound of the log-likelihood (Ho et al., 2020):

$$\mathbb{E}_{q(\mathbf{x}^{(0)})}\big[\log p_\theta(\mathbf{x}^{(0)})\big] \geq \mathbb{E}_{q(\mathbf{x}^{(0:T)})}\left[\log \frac{p_\theta(\mathbf{x}^{(0:T)})}{q(\mathbf{x}^{(1:T)}|\mathbf{x}^{(0)})}\right]$$
$$= \mathbb{E}_{q(\mathbf{x}^{(0)})}\bigg[\log p_\theta(\mathbf{x}^{(0)}|\mathbf{x}^{(1)}) + \sum_{t=2}^{T}\underbrace{-\mathrm{KL}\big[q(\mathbf{x}^{(t-1)}|\mathbf{x}^{(t)},\mathbf{x}^{(0)})\|p_\theta(\mathbf{x}^{(t-1)}|\mathbf{x}^{(t)})\big]}_{\mathcal{J}_t}\bigg] + \text{const.},$$

where $\mathcal{J}_t$ is the learning objective. The learning objective of discrete diffusion can be further simplified into reweighted cross-entropies (Zheng et al., 2023a), resembling masked language modeling at arbitrary noise levels:

$$\mathcal{J}_t = \mathbb{E}_{q(\mathbf{x}^{(0)})} - \mathrm{KL}\big[q(\mathbf{x}^{(t-1)}|\mathbf{x}^{(t)},\mathbf{x}^{(0)})\|p_\theta(\mathbf{x}^{(t-1)}|\mathbf{x}^{(t)})\big]$$
$$= \mathbb{E}_{q(\mathbf{x}^{(0)})}\Big[\lambda^{(t)}\sum_{1\leq i\leq L}b_i(t)\cdot\log p_\theta(x_i^{(0)}|\mathbf{x}^{(t)})\Big], \tag{1}$$

where $\lambda^{(t)}$ is a weighting coefficient induced from the specific noising schedule and $b_i(t) = \mathbf{1}_{x_i^{(t)}\neq x_i^{(0)}}$. For inference, DPLM is able to generate amino acid sequences by the reverse iterative denoising process of discrete diffusion (Hoogeboom et al., 2021; Austin et al., 2021) from the following distribution,

$$p_\theta(\mathbf{x}^{(t-1)}|\mathbf{x}^{(t)}) = \sum_{\tilde{\mathbf{x}}^{(0)}} q(\mathbf{x}^{(t-1)}|\mathbf{x}^{(t)},\tilde{\mathbf{x}}^{(0)})p_\theta(\tilde{\mathbf{x}}^{(0)}|\mathbf{x}^{(t)}).$$

Specifically, at time $t$, it first generates $\tilde{\mathbf{x}}^{(0)}$ from $p_\theta(\cdot|\mathbf{x}^{(t)})$, then a less noisy $\mathbf{x}^{(t-1)}$ is sampled by $q(\cdot|\mathbf{x}^{(t)},\mathbf{x}^{(0)}=\tilde{\mathbf{x}}^{(0)})$. Within absorbing diffusion, the generation process can be viewed as an iterative *mask-predict* approach. For sequence representation for predictive tasks, it can be obtained by simply letting DPLM take the sequence as input.

## 3 DPLM-2: A MULTIMODAL DIFFUSION PROTEIN LANGUAGE MODEL

### 3.1 OVERVIEW

Fig. 1 illustrates DPLM-2's overall architecture. DPLM-2 is built on the state-of-the-art sequence-based generative protein LM, *i.e.*, DPLM (Wang et al., 2024), using a discrete diffusion probabilistic framework to concurrently model both protein sequences and their corresponding structures. To facilitate structure learning in language models, we introduce a token-based representation for protein structure via a tokenizer that converts $\mathbf{x} \in \mathbb{R}^{L\times N_{\text{backb}}\times 3}$, the 3D coordinates of the protein backbone into a discrete structure token sequence, denoted as $\mathbf{z} = (z_1, z_2, \ldots, z_L) \in \{0,1\}^{L\times|\mathcal{Z}|}$, where each token $z_i$ represents a local structural element of the $i$-th residue. Given tokenized structure,

DPLM-2 processes mulitmodal input by concatenating the structure token sequence $\mathbf{z}$ with the corresponding amino acid sequence $\mathbf{s}$ for the same protein. Notably, there exists a position-by-position correspondence between $\mathbf{z}$ and $\mathbf{s}$, where $z_i$ and $s_i$ refer to the two modalities of the $i$-th residue, respectively. To reinforce this correspondence, we assign identical position encodings to both $z_i$ and $s_i$, thereby ensuring that structural and sequence information is aligned at the residue level.

To train DPLM-2, we leverage a high-quality dataset comprising 20K clustered experimental structures from the Protein Data Bank (PDB) (Berman et al., 2000) and 200K predicted structures from the AFDB SwissProt split (Varadi et al., 2022), with length $< 512$. During training, DPLM-2 is tasked with denoising the input sequence across a spectrum of noise levels, ranging from fully noisy to completely clean. The multimodal training objective of DPLM-2 is derived from Eq. (1) as,

$$\mathcal{J}_t = \mathbb{E}_{q(\mathbf{x}^{(0)}, \mathbf{s}^{(0)}), \mathbf{z}^{(0)} \leftarrow tokenize(\mathbf{x}^{(0)})} \left[ \lambda^{(t)} \sum_{1 \leq i \leq L} b_i(t) \cdot \log p_\theta(z_i^{(0)}, s_i^{(0)} | \mathbf{z}^{(t)}, \mathbf{s}^{(t)}) \right], \quad (2)$$

where $\log p_\theta(z_i, s_i | \cdot) = \log p_\theta(z_i | \cdot) + \log p_\theta(s_i | \cdot)$ by assuming conditional independence (see discussion in §G). By learning $p_\theta(\mathbf{z}^{(t-1)}, \mathbf{s}^{(t-1)} | \mathbf{z}^{(t)}, \mathbf{s}^{(t)})$, the model enables the simultaneous generation of highly correlated protein structures and sequences. This eliminates the need for a cascaded generation, allowing us to derive both the protein's structure and sequence in a single step.

To further enhance DPLM-2's ability to differentiate between structure and sequence, noising level for each modality is subjected to distinct scheduler, denoted as $t_\mathbf{z}$ and $t_\mathbf{s}$, respectively. This facilitates a more comprehensive understanding of the relationships between protein sequences and their corresponding structures. This design also allows us to explore arbitrary combinations of $(t_\mathbf{z}, t_\mathbf{s})$, thus providing flexible sampling options, including sampling from the marginals of each modality and conditionals between them for various applications (Fig. 1C). For conditional sampling (e.g., folding and inverse-folding), we set the noise scheduler of the conditioned modality to 0, which means no noise in the conditioned modality. Please refer to §A.2 for more details.

Furthermore, we also identify the exposure bias issue in discrete diffusion for sequence learning (Ranzato et al., 2016; Bengio et al., 2015), and mitigate this by proposing a self-mixup strategy inspired by scheduled sampling, which improves both generation quality and diversity (see §A.5).

### 3.2 Efficient Warm-up from Pre-trained Sequence-based DPLM

Protein sequences encode critical evolutionary information, reflecting co-evolutionary processes where residue pairs mutate together and often interact in 3D space, offering insights for predicting protein folding (Melnyk et al., 2022b). Lin et al. (2022) further showed that protein language models trained on large-scale evolutionary data implicitly capture this information, which can facilitate structure prediction. Motivated by the link between evolutionary knowledge and structural interactions, we propose to built DPLM-2 with an efficient warmup from pre-trained sequence-based DPLM, to make the most of established evolutionary information for protein structure modeling, Since our structure dataset is significantly smaller than UniRef50 sequence database (200K *vs.* 45M), enabling efficient fine-tuning of the pre-trained model. we want to keep the sequence knowledge intact and reduce the risk of catastrophic forgetting, we apply LoRA (Hu et al., 2021) to limit too much deviation to the original parameters. This approach not only lowers training costs compared to starting from scratch but also effectively transfers valuable evolutionary information.

### 3.3 Learning Structure Tokenization

The core difficulty of achieving a mulimodal protein LM lies in enabling the language model to learn structural information, which is challenging and remains elusive, Tokenizing continuous data modalities into discrete representations (Van Den Oord et al., 2017) has gained attraction across domains like image synthesis due to its ability to capture compact, meaningful information, enabling effective compression and efficient generation, especially with sequence-based models like Transformers. Recent efforts have applied this approach to protein structure coordinates (Van Kempen et al., 2024; Liu et al., 2023; Gao et al., 2024; Lu et al., 2024). This allows language models to better learn the composition of local structural elements. However, how to learn an effective structure tokenizer remains an active research question.

Structure tokenization und er a typical VQ-VAE (Van Den Oord et al., 2017) framework can be summarized as follows:

$$\mathbf{x} \xrightarrow{encoder} \mathbf{e} \xrightarrow{quantizer} \mathbf{z} \xrightarrow{decoder} \tilde{\mathbf{x}},$$

where (1) a structure encoder encodes backbone 3D coordinates $\mathbf{x} \in \mathbb{R}^{L \times N_{backb} \times 3}$ into in-

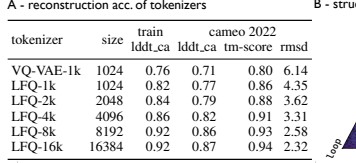
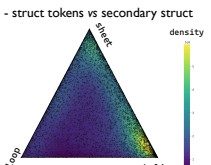

A - reconstruction acc. of tokenizers

| tokenizer | size | train lddt_ca | cameo 2022 lddt_ca | tm-score | rmsd |
|---|---|---|---|---|---|
| VQ-VAE-1k | 1024 | 0.76 | 0.71 | 0.80 | 6.14 |
| LFQ-1k | 1024 | 0.82 | 0.77 | 0.86 | 4.35 |
| LFQ-2k | 2048 | 0.84 | 0.79 | 0.88 | 3.62 |
| LFQ-4k | 4096 | 0.86 | 0.82 | 0.91 | 3.31 |
| LFQ-8k | 8192 | 0.92 | 0.86 | 0.93 | 2.58 |
| LFQ-16k | 16384 | 0.92 | 0.87 | 0.94 | 2.32 |

B - struct tokens vs secondary struct

Figure 2: Reconstruction and secondary structure correspondence of structure tokenizers.

variant features $\mathbf{e} \in \mathbb{R}^{L \times d_{\text{quant}}}$, (2) a quantizer converts $\mathbf{e}$ into $\mathbf{z}$ of $L$ discrete tokens where $z_i \in \{0, 1, \ldots, |\mathcal{Z}|\}$ given a finite-size codebook $\mathcal{Z}$; and (3) a structure decoder reconstructs 3D coordinates $\tilde{\mathbf{x}}$ from the discrete tokens.

We utilize a GVP-based (Jing et al., 2020) structure encoder from pre-trained GVP-Transformer (Hsu et al., 2022) and a IPA-based (Jumper et al., 2021) structure decoder. In terms of quantizer, our preliminary experiment showed that conventional VQ-VAE pretty much struggles in training. To mitigate this, we instead adopts Lookup-Free Quantizer (LFQ) from the currently best visual tokenizer (Yu et al., 2023) to protein structure tokenization. Specifically, the latent space of LFQ is decomposed as the Cartesian product of single-dimensional binary variables, as $\mathbb{C} = \times_{k=1}^{\log_2 |\mathcal{Z}|} \mathcal{C}_k$, where $\mathcal{C}_k = \{-1, 1\}$. Given the encoded feature $\mathbf{e} = \text{encoder}(\mathbf{x}) \in \mathbb{R}^{L \times \log_2 |\mathcal{Z}|}$, each dimension (indexed by $k$) of the quantized representation $\text{quant}(e_i)$ is obtained from:

$$\text{quant}(e_i)[k] = \mathcal{C}_{i,k} = \text{sign}(e_i[k]) = -\mathbf{1}\{z_i[k] \le 0\} + \mathbf{1}\{e_i[k] > 0\}.$$

As such, with LFQ, the token indices for $\mathbf{z} = \{z_1, z_2, ..., z_i, ..., z_L\}$ is given by:

$$z_i = \text{index}(\text{quant}(e_i)) = \sum_{k=1}^{\log_2 |\mathcal{Z}|} 2^{k-1} \mathbf{1}\{e_i[k] > 0\}, \ \forall z_i \in \mathbf{z}.$$

The LFQ-based structure tokenizer is trained on the same structure dataset as mentioned before, using a combination of reconstruction, commitment, and entropy regularization losses, similar to standard VQ-VAE. Here FAPE loss (Jumper et al., 2021) is used as the primary reconstruction loss. (see §B.1 for more details.)

**Evaluation.** As shown in Fig. 2A, LFQ significantly outperforms VQ-VAE regarding reconstruction accuracy while training of LFQ is much faster than VQ-VAE (2 *vs.* 15 days on 8 A100s). Increasing codebook size leads to improved reconstruction while a codebook size of 8192 achieves the best compression-reconstruction trade-off. Meanwhile in Fig. 2B, we observe a strong correlation between structure tokens and secondary structures. For instance, a lot of structure tokens concentrated at the alpha helix and beta sheet vertices, while some tokens lie between regions. This suggests that structure tokens the fine-grained structural elements in backbone local environment.

## 4 EXPERIMENTS

In this section, we evaluate DPLM-2 on various generative and understanding scenarios, including unconditional protein generation (structure, sequence, and structure-sequence co-generation, §4.1), and a variety of conditional tasks, such as folding (§4.2), inverse folding (§4.3) and motif-scaffolding (§4.4), and a series of protein predictive tasks (§4.5).

### 4.1 UNCONDITIONAL PROTEIN GENERATION

The goal of unconditional protein generation is to produce both the 3D structure and amino acid sequence. Typically, this is done using a cascaded approach: either generating the structure first and then use another model to predict the sequence, or vice versa. Here, we focus on generating structure and sequence simultaneously. We evaluate DPLM-2 on both cascaded and simultaneous generation across three tasks: *unconditional structure generation*, *unconditional sequence generation*, and *structure-sequence co-generation*.

Following Multiflow (Campbell et al., 2024), we evaluate the generated proteins in terms of *quality*, *novelty* and *diversity*. **Designability** is measured through *self-consistency evaluation* and *foldability* (Yim et al., 2023; Watson et al., 2023; Wu et al., 2022a). Self-consistency evaluation is assessed by folding the generated sequence with ESMFold (Lin et al., 2022), then using `sc-TMscore` and `sc-RMSD` with the co-generated structure to evaluate similarity. Foldability is evaluated via ESMFold, with `pLDDT` > 70 considered plausible. **Novelty** is assessed by comparing generated structures to known ones in PDB using TMScore (`pdb-TM`), with lower values indicating greater novelty. **Diversity** is measured by calculating pairwise TMscore (`inner-TM`), where lower scores indicate more dissimilarity. The number of clusters identified by FoldSeek (van Kempen et al., 2023) also quantifies diversity, normalized by the total number of structures.

### 4.1.1 DPLM-2 ENABLES HIGH-QUALITY, DIVERSE AND NOVEL PROTEIN SEQUENCE AND STRUCTURE GENERATION

Tab. 2 and Fig. 3 present the results of DPLM-2 for unconditional protein generation. We highlight our key findings in the following aspects:

**(1) DPLM-2 can generate diverse and highly-plausible protein with simultaneous structure-sequence co-generation.** We sampled 100 proteins for each length in 100, 200, 300, 400, and 500. The co-generation can be performed in simultaneous generation (*co-generation*) and cascaded

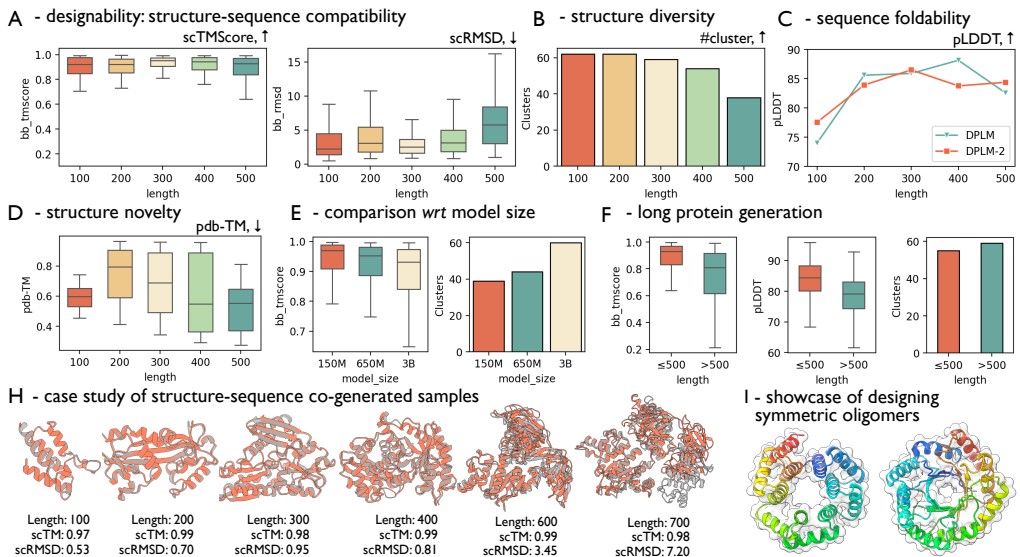

Figure 3: *Evaluation of* DPLM-2 *on unconditional structure-sequence co-generation. Here for designability of co-generated proteins, we use ESMFold to obtain refolded structure of* DPLM-2-*generated sequence and measure the structural similarity between* DPLM-2-*generated structure and the refolded structure, which aims to measure the compatibility of the co-generated structure and sequence pairs.*

workflow: first generating the structure then the sequence conditioned on generated structure (*struct → seq*), and the reverse way (*seq → struct*), without the need of other folding or inverse folding models. Fig. 3A/B demonstrates that DPLM-2 can sample sequence and structures with high designability across various lengths, with most sc-TM values exceeding 0.9, with diverse structure clusters. Fig. 3D shows that the novelty of sampled proteins, measured by pdb-TM, generally increases with longer protein lengths. In addition, DPLM-2 can generate with both modalities simultaneously or a modality-by-modality. As shown in Tab. 2, the co-generation performance exhibit highest scTM, suggesting that co-modeling indeed benefits protein generation.

**(2) DPLM-2 can attains competitive performance with strong baselines on co-generation, as well as backbone-only and sequence-only generation, respectively.** As shown in Tab. 2, DPLM-2 achieves the strong sc-TM compared to strong baselines, approaching the quality of native structures from PDB. We notice that ESM3-Open (Hayes et al., 2024), which runs in a sequence-then-structure order, fails short of unconditional generation. Compared to MultiFlow (Campbell et al., 2024), DPLM-2 achieves comparable co-generation quality. Notably, as also reported in Campbell et al. (2024), Multiflow falls short of sequence generation when directly trained from structures with native sequences, resulting in greatly degraded co-generation performance without data distillation from external inverse folding models (ProteinMPNN). For reference, we also provide the result of Multiflow retrained using our training data, where its co-generation performance remains unsatisfying and lags behind DPLM-2, which suggests that DPLM-2 has advantages of directly and effectively learning from complex structure-sequence joint distribution. Moreover, DPLM-2 can also only produce single modality if needed, where it matches the best competitive models in these settings respectively. These results demonstrate DPLM-2's effectiveness as a mulitmodal generative model.

**(3) DPLM-2 generates longer proteins beyond training data.** As DPLM-2 is trained with a 512 length cutoff, we are curious about its length extrapolation, and evaluate sampled proteins at lengths of [600, 700, 800, 900, 1000]. As shown in Fig. 3F, notably, for proteins exceeding the maximum training length of 512, the pLDDT scores of sequences sampled by DPLM-2 are close to those of DPLM. This suggests that DPLM-2 largely retains its original sequence generation capability inherited from sequence pre-training in DPLM, leading to its capability of length extrapolation.

**(4) Case study.** Fig. 3H shows some generated samples of DPLM-2 up to 700 residues, while in Fig. 3I we showcase that we can manipulate DPLM-2 to design symmetric oligomers by forcing to duplicate the predicted tokens with repetitive structure and sequence patterns.

**(5) Abaltion study on the training strategy.** We investigate the effects of warmup from the sequence-based pre-trained DPLM and data augmentation with high-quality AlphaFold-predicted structures on DPLM-2. The sequence pre-training significantly improve both designability and diversity, while

Table 2: *Benchmarking comparison of unconditional protein generation, in terms of structure-sequence co-generation, backbone-only generation, and sequence-only generation.* For each method, we generate 100 samples for lengths in [100, 200, 300, 400, 500]. * denotes Multiflow variants retrained by us using different dataset – native PDB data without ProteinMPNN distillation and the same training data as DPLM-2 (*i.e.*, PDB+SwissProt), respectively.

| | Structure-sequence Consistency | | | Novelty | Diversity | |
| --- | --- | --- | --- | --- | --- | --- |
| | scTM (↑) | scRMSD (↓) | pLDDT (↑) | avg. pdb-TM (↓) | avg. inner-TM (↓) | MaxCluster (↑) |
| **Structure-sequence co-generation.** | | | | | | |
| Native PDB protein | 0.904 ± 0.129 | 4.623 ± 5.688 | – | – | 0.262 ± 0.025 | 0.776 |
| ESM3-Open (1.4B, seq → struct) | 0.624 ± 0.232 | 24.180 ± 24.109 | – | 0.660 ± 0.000 | 0.220 ± 0.046 | 0.540 |
| MultiFlow *w/* distillation (official ckpt) | **0.930** ± 0.098 | 3.208 ± 4.741 | 79.447 | 0.704 ± 0.000 | 0.356 ± 0.032 | 0.500 |
| *MultiFlow *w/o* distillation | 0.750 ± 0.163 | 9.306 ± 8.499 | 61.519 | – | 0.350 ± 0.038 | 0.490 |
| *MultiFlow (retrained on our training data) | 0.871 ± 0.934 | 6.580 ± 6.258 | 62.624 | – | 0.331 ± 0.052 | 0.440 |
| DPLM-2 (650M, seq → struct) | 0.907 ± 0.117 | 6.337 ± 9.403 | 82.246 | 0.653 ± 0.195 | 0.280 ± 0.038 | 0.651 |
| DPLM-2 (650M, struct → seq) | 0.921 ± 0.098 | 4.969 ± 6.735 | 81.910 | 0.637 ± 0.195 | 0.308 ± 0.089 | 0.575 |
| **DPLM-2 (650M, co-generation)** | 0.925 ± 0.085 | 3.899 ± 3.723 | 82.686 | 0.640 ± 0.204 | 0.287 ± 0.030 | 0.545 |
| **Unconditional backbone generation.** (sequence predicted by ProteinMPNN) | | | | | | |
| Native PDB struct. (seq. from PMPNN) | 0.969 ± 0.000 | 0.864 ± 0.000 | – | – | 0.262 ± 0.025 | 0.782 |
| FrameDiff | 0.818 ± 0.000 | 3.919 ± 0.000 | – | 0.668 ± 0.000 | 0.444 ± 0.064 | 0.252 |
| FoldFlow | 0.540 ± 0.000 | 7.965 ± 0.000 | – | 0.566 ± 0.000 | 0.286 ± 0.023 | 0.762 |
| RFDiffusion | 0.914 ± 0.000 | 1.969 ± 0.000 | – | 0.657 ± 0.000 | 0.352 ± 0.038 | 0.598 |
| **DPLM-2 (650M)** | 0.945 ± 0.082 | 4.451 ± 5.261 | – | 0.637 ± 0.195 | 0.297 ± 0.049 | 0.575 |
| **Unconditional sequence generation.** (structures predicted by ESMFold) | | | | | | |
| EvoDiff | – | – | 35.846 | 0.432 ± 0.106 | 0.265 ± 0.025 | 0.990 |
| DPLM (650M) | – | – | 83.252 | 0.541 ± 0.187 | 0.242 ± 0.041 | 0.735 |
| **DPLM-2 (650M)** | – | – | 82.246 | 0.662 ± 0.199 | 0.280 ± 0.042 | 0.700 |

Figure 4: *Analysis regarding secondary structure of generated proteins.* **(A)** Statistics of averaged proportions of secondary structures for proteins from different methods and PDB; **(B)** Secondary structure *vs.* designability; **(C)** Samples of Multiflow, PDB and DPLM-2, as well as their secondary structure distributions.

data augmentation can further enhance the designability, especially for long proteins. For more details of ablation study, please refer to §A.6.

### 4.1.2 DPLM-2 GENERATES PROTEINS THAT RESEMBLES NATURAL PROTEINS

To further analyze the properties of different model, we examine their secondary structure distribution against natural proteins from PDB.

**Proteins sampled by DPLM-2 have secondary structures most similar to natural proteins.** As seen in Fig. 4A, structure-based models like RFDiffusion and MultiFlow generate proteins with more helices and fewer sheets and loops than natural proteins in PDB. Protein language models like ESM3 and DPLM-2 show no strong bias towards alpha helices, but ESM3 tends to generate more loops. Among the methods, DPLM-2 produces the most natural-like secondary structure proportions, closely matching PDB proteins. In Fig. 4C, proteins generated by MultiFlow contain many helices and become more globular as length increases, exhibiting idealized secondary structures. In contrast,

proteins generated from DPLM-2 resembles natural ones have more balanced structures, with fewer helices and more beta sheets and loops. On the other hands, simplex plots in Fig. 4C shows that while MultiFlow's proteins are clustered in helix-rich regions, DPLM-2's proteins span a wider area similar to natural proteins, while it rarely samples proteins composed mostly of sheets and loops, which do occur in nature. Additionally, Fig. 4B shows that the loop ratio has a significant impact on designability, where a higher proportion of loops will increase `scRMSD`, as loops are highly flexible. Thus, proteins with long loops, which DPLM-2 often generates, tend to have relatively high `scRMSD`, aligning with the results in Tab. 2.

## 4.2 FORWARD FOLDING (SEQUENCE-CONDITIONED STRUCTURE PREDICTION)

The goal of folding is to predict the 3D structure for the given amino acid sequence (Jumper et al., 2021). As a mulitmodal generative model, DPLM-2 spontaneously enables protein structure prediction task (see Fig. 1C-3) given sequence as conditioning. We assess DPLM-2 on CAMEO 2022 and a PDB data split used by Multiflow (Campbell et al., 2024). We utilize `RMSD` and `TMscore` between predicted and ground truth structure for evaluation, while DPLM-2 adopts `argmax` decoding for 100 sampling iterations.

**Tab. 3 indicates that DPLM-2 can perform sufficiently good folding in a zero-shot manner.** Performance can be improved after further supervised fine-tuning (SFT) using folding objective ($\max_\theta \log p_\theta(\mathbf{z}|\mathbf{s})$). Overall, DPLM-2 can outperform or on par with the strong baselines, while achieving close performance with ESMFold. Plus, we observe that DPLM-2 with larger model scales can attain better results than smaller ones. We suggest that DPLM-2 benefits from the evolutionary information inherited from DPLM pre-trained on the vast number of protein sequences, which can be transferred and leveraged into structure modeling.

Table 3: *Structure prediction performance comparison between* DPLM-2 *and different baseline approaches on CAMEO 2022 datasets.* †: PVQD results are quoted from Liu et al. (2023).

| Models | CAMEO 2022 | | PDB date split | |
|---|---|---|---|---|
| | RMSD | TMscore | RMSD | TMscore |
| ESMFold | 3.99/2.03 | 0.85/0.93 | 2.84/1.19 | 0.93/0.97 |
| †PVQD | 4.08/1.95 | 0.81/0.88 | – | – |
| MultiFlow | 17.84/17.96 | 0.50/0.46 | 15.64/16.08 | 0.53/0.49 |
| ESM3 | 6.33/2.98 | 0.85/0.92 | 4.94/2.28 | 0.87/0.93 |
| DPLM-2 (150M) | 9.22/7.64 | 0.75/0.81 | 8.35/5.60 | 0.76/0.82 |
| *w/* folding SFT | 7.66/4.37 | 0.80/0.86 | 6.00/3.41 | 0.83/0.88 |
| DPLM-2 (650M) | 7.37/4.89 | 0.79/0.86 | 5.67/3.33 | 0.83/0.88 |
| *w/* folding SFT | 6.21/3.78 | 0.84/0.89 | 3.40/1.78 | 0.89/0.94 |
| DPLM-2 (3B) | 6.34/3.65 | 0.83/0.89 | 4.54/2.54 | 0.86/0.92 |
| *w/* folding SFT | 5.71/3.23 | 0.85/0.90 | 3.15/1.69 | 0.90/0.95 |

## 4.3 INVERSE FOLDING (STRUCTURE-CONDITIONED SEQUENCE GENERATION)

The goal of inverse folding is to find an amino acid sequence that can fold to a given backbone structure. For evaluation, we employ amino acid recovery (`AAR`) for sequence evaluation, and we also assess the structure by self-consistency TM-score (`scTM`) between the native structure and the ESMFold-predicted structure of the generated sequence.

**DPLM-2 can generate reasonable sequences that fold into the given structures.** Tab. 4 presents that DPLM-2 can outperform or be on par with other co-generation models (MultiFlow, ESM3). As the model size increases, the performance in terms of sequence recovery (`AAR`) and structural consistency (`scTM`) improves, revealing the same scaling law observed in the folding

Table 4: *Comparison on inverse folding task.*

| Models | CAMEO 2022 | | PDB date split | |
|---|---|---|---|---|
| | AAR | scTM | AAR | scTM |
| MultiFlow | 32.28/33.58 | 0.87/0.94 | 37.74/37.59 | 0.94/0.96 |
| ESM3 | 47.06/46.24 | 0.90/0.95 | 49.50/49.42 | 0.94/0.97 |
| DPLM-2 (150M) | 45.22/46.12 | 0.87/0.93 | 48.83/47.96 | 0.89/0.95 |
| DPLM-2 (650M) | 49.01/50.10 | 0.88/0.93 | 54.80/53/07 | 0.91/0.96 |
| DPLM-2 (3B) | 52.36/53.72 | 0.89/0.95 | 61.67/57.91 | 0.92/0.96 |

task. We suggest that multimodal training effectively aligns the structure and sequence into the same space, such that DPLM-2 can yield the corresponding sequence without additional training.

## 4.4 SCAFFOLDING WITH MIXED-MODAL MOTIF CONDITIONING

The objective of motif-scaffolding is to generate a suitable scaffold to preserve the structure of the given motif and maintain its original function. We follow the experimental setting of Yim et al. (2024), with 24 motif-scaffolding problems and we sample 100 scaffolds for each motif, where we (1) first determine the length of scaffold, and then (2) keep the motif segment unchanged and sample the scaffold part conditioned on the motif. The scaffold length is sampled from a range provided by Yim et al. (2024), and when there are multiple motifs, the order of motif segments is consistent with Yim et al. (2024). We provide the 3D structure and sequence of motif as input of DPLM-2. As a multimodal model, we evaluate DPLM-2 using sequence-based, structure-based, and co-generation approaches. A scaffold is considered successful if it satisfies both criteria (1) overall designablity, which is successful when `pLDDT` > 70 (for sequence-based models) or `scTM` > 0.8, and (2) motif-preseving, which is deemed successful when the predicted motif structure matches the native one with `motif-RMSD` <1Å.

Fig. 5 reveals that DPLM-2 is capable of generate reasonable scaffolds for the given functional motifs. In sequence-based, structure-based and co-generation evaluation, DPLM-2 can outperform or be on par with the corresponding approaches in most cases, solving more motif problem and achieving higher average success rate. We compared to sequence-based method, DPLM-2 shows better performance since it allows structural input of motif, which is important for preserving motif's structure hence the functions. Remarkably, DPLM-2 attains comparable performance with RFDiffusion when only generating scaffold structure, while achieve better performance when simultaneously designing scaffold sequence and structure, outperforming ESM3. Despite not experimentally verified, these results suggest that with DPLM-2, mulitmodal conditioning and generation could lead to more successful conditional protein design.

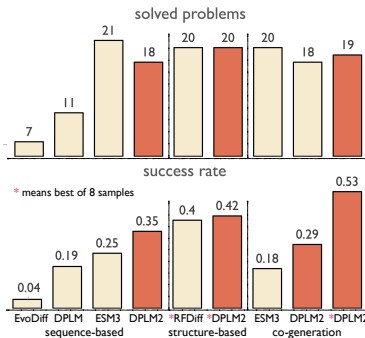

Figure 5: *Evaluation of motif-scaffolding w.r.t. success rate and num. of solved problems.*

## 4.5 EVALUATION OF PROTEIN REPRESENTATION LEARNING

Directly access to structure information is supposed to benefit downstream protein predictive tasks. To inspect this, we evaluate DPLM-2 on a variety of protein predictive tasks utilizing the dataset provided by SaProt (Su et al., 2023), where we provide tokenized protein structure tokens along with the protein sequences to DPLM-2.

DPLM-2 can perform multimodal representation learning by leveraging both structure and sequence information. Tab. 5 presents that DPLM-2 shows further improvement compared to sequence-only methods (ESM2, DPLM) on some tasks, indicating that DPLM-2 can leverage protein structures to generate better representations containing multimodal information for downstream tasks. However, we find that DPLM-2 falls behind the state-of-the-art structure-aware protein LM, *i.e.*, SaProt, in most tasks and even lags behind DPLM in certain tasks. We hypothesize this is because the training data of DPLM-2, consisting of PDB and SwissProt, is smaller and differs from UniRef50, which DPLM is pretrained on, potentially causing catastrophic forgetting and suboptimal representation. To test this, we conducted an ablation study on specific tasks where DPLM-2 underperforms compared to DPLM. We observe that without large-scale sequence pretraining, DPLM-2 outperforms DPLM significantly, suggesting that: (1) Incorporating structure information enhances performance over sequence-only models. (2) Smaller datasets can lead to catastrophic forgetting, diminishing the benefits of large-scale pretraining. Please refer to §E for more details.

Table 5: *Performance on various protein predictive downstream tasks.* †: benchmarked results are quoted from Su et al. (2023).

| Models | Thermostability | HumanPPI | Metal Ion Binding | EC | GO | | | DeepLoc | |
|---|---|---|---|---|---|---|---|---|---|
| | | | | | MF | BP | CC | Subcellular | Binary |
| | Spearman's $\rho$ | Acc (%) | Acc (%) | Fmax | Fmax | Fmax | Fmax | Acc (%) | Acc (%) |
| †SaProt (650M) | **0.724** | **86.41** | **75.75** | **0.884** | *0.678* | 0.356 | **0.414** | **85.57** | *93.55* |
| †MIF-ST (Yang et al., 2022b) | 0.694 | 75.54 | 75.08 | 0.803 | 0.627 | 0.239 | 0.248 | 78.96 | 91.76 |
| †GearNet (Zhang et al., 2023) | 0.571 | 73.86 | 71.26 | 0.871 | 0.650 | 0.354 | 0.404 | 69.45 | 89.18 |
| ESM2 (650M) | 0.691 | *84.78* | 71.88 | 0.866 | 0.676 | 0.344 | 0.402 | 83.68 | 92.28 |
| DPLM (650M) | 0.695 | **86.41** | *75.15* | 0.875 | **0.680** | *0.357* | 0.409 | *84.56* | 93.09 |
| DPLM-2 (650M) | *0.714* | 84.44 | 74.28 | *0.878* | **0.680** | **0.359** | *0.411* | 82.98 | **93.64** |

## 5 DISCUSSIONS

In this paper, we introduce DPLM-2, a multimodal diffusion protein language model that understands, generates and reasons over protein structure and sequence, aiming to severe as a mulimodal foundation for protein. Despite promising performance spanning protein co-generation, folding, inverse folding and conditional motif-scaffolding with mulimodal input and output, there remains several limitations deserving to be addressed. (1) Structure data: Our findings indicate that while structure awareness may help with predictive tasks, the limited structure data constrains DPLM-2's ability to learn robust representations. It is also important to account for longer protein chains and multimers in future studies. (2) Trade-off of discrete latent representation: Tokenizing structure into discrete symbols facilitates multimodal protein language models and co-generation but may come at the cost of losing fine-grained structural details and control, such as precise atomic positions and inter-atomic distances. Future work should aim to also integrate the strengths of data-space structure-based generative models into sequence-based multimodal language models to maximize the best of both worlds.

ACKNOWLEDGEMENT

We would like to thank Dr. Hang Li for insightful discussions on the project and feedback on the manuscript that help shape this study. We thank Yi Zhou, Jing Yuan, Yilai Li, Yuning Shen, Wesley Hsieh and Daiheng Zhang for their valuable comments.

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

# A  DPLM-2 TRAINING

## A.1  IMPLEMENTATION DETAILS

DPLM-2 takes the discrete structure token sequence and amino acid token sequence as input. As demonstrated in Fig. 1, we concatenate the two sequences into one sequence of double length. DPLM-2 employs an efficient warm-up strategy by initializing with pre-trained sequence-based DPLM (§3.2) to leverage the evolutionary information learned by DPLM for protein structure modeling. Considering that the vocabulary of DPLM only consists of amino acids, we expand the vocabulary of DPLM-2 with discrete structure tokens. The embeddings for these new tokens are initialized using the mean and standard deviation of the learned amino acid embeddings. This embedding initialization keeps the distributional statistics of the embedding space consistent with the pre-trained DPLM, ensuring stable early-stage training (for learning structure-sequence alignment) and reducing the risk of extreme gradients that could cause training instability.

## A.2  DISTINCT NOISE SCHEDULER OF TRAINING

We introduce a distinct scheduler to control the noise level of structure and sequence flexibly during training (§3.1). Different combinations of structures and sequence schedulers (denoted as $t_z$ and $t_s$, respectively) imply training for different applications. Specifically, we mainly focus on (1) sequence-conditioned structure generation (e.g., folding), (2) structure-conditioned sequence generation (e.g., inverse-folding), (3) sequence generation, (4) structure generation, (5) structure-sequence co-generation, as shown in Tab. 1. For conditional generation tasks (e.g., folding and inverse-folding), we set the noise scheduler of the conditioned modality to 0, e.g., no noise in the conditioned modality. Specifically, in the folding task, the $t_s$ is always set to 0, while in the inverse-folding tasks the $t_z$ is always set to 0. In the structure-sequence co-generation task, we keep the $t_z$ and $t_s$ for the same, enhancing the structure-sequence consistency in co-generation. The structure or sequence generation tasks do not depend on another modality, so we set the noise scheduler of another modality to $T$, e.g., 100% noise in another modality. For example, in structure generation task, the $t_s$ is always set to $T$.

During training, we train the above 5 tasks simultaneously. We divide the training data in a batch into 5 parts according to a preset proportion, and each part is used for a specific task training. In our experiment, the proportion for each task is the same, which is 20%. After training, we can further enhance a specific generation task by supervised finetuning (SFT). This involves continuing training for the specific task with a proportion of 100%, while the proportion for other tasks is set to 0%. For example, in Tab. 3, the folding supervised finetuning is performed by continue training with folding task based on a pre-trained DPLM-2 with 100% proportion of training data.

## A.3  HYPERPARAMETER

We train all models using AdamW optimizer (Kingma & Ba, 2015) with $\beta_1 = 0.9$ and $\beta_2 = 0.95$. We use a weight decay of 0.01 and gradient clipping of 0.5. We employ 2K warmup steps until reaching the maximum learning rate, and utilize a linear decay scheduler to decay LR to 10% of the maximum learning rate by the end of training. The maximum learning rate is 1e-4, and the overall training step is 100,000. We utilize the pretrained DPLM as the parameter initialization, and the diffusion timestep is set to 500. We train 150M DPLM-2 with 8 A100 GPUs for 3 days, while 650M with 16 A100 GPUs for 3 days and 3B with 16 A100 GPUs for a week.

## A.4  DATASET

The training set of DPLM-2 is composed by experimental data, *i.e.*, PDB (Berman et al., 2000), and high quality synthetic data, *i.e.*, SwissProt (Varadi et al., 2022). We filter the SwissProt data by pLDDT > 85. After filtering, the overall training set contains approximately 200,000 proteins. We limit the maximum length of the training set to 512. For proteins longer than 512, we randomly crop it to 512. We crop the low pLDDT (pLDDT < 50) segments located at the both ends of proteins in the SwissProt dataset. These segments are typically non-structural and may negatively impact the training results. Moreover, we find that the length distribution of the training set is not balanced, where the number of proteins with length less than 100 is relatively small, leading to a suboptimal diversity among the short proteins. Therefore, during training, we randomly crop long proteins to short proteins with a probability of 50% for each batch to improve the diversity.

## A.5  TACKLING EXPOSURE BIAS IN DISCRETE DIFFUSION WITH SELF-MIXUP TRAINING STRATEGY

The exposure bias problem, which is described as the input mismatch between training and sampling, has already garnered attention in the research of continuous diffusion (Ning et al., 2023a;b; Li et al.,

2023) and NLP (Ranzato et al., 2016; Bengio et al., 2015). We find that the discrete diffusion model also encounters this issue. According to the Eq.1, the model is trained to be tasked with $p_\theta(\mathbf{x}_i^{(0)}|\mathbf{x}^{(t)})$, essentially doing masked-prediction. During training, the model makes prediction conditioned on the $\mathbf{x}^{(t)}$, which is a mixture of ground-truth and mask tokens as noise: $\mathbf{x}^{(t)} = \alpha_t \mathbf{x}^{(0)} + (1 - \alpha_t)\mathbf{q}_{\text{noise}}$. However, during inference, the model predicts $p_\theta(\mathbf{x}_i^{(0)}|\hat{\mathbf{x}}^{(t)})$ conditioned on the previously generated sample $\hat{\mathbf{x}}^{(t)}$, which is a mixture of model prediction and masks, essentially requiring denoising and masked-prediction. The difference between $\mathbf{x}^{(t)}$ and $\hat{\mathbf{x}}^{(t)}$ causes a discrepancy between $p_\theta(\mathbf{x}_i^{(0)}|\mathbf{x}^{(t)})$ and $p_\theta(\mathbf{x}_i^{(0)}|\hat{\mathbf{x}}^{(t)})$, potentially leading to error accumulation since the model trend to be over-confident on its predictions (as in training the model is always exposed to ground-truth, hence the name exposure bias), and negatively impacting the generation performance.

To mitigate this, we propose to bridge this gap by training model to make predictions conditioned on its own predicted results: (1) predict $\hat{\mathbf{x}}^{(0)}$ conditioned on the ground truth sample $\mathbf{x}^{(t)}$, (2) construct the generated sample: $\hat{\mathbf{x}}^{(t)} \leftarrow \hat{\mathbf{x}}^{(0)} + (1 - \alpha_t)\mathbf{q}_{\text{noise}}$, (3) compute self-mixup loss according to Eq. 1:

$$\hat{\mathcal{J}}_t = \mathbf{E}_{q(\mathbf{x}^{(0)})}\left[\lambda^{(t)}\sum_{1 \le i \le L} b_i(t) \cdot \log p_\theta(\mathbf{x}_i^{(0)}|\hat{\mathbf{x}}^{(t)})\right]$$

We can illustrate this more clearly with a break-down example. Let the ground-truth $\mathbf{x}^{(0)}$ be `ABCDE`, and the $\mathbf{x}^{(t)}$ be `[m][m][m]DE` as in masked discrete diffusion, where `[m]` represents the mask token. The training process is as below:

(i) Call a model forward to obtain model prediction $\hat{\mathbf{x}}^{(0)}$, which is `abcDE` (with the ground truth token `DE` preserved for masked positions), where `abc` represent model prediction by argmax decoding.

(ii) Construct self-mixup $\hat{\mathbf{x}}^{(t)}$. In our experiments, we always replace the ground-truth token in $\hat{\mathbf{x}}^{(0)}$ (`DE` in this case) with the mask token `[m]`. Therefore $\hat{\mathbf{x}}^{(t)}$ becomes `abc[m][m]`.

(iii) Compute self-mixup loss, which is essentially cross entropy loss between $p_\theta(\mathbf{x}_i^{(0)}|\hat{\mathbf{x}}^{(t)})$ and $\mathbf{x}^{(0)}$ at all positions.

More specifically, this can be seen as mask positions are applied masked language modeling loss while non-masked positions are applied denoising autoencoder loss. Moreover, this also improves sample-efficiency compared to typical masked discrete diffusion where training loss is only applied to mask positions.

In our experiments, we first train DPLM-2 with the original loss in Eq. 1 for 50,000 steps to ensure the prediction quality. This step is crucial; otherwise, the model's predictions might be poor, leading to an excessively large self-mixup loss and causing training instability. After this initial phase, we continue training with self-mixup loss to mitigate the exposure bias issue. Tab. 6 shows that the *self-mixup* training strategy effectively enhances the diversity of samples. We conduct experiments with the DPLM-2 650M model on the unconditional generation task. We sample 100 proteins within each length interval and calculate scTM for structure-sequence compatibility and the number of clusters for diversity. We attribute this to the model producing more accurate logits during inference, leading to more diverse reasonable sampling paths instead of converging on the sampling paths with the highest probability, which results in more diverse proteins.

Table 6: Ablation study on the *self-mixup* training strategy.

| Mixup strategy | length 100 | | length 200 | | length 300 | | length 400 | | length 500 | |
|---|---|---|---|---|---|---|---|---|---|---|
| | scTM | clusters | scTM | clusters | scTM | clusters | scTM | clusters | scTM | clusters |
| ✗ | **0.9237** | 44 | **0.9180** | 53 | 0.9147 | 48 | 0.9059 | 42 | **0.8896** | 33 |
| ✓ | 0.8812 | **62** | 0.8820 | **62** | **0.9172** | **59** | **0.9099** | **54** | 0.8845 | **38** |

### A.6 ABLATION STUDY ON THE SEQUENCE PRE-TRAINING AND SYNTHETIC STRUCTURES

In DPLM-2 training, we start with a warmup from the sequence-based pre-trained DPLM to exploit established evolutionary information and augment the data with high-quality AlphaFold-predicted structures from SwissProt (around 200K) and clustered PDB structures. This section evaluates the effects of sequence pre-training and data augmentation on unconditional protein generation.

We investigate the effect of sequence pre-training by randomly initializing DPLM-2 instead of using DPLM parameters, while for effect of synthetis structures we leverage PDB structures only for training. We conduct experiments on 150M DPLM-2, for each DPLM-2 variant we sample

Table 7: Ablation study on the sequence pre-training and training data augmentation.

| sequence pre-training | synthetic structures | length 100 | | length 200 | | length 300 | | length 400 | | length 500 | |
|---|---|---|---|---|---|---|---|---|---|---|---|
| | | scTM | clusters | scTM | clusters | scTM | clusters | scTM | clusters | scTM | clusters |
| ✗ | ✗ | 0.9241 | 20 | 0.8674 | 34 | 0.7667 | 33 | 0.5016 | 25 | 0.4511 | 25 |
| ✓ | ✗ | **0.9610** | 26 | 0.9349 | **47** | 0.9169 | 38 | 0.8643 | 35 | 0.7673 | **52** |
| ✗ | ✓ | 0.8988 | 27 | 0.9182 | 15 | **0.9343** | 13 | 0.8518 | 21 | 0.8288 | 31 |
| ✓ | ✓ | 0.9348 | **35** | **0.9428** | 40 | 0.9232 | **48** | **0.9260** | **40** | **0.9012** | 32 |

Table 8: Results of codebook size, codebook utilization and `lddt_ca` on the training set and valid set of different structure tokenizers.

| tokenizer | codebook size | codebook utilization | training set | valid set (CAMEO 2022) | | |
|---|---|---|---|---|---|---|
| | | | `lddt_ca` | `lddt_ca` | `TM-score` | `RMSD` |
| VQ-VAE-1K | 1024 | 63.50% | 0.76 | 0.71 | 0.80 | 6.14 |
| LFQ-1K | 1024 | 100.00% | 0.82 | 0.77 | 0.86 | 4.35 |
| LFQ-2K | 1024 | 100.00% | 0.84 | 0.79 | 0.88 | 3.62 |
| LFQ-4K | 4096 | 100.00% | 0.86 | 0.82 | 0.91 | 3.31 |
| LFQ-8K | 8192 | 99.50% | 0.92 | 0.86 | 0.93 | 2.58 |
| LFQ-16K | 16384 | 98.60% | 0.92 | 0.87 | 0.94 | 2.32 |

100 examples for each length in 100, 200, 300, 400 and 500. We compute `scTM` and the number of difference clusters in each length. Tab. 7 demonstrates that *sequence pre-training and data augmentation can significantly improve the designability and diversity*, especially in generating long proteins (length $> 300$). We hypothesize that the limited number of long proteins in PDB leads to insufficient training. In contrast, sequence pretraining, which includes evolutionary data, is essential and can be transferred to improve protein structure modeling and generation quality. Additionally, this evolutionary information boosts sampling diversity. While increasing the amount of training data improves designability, it is less effective in enhancing diversity compared to sequence pretraining. By combining both strategies, we achieve the best overall performance, which forms the core of our training strategy.

# B    DISCUSSIONS ON THE STRUCTURE TOKENIZATION

## B.1    IMPLEMENTATION DETAILS

We utilize a GVP-based (Jing et al., 2020) structure encoder from pre-trained GVP-Transformer (Hsu et al., 2022) with its parameters frozen during training. The structure encoder transforms backbone structures into geometric features, which are projected onto a latent embedding using an MLP layer. The structure decoder follows the IPA-based modules from AlphaFold2 (Jumper et al., 2021), using 4 EvoFormer layers without MSA row attention, following ESMFold (Lin et al., 2022), to generate atomic positions from the structure tokens. We train structure tokenizer using the same structure data as our mulitmodal language model, containing both experimental and high-quality structures. The training objective of structure tokenizer includes reconstruction loss, codebook commitment loss, and entropy regularization loss to ensure effective codebook utilization. For the reconstruction loss, we adopt the FAPE loss, violation loss, and distogram loss from AlphaFold2 (Jumper et al., 2021), measuring the difference between predicted and native structures. To further enhance the training, we introduce a sequence prediction head on top of the structure decoder's final representation and minimize the cross-entropy against the native sequence.

## B.2    THE UTILIZATION AND INTERPRETABILITY OF STRUCTURE TOKENS

In the Fig. 2A we have shown the reconstruction accuracy and an interpretation analysis on the correspondence of structure tokens and local structural elements in terms of secondary structures. We also calculate the codebook utilization in Tab. 8. We find that LFQ-based tokenizers always achieve nearly 100% codebook utilization with more evenly distributed code usage, while vanilla VQ-VAE struggles with codebook collapse. The Fig. 2B demonstrates the interpretability of structure tokens with a informative simplex plot of structure tokens vs second structure. We can observe a strong correlation between a vast majority of the structure tokens and structured local environments, where a lot of structured tokens concentrate on the alpha helix and beta sheet vertices, while some tokens lie between regions or the loop vertex. There are also a subset of structure tokens having less clear clues to specific secondary structures. This suggests that structure tokens mostly capture clear secondary elements, some may correspond to structured local environments (in bewteen helic and sheet), while others could be high-level abstract entities.

Figure 6: The histogram of `num_structural_motifs` v.s. `motif_clusters` for each structure token. We randomly sampling 500 out of 8,192 structure tokens for readability.

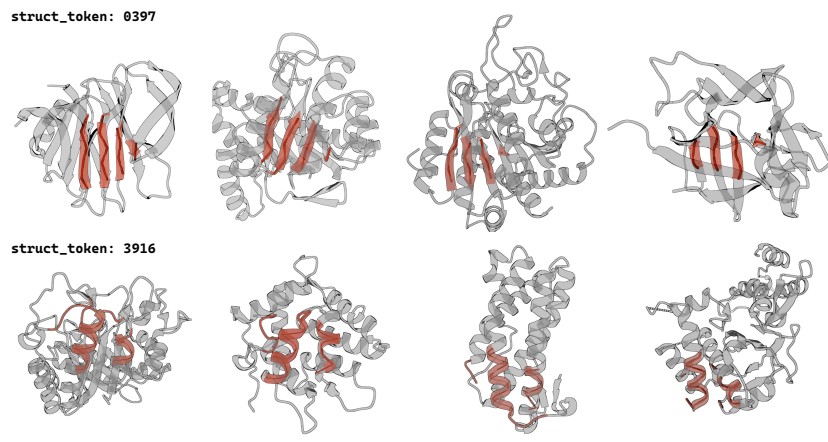

Figure 7: Visualization of mapping structure tokens to structural motifs.

## B.3 VISUALIZATION OF MAPPING STRUCTURE TOKENS TO STRUCTURAL MOTIFS

We provide more fine-grained insights into what structure tokens learn by mapping structure tokens to structural motifs. Specifically, as structure tokens are residue-wise representations, we aim to map each structure token to structural motifs defined as the nearest-neighbor local structural environment of a residue in the training dataset. For efficiency, we used only the PDB dataset. The process is as follow:

(1) For each structure in the PDB dataset (approximately 20K in total), we first tokenize the structure into structure tokens and save the pair of structure tokens and 30-nearest-neighbors structural motifs for each residue. We use 30 nearest neighbors because the pre-trained GVPTransformerEncoder, which we used as the structure encoder, employs 30 nearest neighbors as the hyperparameter for geometric features.

(2) After processing all structures, we obtain a table where each row corresponds to a structure token and its associated structural motifs (i.e., `num_structural_motifs`)

(3) To analyze whether a structure token tends to occur in a similar local structural environment, we use Foldseek (TM-threshold = 0.5) to cluster the structural motifs for each structure token (i.e., `motif_clusters`). Although Foldseek may not be entirely accurate in clustering such short and discontinuous structural regions, it provides a reasonable comparative sense of the similarity or difference among all structural motifs associated with each structure token.

In Fig. 6, we plot the histogram of `num_structural_motifs` v.s. `motif_clusters` for each structure token (randomly sampling 500 out of 8,192 structure tokens to ensure readability). From the visualization, we observe that many structure tokens correspond to highly similar structural motifs (evidenced by a small ratio of the number of motif clusters to the number of total structural motifs), while others exhibit a high degree of ambiguity.

Additionally, we visualize the mapping between structure tokens and structural motifs in specific cases. In Fig. 7, we showcase two structure tokens and their corresponding similar structural motifs

across four different PDB structures, illustrating the diversity or consistency in the mapped local structural environments.

### B.4 Limitation on the structure tokenization

Our approach can be seen as decoupling the learning of the structurally invariant topology to the tokenizer and the language model, and the geometric reasoning on 3D coordinates to the structure decoder parameterized by triangular modules and IPAs from AlphaFold2 with FAPE loss. This shares the similarity to AF2 and ESMfold, where the sequence encoding modules like Evoformer (co-evolution encoding) and ESM (amino-acid encoding) provide invariant features (in the form of single and pair embeddings) to the structure decoder that learns to convert invariant features into 3D coordinates. The AF2-style structure decoder does not enforce strict equivariance to rigid transformations. Instead, it relies on the FAPE loss to ensure structural consistency, which minimizes coordinate errors in a manner that is invariant to global rotations and translations.

*As such, we suggest that the primary trade-off when using invariant structure tokens instead of 3D coordinates mainly lies in the potential loss of fine-grained structural details.* Albeit being the key enabler to multimodal PLMs, structure tokenization is essentially clustering similar local structured environments, which results in lossy compression and the absence of fine-grained structural variation. The primary principle of the solution is that we need to "recover" and preserve the high-frequency variation that gets lost during quantization. We propose some potential directions for mitigation:

**Separate structure encodings for DPLM-2**. We can introduce different structure encoders for encoding and generation purposes, respectively. For parts of a protein where atomic coordinates are already provided, lossy tokenization may not be necessary. Instead, we can use robust features from powerful structure encoders like GVP-Transformer while continuing to use structure tokens for generating the remaining parts. To achieve this, the model can be trained to alternate between these two types of encodings. A similar approach has been applied successfully in recent vision-language MLLMs (Wu et al., 2024), as the vision-language community has also recognized that understanding and generation often require different types of representations.

**Modeling continuous structure features with hybrid tokenization.** In the structure tokenizer, the vector quantizer module converts encoder features into discrete structure token features, but the residuals—differences between the original and quantized features—are lost, removing fine-grained structural details. To address this, we can using continuous generative modeling, such as diffusion/flow-based models, to learn to recover these residuals. This would work by conditioning on the structure tokens and possibly the final hidden states of DPLM-2. The protein structure generation process would involve first generating discrete structure tokens that capture the overall topology, then using those tokens to generate the missing residuals. These residuals would be added up to the structure token embeddings to recover a more complete and accurate structure representation, closer to the features produced by the structure encoder. This approach could significantly improve structure generation. By combining this idea with hybrid structure encodings, DPLM-2 could not only interpret given structures at atomic accuracy but also generate structures that include the missing fine-grained variations. Similar strategies have shown significant success in visual autoregressive generation with visual tokenizers (Tang et al., 2024).

## C Analysis on the sampling strategy

We utilize argmax decoding for conditional generation tasks (e.g., folding and inverse folding) to maximize generation accuracy and ensure a fair comparison with DPLM. On the other hand, stochastic sampling was employed for unconditional generation or motif-scaffolding tasks to encourage generation diversity while maintaining good generation quality.

Specifically, we utilize a temperature-based stochastic approach. We mainly focus on the temperature-annealed version based on the sampling procedure of DPLM (Wang et al., 2024) for better sampling diversity. The overall sampling approach is shown in algorithm 1. The temperature annealing sampling approach introduces more randomness during the initial stage of sampling by using a large temperature, and more fidelity during the final stage of sampling by using a small temperature. This method improves generation diversity while maintaining generation quality.

Moreover, we observe that stochastic sampling could also improve the generation diversity in conditional tasks (e.g., inverse folding) while keeping the quality. As shown in Tab. 9, the argmax decoding strategy picks the token with highest probability at each timestep, yielding sequence with high probability and resulting in high amino acid recovery (AAR). On the other hand, we employ a sampling strategy with annealing temperature from 2.2 to 0.1 to improve diversity, and the generated

Table 9: Ablation study on the sampling strategy in inverse folding task.

| Model | CAMEO 2022 | |
| --- | --- | --- |
| | AAR | scTM |
| MultiFlow | 32.28/33.58 | 0.87/0.94 |
| ESM3-Open | 47.06/46.24 | 0.90/0.95 |
| DPLM2 650M (argmax decoding) | 49.01/50.10 | 0.88/0.93 |
| DPLM2 650M (temperature-annealed sampling) | 43.15/42.24 | 0.88/0.93 |

sequence has a lower AAR while maintaining the same scTM as argmax decoding. This demonstrates that temperature annealing sampling strategy is capable of generating more diverse sequences that, while not similar to the ground truth, still meet the given structural conditions.

---

**Algorithm 1** Temperature-annealed stochastic sampling

---

**Input:** trained network $f_\theta(\cdot)$, the total sampling steps $T$, the minimum temperature $\tau_{\min}$ and the maximum temperature $\tau_{\max}$.
**Output:** generated sample $\mathbf{x}^{(0)}$.
**for** $n = 1, 2, \ldots, N$ **do**
    Initialize $\mathbf{x}_{T,n} \sim q_{\text{noise}}$;
    Initialize $b_{T,n} = 0$;
**end for**
**for** $t = T, \ldots, 1$ **do**
    # Determine the temperature $\tau$ of the current timestep $t$.
    $\tau = \tau_{\min} + \frac{t-1}{T-1}(\tau_{\max} - \tau_{\min})$
    **for** $n = 1, 2, \ldots, N$ **do**
        Draw $\widetilde{\mathbf{x}}_{0,n} \sim \text{Categorical}\left(f_\theta(\mathbf{x}_{t,n})/\tau\right)$;
        Generate $\mathbf{v}_{t-1,n}$ according to $\log p(\widetilde{\mathbf{x}}_{0,n})$
        **if** $b_{t,n} = 1$ **then**
            Draw $\mathbf{u}_{t,n}^{(1)} \sim q_{\text{noise}}$;
            $\mathbf{x}_{t-1,n} = v_{t-1,n}^{(1)}\mathbf{x}_{t,n} + \left(1 - v_{t-1,n}^{(1)}\right)\mathbf{u}_{t,n}^{(1)}$;
        **else**
            Draw $\mathbf{u}_{t,n}^{(2)} \sim q_{\text{noise}}(\mathbf{x}_{t,n})$;
            $\mathbf{x}_{t-1,n} = v_{t-1,n}^{(2)}\widetilde{\mathbf{x}}_{0,n} + \left(1 - v_{t-1,n}^{(2)}\right)\mathbf{x}_{t,n}^{(2)}$;
        **end if**
        Let $b_{t-1,n} = b_{t,n} \wedge v_{t-1,n}^{(1)} \vee v_{t-1,n}^{(2)}$;
    **end for**
**end for**
**Return** $\mathbf{x}_{0,1:N}$.

---

# D ANALYSIS ON THE EVALUATION METRICS IN THE UNCONDITIONAL GENERATION

In Tab. 2, we observe that DPLM-2 achieves a high scTM score while the scRMSD score is a bit higher than other baselines, e.g., MultiFlow. We will make a detailed discussion of this.

We first highlight that the generated samples from DPLM-2 share similar scTM (0.925) and scRMSD (3.9) as native PDB samples, which also exhibit good scTM (0.904) with a little bit higher scRMSD (4.623). Moreover, Additionally, DPLM-2 maintains a balanced structural composition (helix: 0.4, strand: 0.2, coil: 0.45), closely resembling natural distributions. In contrast, for MultiFlow, the officially released model with distillation attains much lower scRMSD (3.2), while the performance of our retrained version (on the same DPLM-2 training set) degrades in both scTM (0.871) and scRMSD (6.58). Lower scRMSD in MultiFlow with distillation, appears to be driven by overrepresentation of structured elements (Fig. 4A), i.e., significantly biasing towards proteins with more helices, with less strands and loops (Fig. 4C). This overrepresentation drives the observed scRMSD improvement but deviates from natural protein diversity.

Then we delve into the insight and purpose of the TM-score and RMSD metrics. TM-score emphasizes global topology, while RMSD is sensitive to local structural errors. As such, although scTM and scRMSD are generally correlated, discrepancies can arise. The purpose of TM-score is to solve this

Table 10: Analysis on the performance degradation in the representation learning task, including HumanPPI, MetalIonBinding and DeepLoc Subcellular.

| Exp id | Model | Training set | HumanPPI Acc(%) | MetalIonBinding Acc(%) | DeepLoc Subcellular Acc(%) |
|---|---|---|---|---|---|
| 0 | SaProt | AFDB data (40M) | 86.41 | **75.75** | **85.57** |
| 1 | DPLM | PDB + swissprot (200K) | 73.33 | 62.25 | 63.49 |
| 2 | DPLM-2 | PDB + swissprot (200K) | 77.22 | 69.47 | 66.77 |
| 3 | DPLM w/ fully pretraining on UniRef50 | UniRef50 (45M) | 86.41 | *75.15* | *84.56* |
| 4 | DPLM-2 w/ finetuning from DPLM | PDB + swissprot (200K) | 84.44 | 74.28 | 82.98 |
| 5 | DPLM-2 w/ finetuning from DPLM | AFDB_reps + PDB + swissprot (1.5M) | **87.78** | - | 83.42 |

sensitivity of RMSD, because RMSD is an average distance of all residue pairs in two structures, a local error (e.g. a misorientation of the tail) will raise a big RMSD value although the global topology is correct. In TM-score, however, the small distance is weighted stronger than the big distance, which makes the score insensitive to the local modeling error. As shown in Fig. 4B, some samples from DPLM-2 with higher loop proportion are more conformationally flexible, hence may show high scTM ($>0.9$) but worse scRMSD ($>2.0$), similar to natural protein. However, this does not necessarily indicate a limitation in generation quality but reflects differences in metric sensitivity.

As a result, the in-silico designability of protein generation should be evaluated comprehensively using both scTM and scRMSD, as each metric offers distinct insights and serves different purposes. For users aiming to generate samples with accurate global topology, scTM serves as a reliable indicator, whereas scRMSD may occasionally exclude reasonable structures. Conversely, for applications requiring structurally rigid and stable proteins, such as functional designs (e.g., binder design), scRMSD has been shown to correlate more strongly with in vitro success rates, as suggested by RFDiffusion.

## E  ANALYSIS ON THE PERFORMANCE DEGRADATION IN REPRESENTATION LEARNING

In Tab. 5, we find DPLM-2 demonstrates a performance degradation compared with the DPLM, which is used for parameter initialization for DPLM-2, in some tasks (e.g., DeepLoc Subcellular). We hypothesize two potential causes for the observed degradation: (1) DPLM-2 needs to accommodate additional structural representations given the same model capacity (parameters), which could negatively impact the representation learning performance. (2) As continuous training on smaller magnitude of structure data, DPLM-2 may experience catastrophic forgetting of the representation power gained during DPLM's large-scale sequence pretraining.

To explore (1), we eliminated pretraining factors by retraining both DPLM and DPLM-2 with random initialization on the SwissProt and PDB datasets for 100K training steps. Additionally, we evaluated performance across all three tasks (HumanPPI, MetalIonBinding and DeepLoc Subcellular) where DPLM-2 underperformed compared to DPLM. As shown in the Tab. 10, when large-scale sequence pretraining is removed, DPLM-2 significantly outperforms DPLM (exp 2 vs exp 1). This indicates that incorporating structural information enhances performance rather than harming it, which rejects the hypothesis (1).

However, when DPLM undergoes large-scale pretraining and DPLM-2 is subsequently trained from the pretrained DPLM, the performance of DPLM-2 on certain tasks diminishes (exp 4 vs exp 3). Given the relatively smaller structure data for DPLM-2 training, this suggests that catastrophic forgetting occurs during DPLM-2's multimodal training, reducing the advantages of large-scale pretraining. To verify and mitigate this, we curate additional 1.3M predicted structures from AFDB representative (Barrio-Hernandez et al., 2023), and trained DPLM-2 on this larger data. The experimental results show that the amount of structure data is indeed a key factor for better multimodal protein representations, leading to significantly improved performance over the original data (exp 5 vs exp 4). In particular, on HumanPPI, enlarging data from 200K to 1.5M helps DPLM-2 attain 2.3% improvement, and also outperforms SaProt, a strong multimodal PLM trained with 40M Foldseek tokenized AFDB data.

## F  MORE EMPIRICAL RESULTS

### F.1  COMPREHENSIVE EVALUATION ON THE UNCONDITIONAL SEQUENCE GENERATION

In addition to the Tab. 2, we have conducted more comprehensive evaluations on the unconditional generation in terms of protein sequence, including: (1) **sequence and structural diversity**: we conduct MMseqs2 clustering and Foldseek structural clustering at different thresholds. For MMseqs2

Table 11: Comprehensive analysis on the protein sequence generation. We evaluate the performance in terms of pLDDT, sequence and structural diversity, sequence naturalness and sequence novelty.

| Evaluation Metric | MultiFlow (official w/ distillation) | MultiFlow (retrained on DPLM-2 data) | DPLM | DPLM-2 |
|---|---|---|---|---|
| structural plausibility ($\uparrow$) | | | | |
| pLDDT | 79.4 | 62.6 | 84.0 | 83.7 |
| sequence diversity ($\uparrow$) | | | | |
| MMseqs2 cluster at **seq-id=0.3** & plddt > 70 | 0.804 | 0.204 | 0.740 | 0.745 |
| MMseqs2 cluster at **seq-id=0.5** & plddt > 70 | 0.860 | 0.294 | 0.745 | 0.755 |
| MMseqs2 cluster at **seq-id=0.7** & plddt > 70 | 0.862 | 0.294 | 0.815 | 0.795 |
| MMseqs2 cluster at **seq-id=0.9** & plddt > 70 | 0.862 | 0.294 | 0.885 | 0.895 |
| structural diversity ($\uparrow$) | | | | |
| Foldseek at **TMscore=0.3** & scTM > 0.5 | 0.030 | 0.080 | – | 0.198 |
| Foldseek at **TMscore=0.5** & scTM > 0.5 | 0.500 | 0.440 | – | 0.545 |
| Foldseek at **TMscore=0.7** & scTM > 0.5 | 0.962 | 0.830 | – | 0.646 |
| Foldseek at **TMscore=0.9** & scTM > 0.5 | 0.990 | 0.910 | – | 0.746 |
| sequence naturalness ($\downarrow$) | | | | |
| ProGen2 ppl | $8.11 \pm 2.08$ | $9.15 \pm 2.77$ | $4.33 \pm 2.51$ | $4.08 \pm 2.00$ |
| sequence novelty ($\downarrow$) | | | | |
| MMseq2 search against PDB+swissprot | 0.306 | 0.312 | 0.304 | 0.475 |

Table 12: Unconditional generation from the empirical length distribution.

| Length | scTM | scRMSD | pLDDT |
|---|---|---|---|
| Length interval: $[100, 200, ..., 500]$ | $0.925 \pm 0.085$ | $3.899 \pm 3.723$ | 82.686 |
| Training set (PDB+Swissprot) length dist. | $0.929 \pm 0.086$ | $3.967 \pm 3.257$ | 83.698 |

clustering, we cluster samples with pLDDT > 70, while for foldseek clustering we cluster samples with scTM > 0.5. This quality threshold for diversity is inspired by MultiFlow, which is more informative by avoiding diverse but messy sequences. Then, we divide the number of clusters by the total number of samples to measure the diversity. (2) **sequence naturalness**: we calculate perplexity as a measure of naturalness with ProGen2-large (Nijkamp et al., 2022). (3) **sequence novelty**: we calculate novelty through sequence identity to the nearest neighbor in the training set.

All models generate 100 samples per length in the range of 100, 200, 300, 400 and 500 for evaluation, with the results demonstrated in the Tab. 11. One particularly insightful observation is the distinct behavior of MultiFlow (w/ distillation) and DPLM-2 regarding structural diversity. Specifically, DPLM-2 exhibits greater diversity under strict TM-score thresholds ($\leq 0.5$), while MultiFlow achieves better diversity at higher TM-score thresholds ($\geq 0.7$). Combined with the average inner-TM scores (DPLM-2: 0.275, MultiFlow: 0.356) , this suggests that DPLM-2 excels at generating diverse structures in terms of global topologies but exhibits limited structural variation within each cluster. This finding highlights a key limitation of the current structural tokenization approach: the loss of fine-grained structural variations, emphasizing the need for future improvements in this area. Additionally, DPLM-2 achieves the lowest ProGen2 perplexity, while its sequence identity to training data (0.475) is higher than that of DPLM and MultiFlow. This indicates that the sequences generated by DPLM-2 align more closely with the natural distribution.

## F.2 UNCONDITIONAL GENERATION FROM THE EMPIRICAL LENGTH DISTRIBUTION

In our paper, we follow the setting in the MultiFlow and sample within length intervals in the unconditional generation, ensuring fair comparisons with previous models under the similar settings to better assess the strengths and limitations of our models.

Meanwhile, DPLM-2 is capable of generating proteins from the empirical length distribution. Specifically, we sample 2048 sequences with length sampled from the length distribution of PDB and SwissProt datasets. Tab. 12 demonstrates that DPLM-2 can generate highly plausible proteins from the empirical length distribution, which is consistent with sampling with length intervals.

## F.3 REPRESENTATION LEARNING EVALUATION WITH MORE BASELINES

We have added more recent strong baselines, such as GNN-based methods (e.g., GearNet), in addition to Tab. 5 to make a more comprehensive comparison on the representation learning tasks, as shown in Tab. 13. This demonstrates that DPLM-2 is capable of utilizing both protein structure and sequence to generate more informative representations for series of downstream tasks.

Table 13: Representation learning performance on various protein predictive tasks, comparing between DPLM-2 and more recent strong baselines. † means results are quoted from SaProt paper, while ∗ means results are quoted from their respective paper.

| Models | Thermostability | HumanPPI | Metal Ion Binding | EC | GO | | | DeepLoc | |
| | | | | | MF | BP | CC | Subcellular | Binary |
| --- | --- | --- | --- | --- | --- | --- | --- | --- | --- |
| | Spearman's $\rho$ | Acc (%) | Acc (%) | Fmax | Fmax | Fmax | Fmax | Acc (%) | Acc (%) |
| †SaProt (650M) | **0.724** | **86.41** | **75.75** | 0.882 | *0.682* | 0.486 | 0.479 | **85.57** | *93.55* |
| †SaProt-GearNet (650M) | 0.660 | 85.80 | 74.44 | 0.889 | 0.678 | *0.522* | 0.508 | 84.16 | 93.63 |
| †MIF-ST (Yang et al., 2022b) | 0.694 | 75.54 | 75.08 | 0.807 | 0.633 | 0.375 | 0.322 | 78.96 | 91.76 |
| †GearNet (Zhang et al., 2023) | 0.571 | 73.86 | 71.26 | 0.874 | 0.644 | 0.481 | 0.476 | 69.45 | 89.18 |
| ∗GearNet updated (Zhang et al., 2023) | – | – | – | 0.890 | 0.681 | 0.488 | 0.464 | – | – |
| ∗CoupleNet [1] | – | – | – | 0.866 | 0.669 | 0.467 | 0.494 | – | – |
| ∗CDConv [2] | – | – | – | 0.820 | 0.654 | 0.453 | 0.479 | – | – |
| ∗ESM2-650M-S [3] | – | – | – | 0.823 | 0.649 | 0.463 | *0.519* | – | – |
| ∗VABS-NET [4] | – | – | – | **0.900** | **0.695** | **0.531** | **0.579** | – | – |
| ∗ESM-GearNet-INR-MC [5] | – | – | – | *0.896* | *0.683* | 0.518 | 0.504 | – | – |
| ESM2 (650M) | 0.691 | *84.78* | 71.88 | 0.868 | 0.670 | 0.473 | 0.470 | 83.68 | 92.28 |
| DPLM (650M) | 0.695 | **86.41** | *75.15* | 0.875 | 0.680 | 0.480 | 0.478 | *84.56* | 93.09 |
| DPLM-2 (650M) | *0.714* | 84.44 | 74.28 | 0.881 | 0.682 | 0.493 | 0.481 | 82.98 | **93.64** |

## F.4 INVERSE FOLDING EVALUATION WITH MORE BASELINES

For inverse folding task, we mainly focus on the comparison with other multimodal generative models (MultiFlow, ESM3) in the Tab. 4. We have also added more recognized baseline methods in inverse folding evaluation, as shown in Tab. 14.

We conduct experiments on CATH 4.2 testset. We observe that DPLM-2 is able to achieve close results with the strong baselines despite slightly lower scTM. To further improve scTM to bridge the last gap, there are several potential directions: (1) inverse folding SFT training: DPLM-2 conducts this task in a zero-shot manner while other systems are purpose-built models, thus task-oriented SFT training could help as we have observed in folding; (2) better structure modeling includes introducing separate structure encoders for structure encoding and generation purposes, or hybrid tokenization for recovering the lost fine-grain structural variations, as discussed in the §B.4.

Table 14: Inverse folding performance comparison between DPLM-2 and other baselines on the CATH 4.2 testset. † means results are quoted from their respective paper.

| Model | AAR | scTM |
| --- | --- | --- |
| †Knowledge-Design (Gao et al., 2023) | 60.77 | – |
| †GraDe-IF (Yi et al., 2023) | 52.21 | – |
| †MMDesign (Zheng & Li, 2024) | 54.88 | – |
| †VFN-IFE (Mao et al., 2023) | 62.67 | – |
| PiFold (Gao et al., 2022) | 51.66 | – |
| †Bridge-IF (Zhu et al., 2024) | 58.59 | – |
| ProteinMPNN (Dauparas et al., 2022) | 45.96 | 0.87 |
| LM-Design (Zheng et al., 2023b) | 54.41 | 0.88 |
| DPLM-2 w/ argmax decoding | 42.70 | 0.84 |
| DPLM-2 w/ temperature-annealed sampling | 36.30 | 0.84 |

## F.5 MOTIF SCAFFOLDING

**Evaluation Pipeline.** We evaluate DPLM-2 in sequence-based, structure-based and co-generation ways. The overall illustration is shown in Fig. 8. We focus on the two aspects: overall quality and motif part consistency. The assessment of overall quality varies across different approaches. Specifically, (1) For sequence-based method, we only take the generated sequence and utilize ESMFold to obtain the predicted structure, and the pLDDT score provided by ESMFold is used to assess overall quality. (2) For structure-based method, we only take the generated structure, and then leverage ProteinMPNN to predict the sequence, followed by ESMFold to predict the structure, where overall quality is assessed by scTM. (3) For co-generation method, we take both the generated structure and sequence, and predict structure given generated sequence with ESMFold, where scTM is calculated between generated structure and ESMFold predicted structure to evaluate overall quality. Considering that the ground truth motif structure is given, we only utilize the ESMFold predicted structure to calculate motif-RMSD.

**Result of Each Problem.** Tab. 15 presents the result of each motif-scaffolding problem. DPLM-2 achieves the best average success rate in each evaluation. Compared with ESM3, DPLM-2 shows better results in 12 problems in co-generation evaluation and 10 problems in sequence-based evaluation. Meanwhile, DPLM-2 outperforms RFDiffusion in 14 problems in structure-based evaluation. This demonstrates that DPLM-2 can achieve strong performance under various evaluation methods. We also find that taking the best result from 8 samples can bring significant improvement compared to 1 sample, especially in terms of success rate. In the co-generation evaluation, DPLM2 with sampling 8 times improves the success rate of most of the problems by a large margin. We hypothesize that sampling eight times largely alleviates errors caused by randomness in the sampling process, thereby producing a more suitable scaffold for the given motif.

Figure 8: Sequence-based, structure-based and co-generation evaluation pipeline of motif-scaffolding.

| | sequence-based | |
|---|---|---|
| prediction | seq_pred: ✓ | struct_pred: ✗ |
| motif-preserving | RMSD(ESMFold(seq_pred)[motif],struct_native[motif])<1.0 | |
| designability | pLDDT(ESMFold(seq_pred))>70 | |
| | **structure-based** | |
| prediction | seq_pred: ✗ | struct_pred: ✓ |
| motif-preserving | RMSD(ESMFold(PMPNN(struct_pred))[motif],struct_native[motif])<1.0 | |
| designability | TMScore(ESMFold(PMPNN(struct_pred)), struct_pred)>0.8 | |
| | **co-generation** | |
| prediction | seq_pred: ✓ | struct_pred: ✓ |
| motif-preserving | RMSD(ESMFold(seq_pred)[motif],struct_native[motif])<1.0 | |
| designability | TMScore(ESMFold(seq_pred), struct_pred)>0.8 | |

Table 15: Motif-scaffolding results of each problem. * means best result from 8 samples.

| | sequence-based | | | | structure-based | | co-generation | | |
|---|---|---|---|---|---|---|---|---|---|
| | EvoDiff | DPLM | ESM3 | DPLM2 | *RFDiffusion | *DPLM2 | ESM3 | DPLM2 | *DPLM2 |
| 1BCF | 0.00 | 0.00 | **0.89** | 0.01 | **1.00** | 0.07 | **0.23** | 0.01 | 0.05 |
| 1PRW | 0.61 | 0.83 | **0.96** | 0.86 | 0.08 | **0.96** | 0.54 | 0.84 | **0.95** |
| 1QJG | 0.00 | 0.00 | 0.02 | **0.03** | 0.00 | 0.00 | 0.03 | 0.02 | **0.05** |
| 1YCR | 0.02 | 0.38 | 0.41 | **0.77** | 0.74 | **0.93** | 0.18 | 0.53 | **0.98** |
| 2KL8 | 0.04 | 0.08 | 0.11 | **0.47** | 0.88 | **0.94** | 0.11 | 0.57 | **1.00** |
| 3IXT | 0.06 | 0.17 | 0.18 | **0.67** | 0.25 | **0.77** | 0.02 | 0.41 | **0.73** |
| 4JHW | 0.00 | 0.00 | 0.00 | 0.00 | 0.00 | 0.00 | 0.00 | 0.00 | 0.00 |
| 4ZYP | 0.00 | 0.00 | 0.03 | **0.16** | 0.40 | **0.51** | 0.08 | 0.10 | **0.64** |
| 5IUS | 0.00 | 0.00 | 0.00 | 0.00 | **0.02** | 0.00 | 0.00 | 0.00 | 0.00 |
| 5TPN | 0.00 | 0.00 | **0.03** | 0.00 | **0.61** | 0.06 | **0.01** | 0.00 | 0.00 |
| 5TRV_long | 0.00 | 0.00 | **0.19** | 0.00 | **0.37** | 0.08 | **0.19** | 0.00 | 0.07 |
| 5TRV_med | 0.00 | 0.00 | **0.16** | 0.03 | **0.24** | 0.07 | 0.16 | 0.02 | **0.19** |
| 5TRV_short | 0.00 | 0.00 | 0.01 | **0.07** | 0.04 | **0.10** | 0.01 | 0.03 | **0.11** |
| 5WN9 | 0.00 | 0.00 | **0.02** | 0.00 | 0.00 | **0.20** | 0.00 | 0.00 | 0.00 |
| 5YUI | 0.00 | 0.00 | 0.00 | 0.00 | **0.02** | 0.00 | 0.00 | 0.00 | 0.00 |
| 6E6R_long | 0.01 | 0.65 | 0.07 | **0.91** | 0.86 | **0.92** | 0.04 | 0.78 | **1.00** |
| 6E6R_med | 0.03 | **0.94** | 0.24 | 0.93 | **0.89** | 0.88 | 0.14 | 0.77 | **0.97** |
| 6E6R_short | 0.07 | **0.87** | 0.09 | 0.86 | 0.39 | **0.78** | 0.06 | 0.64 | **0.99** |
| 6EXZ_long | 0.00 | 0.01 | 0.32 | **0.61** | **0.76** | 0.63 | 0.13 | 0.44 | **0.95** |
| 6EXZ_med | 0.00 | 0.00 | 0.31 | **0.66** | 0.49 | **0.63** | 0.31 | 0.55 | **0.96** |
| 6EXZ_short | 0.00 | 0.00 | 0.31 | **0.66** | 0.39 | **0.41** | 0.28 | 0.58 | **0.87** |
| 7MRX_long | 0.00 | 0.02 | **0.36** | 0.23 | 0.09 | **0.32** | 0.37 | 0.20 | **0.73** |
| 7MRX_med | 0.00 | 0.31 | **0.65** | 0.28 | 0.11 | **0.31** | 0.59 | 0.22 | **0.70** |
| 7MRX_short | 0.00 | 0.34 | **0.68** | 0.26 | 0.02 | **0.41** | 0.74 | 0.24 | **0.88** |
| pass rate | 7/24 | 11/24 | **21/24** | 18/24 | 20/24 | 20/24 | **20/24** | 18/24 | 19/24 |
| avg. success rate | 0.04 | 0.19 | 0.25 | **0.35** | 0.40 | **0.42** | 0.18 | 0.29 | **0.53** |

## G DISCUSSION ON THE CONDITIONAL INDEPENDENCE ASSUMPTION

In the Eq. 2, we make a conditional independence assumption between the protein structure and sequence. However, conditional independence is not a special assumption made by DPLM-2, it is a fundamental assumption made by diffusion models in general and their multimodal extensions, derived from the nature of their forward and backward processes. Previous theoretical studies on diffusion models have shown the convergence between generated samples distribution and data distribution is guaranteed under such conditional independence. In this paper, we have empirical evidence showing the consistency/compatibility between co-generated structures and sequences (e.g., scTM for co-generation), and we believe a mathematical proof of this is beyond the scope of this paper and can refer to the established theoretical results on diffusion. Nevertheless, we do love to elaborate on our thoughts and understanding of this as follows.

**Conditional independence in diffusion models in general.** Conditional independence over the elements of high-dimensional data, i.e., $p_\theta(\mathbf{x}_{t-1}|\mathbf{x}_t) = \prod_{i=1}^{d} p_\theta(x_{t-1,[i]}|\mathbf{x}_t)$, is a prevailing assumption in diffusion probablistic models, both continuous and discrete variants, thanks to their iterative nature of probabilistic modeling. For example, in continuous diffusion models for vision generation, the denoising networks learn to reconstruct a denoised image at each timestep $t-1$ by simultaneously and independently operating over all pixels conditioned on the previous noisier pixels of the image

and the current timestep (or equivalently noise level) $t$. So as for discrete diffusion, where discrete diffusion for text or protein sequence treats tokens of a sequence of $\mathbf{x_{t-1}}$ independently given $\mathbf{x_t}$. Several recent works have established the theoretical foundations on the convergence analysis of both continuous diffusion (Chen et al., 2023) and discrete diffusion (Chen & Ying, 2024; Zhang et al., 2024), showing that there are theoretical guarantees of the convergence of the generated sample distribution of the diffusion models and the data distribution, which means that a well-learned diffusion models can preserve the statistical structure of the data (in other words, the consistency between the elements $\mathbf{x} = \{x_1, ..., x_d\}$ of the generated samples).

**Conditional independence in multimodal diffusion models.** Multimodal diffusion models aim to accommodate two or more modalities using a unified models. In this case, conditional independence between modalities is generally made $p_\theta(\mathbf{x}_{t-1}, \mathbf{y}_{t-1}|\mathbf{x}_t, \mathbf{y}_t) = \prod_i p_\theta(x_{t-1,[i]}|\mathbf{x}_t, \mathbf{y}_{t-1}) \prod_j p_\theta(y_{t-1,[j]}|\mathbf{x}_t, \mathbf{y}_t)$. For instance, UniDiffuser (Bao et al., 2023) is a multimodal continuous diffusion model that handles text and image modalities independently at each timestep, conditioned on the predictions from the previous timestep. Multiflow (Campbell et al., 2024), on the other hand, factorizes protein data into three modalities—translation, orientation, and amino acid type—assuming conditional independence. It establishes a multimodal diffusion/flow-based model by combining three types of stochastic processes over Euclidean, SO(3), and categorical spaces for these modalities. In DPLM-2, we adopt a unified discrete diffusion approach where structure tokens and amino acid tokens are treated as conditionally independent. While theoretical guarantees for the convergence of mixture diffusion processes are still under-explored, existing discrete diffusion theory (Chen & Ying, 2024) ensures that a well-trained DPLM-2 can converge to the tokenized structure-sequence data distribution, supporting consistency between structure and sequence tokens.

Additionally, theoretical studies on non-autoregressive Transformers for text generation, which are akin to masked discrete diffusion, indicate that the learning difficulty of such models can be evaluated through conditional total correlation. This dataset-dependent measure captures the discrepancy between a joint data distribution and a fully factorized distribution under conditional independence (Huang et al., 2022). These studies suggest that simplifying the target data, for instance, by using tokenized structure instead of 3D coordinates, reduces conditional total correlation, thereby enhancing both learning and generation quality.

Given the consistency of structure tokens and amino acid can be ensured to learn in DPLM-2 by previous theoretical results, the overall structure and sequence consistency can be achieved with a decent structure tokenizer, such as the one proposed in this paper, which accurately maps structure tokens to their atomic coordinates.

## H RELATED WORK

### H.1 PROTEIN LANGUAGE MODELS

There is growing interest in developing protein LMs at the scale of evolution, such as the series of ESM (Rives et al., 2019; Lin et al., 2022), TAPE (Rao et al., 2019), ProtTrans (Elnaggar et al., 2021), PRoBERTa (Nambiar et al., 2020), PMLM (He et al., 2021), ProteinLM (Xiao et al., 2021), PLUS (Min et al., 2021), Adversarial Masked LMs (McDermott et al., 2021), ProteinBERT (Brandes et al., 2022), CARP (Yang et al., 2022a) in masked language modeling (MLM) paradigm, Prot-GPT2 (Ferruz et al., 2022) in causal language modeling paradigm, and several others (Melnyk et al., 2022a; Madani et al., 2021; Unsal et al., 2022; Nourani et al., 2021; Lu et al., 2020; Sturmfels et al., 2020; Strodthoff et al., 2020). These protein language models exhibit remarkable generalization ability on various downstream tasks and be able to capture evolutionary information about secondary and tertiary structures from sequences alone. Meanwhile, recent study shows these models' potency in revealing protein structures (Lin et al., 2022), predicting the effect of sequence variation on function (Meier et al., 2021), antibody infilling (Melnyk et al., 2022a) and many other general purposes (Rives et al., 2019). Simultaneously, Verkuil et al. (2022) demonstrate that the large scale protein LMs can generate *de novo* proteins by generalizing beyond natural proteins, both theoretically and experimentally validating their hypothesis in exhaustive detail, in which protein LMs demonstrate competency in designing protein structure despite being exclusively trained on sequences.

### H.2 PROTEIN STRUCTURE GENERATIVE MODELS

Diffusion models have become popular tools in structural biology for protein generation, and their utility has been demonstrated across a range of generative tasks in recent years. Trippe et al. (2022), along with others, have introduced several diffusion model variants, each with its unique

approach. For instance, while some models focus on generating the protein backbone by diffusing over protein coordinates, others, such as those proposed by Wu et al. (2022b), target inter-residue angles. Lin & AlQuraishi (2023) and Yim et al. (2023) have developed models that handle both the position and orientation of residue frames. RFDiffusion (Watson et al., 2023) is a model that assists in designing protein structures for specific functions, such as enzymes. It is versatile in protein design and has been used to create therapeutic proteins, with some designs being confirmed in the laboratory. ProteinSGM (Lee et al., 2022) is a model that uses 2D matrices, which represent the distances and angles between protein parts, to create 3D protein structures for novel protein designs. FoldingDiff (Wu et al., 2022a) is a model that generates protein sequences expected to fold into a specific structure. These sequences are verified with prediction tools, although they have not been experimentally confirmed yet. Chroma (Ingraham et al., 2023) is a model designed for creating large proteins and protein complexes, considering various constraints like distances and symmetry. It transforms a collapsed polymer into protein backbone and sequence more quickly than older methods, thereby allowing for the efficient generation of large structures. Multiflow (Campbell et al., 2024) develop mulitmodal flow matching for protein structure-sequence co-generation (Jin et al., 2021; Shi et al., 2022). ProtPardelle (Chu et al., 2024) propose an all-atom generative approach for co-design.

