# OpenReview forum: "DPLM-2: A Multimodal Diffusion Protein Language Model"
_ICLR.cc/2025/Conference — ICLR 2025 Poster_

### Official Review · Reviewer_FqCq · 2024-10-27

**Soundness:** 3
**Presentation:** 4
**Contribution:** 3
**Rating:** 8
**Confidence:** 5

**Summary:**

The paper presents a multimodal diffusion protein language model that integrates information from both sequence and structure modalities. The model is built upon a pre-trained DPLM model, using LoRA technology to extend the original sequence-only model’s capability to process structural knowledge. This approach not only reuses the knowledge embedded in the existing sequence-based protein language model but also reduces training costs and expenses. Compared to traditional autoregressive language models, the diffusion language model offers more flexible generation capabilities, making it better suited for modeling protein data with extensive non-unidirectional semantic dependencies. Overall, this work represents a natural extension and generalization of the DPLM model, exploring the performance of such diffusion language models on multimodal protein data and demonstrating the potential advantages of multimodal protein language models.

**Strengths:**

1. This is the first multimodal diffusion protein language model, which effectively explores the application of such models in multimodal scenarios. Considering the potential of diffusion language models, I believe this is a highly meaningful research question.

2. The paper has a clear logical structure, and its content is straightforward and easy to understand.

3. The paper demonstrates the performance of DPLM-2 across a variety of tasks, providing substantial and comprehensive content that analyzes the model’s performance from multiple perspectives.

**Weaknesses:**

1. The main results reported in this paper are based on unconditional generation tasks; however, some evaluation metrics may have inherent limitations.

2. The model in this paper is trained on a relatively small protein structure dataset, lacking exploration on larger-scale structural data. I believe that extending multimodal protein language models to larger protein sequence and structure datasets could further unlock the potential of this approach.

3. There are some minor spelling and formatting errors: the equation on line 197 is missing a right parenthesis, and $ b_{i}(t) $ introduced in equation (1) (on line 192) is undefined in the paper.

**Questions:**

1. **Regarding structure tokens:** Figure 2B shows that most structure tokens are concentrated in alpha helix and beta sheet regions, with lower density in loop regions. However, in line 409 of the paper, you state that "loops are highly flexible." Could this pose a potential issue? More variable regions typically contain richer and more diverse information, which may require more structure tokens to model this diversity effectively. Yet, your experimental results do not align with this. Moreover, your results in Figure 4B also indicate that the model’s designability is notably worse in regions with a higher proportion of loops. Could one possible reason for this issue be that the structure tokens in loop regions are not well-learned, resulting in weaker modeling capability?

2. **Regarding the pLDDT metric:** Table 2 shows that MultiFlow (retrained on our training data) and DPLM-2 (co-generation) perform similarly on scTM and scRMSD metrics but diverge significantly on the pLDDT metric. A similar trend is observed when comparing Figures 3A and 3C. In Figure 3A, scTM and scRMSD metrics remain relatively stable as sequence length increases, with scRMSD even increasing for sequences of length 500. However, Figure 3C shows a notable upward trend in pLDDT as sequence length increases (from 100 to 300), without a significant drop in pLDDT even for sequences of length 500. These phenomena indicate that the scTM and scRMSD metrics convey somewhat contradictory information to the pLDDT metric. Generally, a significant drop in pLDDT should correspond with a decrease in scTM and an increase in scRMSD. My question here is whether this inconsistency might indicate that the pLDDT metric is unreliable. I raise this because, in previous experiments, we observed that pLDDT, being predicted by a neural network (e.g., ESM-2) trained on natural protein datasets, sometimes fails to handle model-generated proteins that deviate significantly or exhibit severe irregularities, leading to inflated pLDDT values. This phenomenon is quite common in unconditional generation. Therefore, I believe using a more diverse set of metrics is essential. However, you did not provide evaluation results for MultiFlow (retrained on our training data) on the Novelty and Diversity dimensions, which makes it harder to assess the model’s performance on this part of the task.

3. **Regarding scTM and scRMSD metrics:** In Table 2, under the unconditional backbone generation task, DPLM-2 achieves the highest scTM metric while also showing a significantly worse scRMSD metric, both in terms of mean and variance, which are at high levels. Is this phenomenon somewhat unusual? Given that these two metrics should have a certain level of correlation.

4. **Regarding the inverse folding task:** Comparing the performance of ESM3 and DPLM-2 in Table 5 reveals a pattern where DPLM-2 often exhibits a higher AAR while frequently showing a lower scTM. Could this possibly be due to biases in the protein structure prediction model (ESMFold) or because the sequences generated by DPLM-2, despite having higher ARR, may deviate further from the target protein’s data distribution?

5. **Regarding repetition:** In practice, we observed that diffusion protein language models tend to generate a large number of repetitive amino acids. I noticed that DPLM employed a resampling scheme to address this issue (code link: [here](https://github.com/bytedance/dplm/blob/5545c6d4166f515b4eb66ada41d0ab3178dfe6ca/src/byprot/models/lm/dplm.py#L279)). Therefore, regarding the DPLM-2 model, I would like to know whether it also encounters similar repetition issues. If so, what approach did you adopt to resolve it? If not, I would like to know what prevented this issue in DPLM-2.

6. **Regarding VQ-VAE:** I am curious about the vocabulary usage rate of VQ-VAE when the vocabulary size is 1024. I noticed a significant performance difference between VQ-VAE and LFQ in this case, so I wonder if the vocabulary usage rate might be a contributing factor to this issue.

7. **Regarding catastrophic forgetting:** I would like to confirm whether DPLM-2 in Table 7 still utilizes structural pre-training, given that neither of these models used large-scale sequence pre-training. If structural pre-training was indeed applied to DPLM-2, then there are two differences between DPLM and DPLM-2: one is the difference in model architecture, and the other is that DPLM-2 was pre-trained on structural data. With these two variables present, how can we determine that the performance decline in DPLM-2 is due to catastrophic forgetting?

8. **Regarding the structural information:** Similar to ESM-3, you also use discrete structure tokens to represent structural information. This approach offers scalability benefits for the model, yet it also limits the model’s ability to capture finer structural details accurately. This trade-off represents an important challenge in the current field of multimodal protein language models. I am interested to know whether you would consider any methods to mitigate this issue when applying DPLM-2 to more structure-related tasks, as you described in Section 5.

---

> ### Author Response · Authors · 2024-11-24
> **Rebuttal by Authors**
>
> Thank you so much for your support of our work and for so many insightful and thoughtful comments! We would like to address your concern and questions point-by-point below. Your insightful comments and suggestions have greatly improved our manuscript. Please have a check and we are happy to address any further feedback!

---

> ### Author Response · Authors · 2024-11-24
> **Rebuttal by Authors**
>
> > **`Q1`**: The model in this paper is trained on a relatively small protein structure dataset, lacking exploration on larger-scale structural data. I believe that extending multimodal protein language models to larger protein sequence and structure datasets could further unlock the potential of this approach.
>
> **`A1`**:
> Thanks for your suggestions! As we stated in the discussion section, we fully agree that the limited numbers of structure data may hinder DPLM-2 from unlocking its full potential. As you suggested, during the last week, we curated a 1.3 million predicted structure dataset from AlphaFoldDB, where predicted structures were representative entries obtained by MMSeq2 sequence clustering as well as Foldseek structure clustering. This data size is an order of magnitude larger than the current structure data we used, and not too much larger to help us perform an efficient proof-of-concept of whether enlarging data size can boost our model in a course of a week.
>
> The following table summarizes our primary explorations on how the size of the data, data mixing strategy as well as training strategies affect the model performance (DPLM-2 650M).
> | exp_id |                                                              | size            | scTM          | scRMSD          | Foldseek cluster |
> |--------|--------------------------------------------------------------|-----------------|---------------|-----------------|------------------|
> | 0      | PDB + swissprot (unclustered, original paper)                | 200K            | 0.925 ± 0.085 | 3.899 ± 3.723   | 54.5             |
> | 1      | PDB + swissprot (unclustered, original paper)  w/o selfmixup | 200K            | 0.916 ± 0.099 | 4.656 ± 6.366   | 44.0             |
> | 2      | PDB+swissprot (clustered with seq-identity=0.5)              | 110K clusters   | 0.883 ± 0.128 | 5.661 ± 6.532   | 52.8             |
> | 3      | AFDB_reps (clustered)                                        | 1.3M clusters   | 0.726 ± 0.237 | 22.557 ± 24.516 | 60.8             |
> | 4      | pretrained on exp 3 then finetuned using data as exp 2       | 1.3M  then 110K | 0.904 ± 0.101 | 5.411 ± 6.882   | 53.0             |
>
> Meanwhile, we find that training on the larger-scale structural data can further enhance the representation learning, please refer to **Q8** for more details.

---

> ### Author Response · Authors · 2024-11-24
> **Rebuttal by Authors**
>
> > **`Q2`**: Regarding scTM and scRMSD metrics: In Table 2, under the unconditional backbone generation task, DPLM-2 achieves the highest scTM metric while also showing a significantly worse scRMSD metric, both in terms of mean and variance, which are at high levels. Is this phenomenon somewhat unusual? Given that these two metrics should have a certain level of correlation.
>
>  **`A2`**: Thank you for pointing this out. We would like to address your concern as follows:
> The pLDDT metric reflects the structural plausibility of sequence. In addition to the pLDDT metric, we also include sequence-related metric that we calculate sequence diversity using mmseq2 clustering at sequence identy = 0.5 for good quality samples with pLDDT > 70. This quality threshold for diversity is inspired by Multiflow, which is more informative by avoiding diverse but messy sequences.  We follow the experimental setting in DiMA [1], generating 2048 sequences with length sampled from the length distribution of PDB + SwissProt. The results highlight that DPLM-2 is able to generate structurally plausible and diverse sequences for protein generation. We also find that training data distillation greatly helps Multiflow's sequence quality in terms of pLDDT and diversity.
>
> |                                            | scTM           | scRMSD        | helix ratio | strand ratio | coil ratio |
> |--------------------------------------------|----------------|---------------|-------------|--------------|------------|
> | native PDB samples                         | 0.904 ± 0.129  | 4.623 ± 5.688 | 0.36        | 0.22         | 0.42       |
> | Multiflow (w/ distilation)                 | 0.930 ± 0.098  | 3.208 ± 4.741 | 0.75        | 0.10         | 0.15       |
> | Multiflow (w/o distillation)               | 0.750 ± 0.163  | 9.306 ± 8.499 | 0.73        | 0.06         | 0.21       |
> | Multiflow (retrained on our training data) | 0.871 ± 0.934  | 6.580 ± 6.258 | 0.56        | 0.17         | 0.26       |
> | DPLM-2                                     | 0.925 ± 0.085  | 3.899 ± 3.723 | 0.47        | 0.16         | 0.37       |
>
> We first need to highlight that the generated samples from DPLM-2 share similar scTM (0.925) and scRMSD (3.9) as native PDB samples, which also exhibit good scTM (0.904) with a little bit higher scRMSD (4.623). Moreover, Additionally, DPLM-2 maintains a balanced structural composition (helix: 0.4, strand: 0.2, coil: 0.45), closely resembling natural distributions. In contrast, for MultiFlow, the officially released model with distillation attains much lower scRMSD (3.2), while the performance of our retrained version (on the same DPLM-2 training set) degrades in both scTM (0.871) and scRMSD (6.58). Lower scRMSD in MultiFlow with distillation, appears to be driven by overrepresentation of structured elements (Figure 4A), i.e., significantly biasing towards proteins with more helices, with less strands and loops (also see Figure 4C). This overrepresentation drives the observed scRMSD improvement but deviates from natural protein diversity.
>
> TM-score emphasizes global topology, while RMSD is sensitive to local structural errors. As such, although scTM and scRMSD are generally correlated, discrepancies can arise. The purpose of TM-score is to solve this sensitivity of RMSD, because RMSD is an average distance of all residue pairs in two structures, a local error (e.g. a misorientation of the tail) will raise a big RMSD value although the global topology is correct. In TM-score, however, the small distance is weighted stronger than the big distance, which makes the score insensitive to the local modeling error.
>
> As shown in Fig 4B, some samples from DPLM-2 with higher loop proportion are more conformationally flexible, hence may show high scTM (>0.9) but worse scRMSD (>2.0), similar to natural protein. However, this does not necessarily indicate a limitation in generation quality but reflects differences in metric sensitivity.
>
> As a result, the in-silico designability of protein generation should be evaluated comprehensively using both scTM and scRMSD, as each metric offers distinct insights and serves different purposes. For users aiming to generate samples with accurate global topology, scTM serves as a reliable indicator, whereas scRMSD may occasionally exclude reasonable structures. Conversely, for applications requiring structurally rigid and stable proteins, such as functional designs (e.g., binder design), scRMSD has been shown to correlate more strongly with in vitro success rates, as suggested by RFDiffusion.

---

> ### Author Response · Authors · 2024-11-24
> **Rebuttal by Authors**
>
> > **`Q3`**: Regarding structure tokens: Figure 2B shows that most structure tokens are concentrated in alpha helix and beta sheet regions, with lower density in loop regions. However, in line 409 of the paper, you state that "loops are highly flexible." Could this pose a potential issue? More variable regions typically contain richer and more diverse information, which may require more structure tokens to model this diversity effectively. Yet, your experimental results do not align with this. Moreover, your results in Figure 4B also indicate that the model’s designability is notably worse in regions with a higher proportion of loops. Could one possible reason for this issue be that the structure tokens in loop regions are not well-learned, resulting in weaker modeling capability?
>
> **`A3`**: Thanks for this great question! This leads to the discussion on how to explain different in silico designablity metrics (scTM & scRMSD) and what information the structure tokens learn. For the former, we have provided our thoughts on this in the previous question and we empirically notice that scTM for the samples indeed drops when proportion of loops become higher but still keep greater than 0.8 overall.
>
> **About what information the structure tokens learn.** For interpratability, we update a more informative simplex plot of struct token vs second struct in Fig. 2B. As you are pointing out, we can observe a strong correlation between a vast majority of the structure tokens and structured local environments, where a lot of structured tokens concentrate on the alpha helix and beta sheet vertices, while some tokens lie between regions or the loop vertice. There are also a subset of struct tokens having less clear clues to specific secondary structures. This suggests that structure tokens mostly capture clear secondary elements, some may correspond to structured local environments (in bewteen helic and sheet), while others could be high-level abstract entities or just not well-learned entries. On one hand, we agree that "more variable regions contain richer and diverse information" as these regions are of high entropy. On the other hand, the nature of lossy and clustering-like vector-quantization methods is highly likely to eliminate such high-frequency high-entropy structural variations, and only keep low-frequency, low-entropy content. We suggest that this could be the major reason for the not-well learned flexible regions. And this also exactly corresponds to your last question about the trade-offs and further directions of structure modeling in multimodal PLM. Please take a look at our elaborate discussion on this in **Q9**.

---

> ### Author Response · Authors · 2024-11-24
> **Rebuttal by Authors**
>
> > **`Q4`**: Regarding the pLDDT metric: Table 2 shows that MultiFlow (retrained on our training data) and DPLM-2 (co-generation) perform similarly on scTM and scRMSD metrics but diverge significantly on the pLDDT metric. A similar trend is observed when comparing Figures 3A and 3C. In Figure 3A, scTM and scRMSD metrics remain relatively stable as sequence length increases, with scRMSD even increasing for sequences of length 500. However, Figure 3C shows a notable upward trend in pLDDT as sequence length increases (from 100 to 300), without a significant drop in pLDDT even for sequences of length 500. These phenomena indicate that the scTM and scRMSD metrics convey somewhat contradictory information to the pLDDT metric. Generally, a significant drop in pLDDT should correspond with a decrease in scTM and an increase in scRMSD. My question here is whether this inconsistency might indicate that the pLDDT metric is unreliable. I raise this because, in previous experiments, we observed that pLDDT, being predicted by a neural network (e.g., ESM-2) trained on natural protein datasets, sometimes fails to handle model-generated proteins that deviate significantly or exhibit severe irregularities, leading to inflated pLDDT values. This phenomenon is quite common in unconditional generation. Therefore, I believe using a more diverse set of metrics is essential. However, you did not provide evaluation results for MultiFlow (retrained on our training data) on the Novelty and Diversity dimensions, which makes it harder to assess the model’s performance on this part of the task.
>
> **`A4`**: Thanks for pointing this out! After checking our code, we find that the results of MultiFlow (retrained on our training data) were obtained incorrectly. This occurred because we leveraged default MultiFlow sampling configuration that sampling proteins in lengths of [70, 100, 200, 300], inconsistent with the experimental settings in our paper that sample in lengths of [100, 200, 300, 400, 500]. We apologize for this mistake and we provide a updated results in the Table below with corrected scTM and scRMSD, where MultiFLow (retrained on our training data) achieves significantly lower scTM and scRMSD results compared with DPLM-2, aligning with the performance gap seen in pLDDT. We believe this is because MultiFlow's weaker ability to generate long proteins, resulting in a significant decline in designability when generating proteins of lengths 400 and 500.
>
> | Model                                        | scTM           | scRMSD         | pLDDT  | avg. inner-TM | MaxCluster |
> |----------------------------------------------|----------------|----------------|--------|---------------|------------|
> | Native PDB                                   | 0.904 ± 0.129  | 4.623 ± 5.688  | --     | 0.271 ± 0.020 | 0.776      |
> | MultiFlow (official ckpt)                    | 0.930 ± 0.098  | 3.208 ± 4.741  | 79.447 | 0.356 ± 0.013 | 0.500      |
> | Multiflow (w/o distillation) *               | 0.750 ± 0.163* | 9.306 ± 8.499* | 65.861 | 0.350 ± 0.038 | 0.490      |
> | Multiflow (retrained on our training data) * | 0.871 ± 0.934* | 6.580 ± 6.258* | 67.870 | 0.331 ± 0.052 | 0.440      |
> | DPLM-2 (650M)                                | 0.925 ± 0.085  | 3.899 ± 3.723  | 82.686 | 0.270 ± 0.018 | 0.545      |

---

> ### Author Response · Authors · 2024-11-24
> **Rebuttal by Authors**
>
> The pLDDT metric reflects the structural plausibility of sequence. In addition to the pLDDT metric, we also include sequence-related metric that we calculate sequence diversity using mmseq2 clustering at sequence identy = 0.5 for good quality samples with pLDDT > 70. This quality threshold for diversity is inspired by Multiflow, which is more informative by avoiding diverse but messy sequences.  We follow the experimental setting in DiMA [1], generating 2048 sequences with length sampled from the length distribution of PDB + SwissProt. The results highlight that DPLM-2 is able to generate structurally plausible and diverse sequences for protein generation. We also find that training data distillation greatly helps Multiflow's sequence quality in terms of pLDDT and diversity.
>
> |                                                          | ProGen2 [2] | CARP [3] | DiMA [1] (result from their paper) | EvoDiff | MultiFlow (official w/ distillation) | MultiFlow (retrained on DPLM-2 data) | DPLM  | DPLM2 |
> |----------------------------------------------------------|-------------|----------|------------------------------------|---------|--------------------------------------|--------------------------------------|-------|-------|
> | pLDDT                                                    | 57.2        | 30.0     | 83.3                               | 35.846  | 79.4                                 | 62.6                                 | 84.0  | 83.7  |
> | diversity (↑) / mmseq cluster at seq-id=0.5 & plddt > 70 | -           | 0.0      | -                                  | 0.020   | 0.860                                | 0.294                                | 0.745 | 0.755 |
>
> **<End of `Q4` >**

---

> ### Author Response · Authors · 2024-11-24
> **Rebuttal by Authors**
>
> > **`Q5`**: Regarding the inverse folding task: Comparing the performance of ESM3 and DPLM-2 in Table 5 reveals a pattern where DPLM-2 often exhibits a higher AAR while frequently showing a lower scTM. Could this possibly be due to biases in the protein structure prediction model (ESMFold) or because the sequences generated by DPLM-2, despite having higher ARR, may deviate further from the target protein’s data distribution?
>
> |                                                                                                            | AAR (mean/median) | scTM (mean/median) |
> |------------------------------------------------------------------------------------------------------------|-------------------|--------------------|
> | ESM-3 (1.4B)                                                                                               | 47.06/46.24       | 0.90/0.95          |
> | DPLM-2 (650M) + argmax                                                                                     | 49.01/50.10       | 0.88/0.93          |
> | DPLM-2 (650M) + temperature annealing sampling (linearly annealing from 2.2 -> 0.1 for 100 decoding steps) | 43.15/42.24       | 0.88/0.93          |
>
> **`A5`**: Thanks for your insightful question!
>
> **About scTM**: Although the actual gap between DPLM-2 (650M) and ESM-3 is not that large (0.88 vs 0.90 on cameo 2022), we suggest that this difference can be attributed to the lossy structure encoding caused by discrete tokenization. ESM3 introduces an important geometric attention module to encode atomic coordinate of input structure when available, while DPLM-2 currently relies on pure structure tokens. This also leads to the discussion of trade-offs and further directions of structure modeling in multimodal PLM, which exactly corresponds to your last question. Please take a look at our elaborate discussion on this in Q9.
>
> **About high AAR**: This mainly arises from the sampling strategy. Here we ablated the sampling strategy to study its impact. By default, we followed DPLM's approach, using argmax decoding, which selects the token with the highest probability at each timestep. This method generates sequences with high probabilities, resulting in strong amino acid recovery (AAR). In contrast, we introduced a sampling strategy with annealing temperature ranging from 2.2 to 0.1 to enhance sequence diversity. While this approach lowers AAR, it maintains the same scTM score as argmax decoding. This demonstrates that the temperature annealing strategy generates more diverse sequences. Although these sequences are less similar to the ground truth, they still satisfy the structural requirements, highlighting the trade-off between diversity and similarity to the target sequence.

---

> ### Author Response · Authors · 2024-11-24
> **Rebuttal by Authors**
>
> > **`Q6`**: Regarding repetition: In practice, we observed that diffusion protein language models tend to generate a large number of repetitive amino acids. I noticed that DPLM employed a resampling scheme to address this issue. Therefore, regarding the DPLM-2 model, I would like to know whether it also encounters similar repetition issues. If so, what approach did you adopt to resolve it? If not, I would like to know what prevented this issue in DPLM-2.
>
> **`A6`**: This is a great question.
>
> **About repetition issues in DPLM.** Generative models, especially language models, are good at learning what you provide with them. DPLM trained with UniRef50 contains a great number of sequences with high proportion repetition patterns. As a consequence, maximum-a-posterior (MAP) based sampling methods, such as default mask-predict sampling method for masked discrete diffusion, are likely to firstly recover these high likelihood patterns, and then get stuck in this local optimal and turn out to be low quality generation. This phenomenon, i.e., high likelihood != high generation quality, has been widely noticed in the studies of neural text/sequence generation [1,2]. In the original DPLM, authors address this issue with resampling for repetition patterns given a threshold.
>
> **About repetition issues in DPLM-2.** In DPLM-2, we train the model on pairs of structure tokens and amino acid tokens. As structure vocabulary is much larger than amino acid and structure data is much more complicated, there are basically no such repetitive structure tokens in the resulting structure tokens training data. Meanwhile, PDB and swissprot data are curated much more carefully than UniRef and AFDB in general (predicted structure from unannotated sequences, incl. UniRef), also resulting in less amino acid repetition in our training data. As a result, DPLM-2 just samples "good-looking" proteins (while amino acid repetitions often lead to long helical or disordered loopy proteins) using straightforward sampling strategy with no need for resampling. We can also further enhance sampling diversity in general using temperature annealing sampling, starting with high temperature (e.g., 2.2) for high stochacity and gradually decreasing towards low temperature (e.g., 0.1) for high likelihood and fidelity, similar to langevin dynamics or simulated annealing.
>
> [1] Is MAP Decoding All You Need? The Inadequacy of the Mode in Neural Machine Translation. Coling 2020
>
> [2[ The Curious Case of Neural Text Degeneration. ICLR 2020.

---

> ### Author Response · Authors · 2024-11-24
> **Rebuttal by Authors**
>
> > **`Q7`**: Regarding VQ-VAE: I am curious about the vocabulary usage rate of VQ-VAE when the vocabulary size is 1024. I noticed a significant performance difference between VQ-VAE and LFQ in this case, so I wonder if the vocabulary usage rate might be a contributing factor to this issue.
>
> **`A7`**:
> Thank you for your suggestion. Following your advice, we calculated the codebook utilization, as shown in the table below. We observed that LFQ-based tokenizers consistently achieve nearly 100% codebook utilization, with more evenly distributed code usage, whereas the vanilla VQ-VAE suffers from codebook collapse (63.5%). This suggests that severe codebook collapse limits the vanilla VQ-VAE’s ability to learn meaningful vocabulary, at least in our implementation. We are also aware of the successful use of vanilla VQ-VAE in ESM-3, so there might be significant optimization efforts that matter. Nevertheless, the combination of pre-trained structure encoder (GVP-Transformer) + LFQ + AF2-style structure decoder has been found to be an effective and efficient approach, requiring minimal twists for robust development.

---

> ### Author Response · Authors · 2024-11-24
> **Rebuttal by Authors**
>
> > **`Q8`**: Regarding catastrophic forgetting: I would like to confirm whether DPLM-2 in Table 7 still utilizes structural pre-training, given that neither of these models used large-scale sequence pre-training. If structural pre-training was indeed applied to DPLM-2, then there are two differences between DPLM and DPLM-2: one is the difference in model architecture, and the other is that DPLM-2 was pre-trained on structural data. With these two variables present, how can we determine that the performance decline in DPLM-2 is due to catastrophic forgetting?
>
> **`A8`**: The DPLM-2 in Table 7 does not utilize structural pretraining. We conduct this experiment because we find DPLM-2 demonstrates a significantly larger performance gap compared with the DPLM, which is used for parameter initialization for DPLM-2. As such, we think this task can be a good testbed for us to figure out the factors causing this degradation.
> We hypothesize two potential causes for the observed degradation:
>
> 1. DPLM-2 needs to accommodate additional structural representations given the same model capacity (parameters), which could negatively impact the representation learning performance.
> 2. As continuous training on smaller magnitude of structure data, DPLM-2 may experience catastrophic forgetting of the representation power gained during DPLM's large-scale sequence pretraining.
>
> To explore (1), we eliminated pretraining factors by retraining both DPLM and DPLM-2 with random initialization on the SwissProt and PDB datasets for 100K training steps. Additionally, we evaluated performance across all three tasks (HumanPPI, MetalIonBinding & DeepLoc) where DPLM-2 underperformed compared to DPLM. As shown in the table below, when large-scale sequence pretraining is removed, DPLM-2 significantly outperforms DPLM (exp 2 vs exp 1). This indicates that incorporating structural information enhances performance rather than harming it, which rejects the hypothesis (1).
>
> However, when DPLM undergoes large-scale pretraining and DPLM-2 is subsequently trained from the pretrained DPLM, the performance of DPLM-2 on certain tasks diminishes (exp 4 vs exp 3). Given the relatively smaller structure data for DPLM-2 training, this suggests that catastrophic forgetting occurs during DPLM-2's multimodal training, reducing the advantages of large-scale pretraining. To verify and mitigate this, during the course of rebuttal of last week, we have curated additional 1.3M predicted structures from AFDB_rep [1], and trained DPLM-2 on this larger data. The experimental results show that the amount of structure data is indeed a key factor for better multimodal protein representations, leading to significantly improved performance over the original data (exp 5 vs exp 4). In particular, on HumanPPI, enlarging data from 200K to 1.5M helps DPLM-2 attain 2.3% improvement, and also outperforms SaProt, a strong multimodal PLM trained with 40M foldseek tokenized AFDB data.
>
> | exp id |                                                                                                | HumanPPI (Acc%) | MetalIonBinding (Acc%) | DeepLoc (Acc%) |
> |--------|------------------------------------------------------------------------------------------------|-----------------|------------------------|----------------|
> | 0      | SaProt                                                                                         | 86.41           | 75.75                  | 85.57          |
> | 1      | DPLM (PDB + swissprot only)                                                                    | 73.33           | 62.25                  | 63.49          |
> | 2      | DPLM-2 (PDB + swissprot only)                                                                  | 77.22           | 69.47                  | 66.77          |
> | 3      | DPLM w/ fully pretraining on UniRef50                                                          | 86.41           | 75.15                  | 84.56          |
> | 4      | DPLM-2 w/ seq-pretraining (finetuned from DPLM) data: swissprot + pdb (200K)                   | 84.44           | 74.28                  | 82.98          |
> | 5      | DPLM-2 w/ seq-pretraining (finetuned from DPLM) data: AFDB_rep + Swissprot + pdb (1.3M + 200K) | 87.78           | -                      | 83.42          |
>
> [1] Clustering predicted structures at the scale of the known protein universe. Nature 2023

---

> ### Author Response · Authors · 2024-11-24
> **Rebuttal by Authors**
>
> > **`Q9`**: Regarding the structural information: Similar to ESM-3, you also use discrete structure tokens to represent structural information. This approach offers scalability benefits for the model, yet it also limits the model’s ability to capture finer structural details accurately. This trade-off represents an important challenge in the current field of multimodal protein language models. I am interested to know whether you would consider any methods to mitigate this issue when applying DPLM-2 to more structure-related tasks, as you described in Section 5.
>
> **`A9`**: This is a great question, and we fully agree that it represents a crucial challenge in token-based multimodal protein language models. albeit being the key enabler to multimodal PLMs, Structure tokenization is essentially clustering similar local structured environments, which results in lossy compression and the absence of fine-grained structural variation. We are very aware of this issue. The primary principle of the solution is that we need to "recover" and preserve the high-frequency variation that gets lost during quantization. We propose some potential directions for mitigation:
> - Separate structure encodings for DPLM-2. We can introduce different structure encoders for encoding and generation purposes, respectively. For parts of a protein where atomic coordinates are already provided, lossy tokenization may not be necessary. Instead, we can use robust features from powerful structure encoders like GVP-Transformer while continuing to use structure tokens for generating the remaining parts. To achieve this, the model can be trained to alternate between these two types of encodings. A similar approach has been applied successfully in recent vision-language MLLMs [1], as the vision-language community has also recognized that understanding and generation often require different types of representations.
> - Modeling continuous structure features with hybrid tokenization. In the structure tokenizer, the vector quantizer module converts encoder features into discrete structure token features, but the residuals—differences between the original and quantized features—are lost, removing fine-grained structural details. To address this, we can using continuous generative modeling, such as diffusion/flow-based models, to learn to recover these residuals. This would work by conditioning on the structure tokens and possibly the final hidden states of DPLM-2. The protein structure generation process would involve first generating discrete structure tokens that capture the overall topology, then using those tokens to generate the missing residuals. These residuals would be added up to the structure token embeddings to recover a more complete and accurate structure representation, closer to the features produced by the structure encoder. This approach could significantly improve structure generation. By combining this idea with hybrid structure encodings, DPLM-2 could not only interpret given structures at atomic accuracy but also generate structures that include the missing fine-grained variations. Similar strategies have shown significant success in visual autoregressive generation with visual tokenizers [2].
>
> [1] Janus: Decoupling visual encoding for unified multimodal understanding and generation. Arxiv 2024.
>
> [2] HART: Efficient Visual Generation with Hybrid Autoregressive Transformer

---

> ### Comment · Reviewer_FqCq · 2024-11-24
> **Thanks for your rebuttal !**
>
> I have carefully read all the responses, and I can say that these responses have addressed all my concerns. This is an excellent paper, and I believe it should be accepted by ICLR. Therefore, I have raised my score to 8.

---

> > ### Author Response · Authors · 2024-11-24
> > **Thank you very much!**
> >
> > Thank you for reading our rebuttal and for your supportive words! We're happy that we have addressed all your concerns! We would like to once again thank you for your comments, which are super inspiring and have indeed helped greatly enhance our paper.
> >
> >
> > Authors

---

### Official Review · Reviewer_4A34 · 2024-11-01

**Soundness:** 3
**Presentation:** 3
**Contribution:** 3
**Rating:** 8
**Confidence:** 4

**Summary:**

The paper presents DPLM-2, a multimodal protein foundation model designed to represent and generate protein sequences and structures. DPLM-2 introduces a token-based approach to protein structures, converting 3D backbone coordinates into discrete structure tokens. The model then processes structure token sequences alongside amino acid sequences, with aligned position encodings to reinforce residue-level correspondence between them, and is trained using a denoising objective.

Addressing the limitations of previous models like Multiflow, which lack sequence-based pretraining to capture co-evolutionary relationships, DPLM-2 leverages pretraining on unlabeled sequences and finetuning on structures, to learn the joint distribution of sequences and structures. This allows DPLM-2 to capture both modalities effectively and enables it to model joint, marginal, and conditional distributions.

Additionally, DPLM-2 demonstrates competitive performance across tasks such as unconditional generation, folding, inverse folding, and motif-scaffolding, showing its capability as a comprehensive multimodal protein modeling tool.

**Strengths:**

- Originality: The work's originality lies primarily in (1) the DPLM framework that combines sequence tokens and structure tokens with a lookup-free quantization (LFQ) VAE, and (2) the use of sequence pretraining followed by LoRA fine-tuning. These contributions provide valuable insights to the field. The authors also address relevant concurrent work effectively.

- Clarity: The paper is generally easy to follow, though some sections would benefit from further explanation and technical details.

- Empirical Performance: DPLM-2 demonstrates strong empirical performance across a wide range of generative tasks, including sequence-structure co-generation, folding, inverse folding, and motif-scaffolding. The paper also includes valuable ablation studies on sequence pretraining and data augmentation, offering insights into the model’s effectiveness and robustness.

**Weaknesses:**

-Missing Details: Key details are missing in certain sections. For instance:

     (1)A core contribution is the DPLM-2 model itself, but essential details, such as how sequence and structure tokens are combined and how the structure tokenizer is trained, are not included in the main text. Moving these details from the appendix to the primary text, with clear explanations including the frozen pretrained structure encoder and the sequence prediction head on top of the structure decoder, would significantly improve clarity.

     (2)The distinct noise level for each modality could be better explained, as this aspect is currently underdeveloped.

     (3)In Section 4.2, the authors mention that performance can improve with supervised fine-tuning using a folding objective; however, the paper lacks details on this fine-tuning process.

- Clarity and Consistency Issues: Minor inconsistencies reduce clarity, such as inconsistent bolding of best results in Tables 2 and 6. In Table 4, DPLM-2 is presented as performing well in zero-shot folding, yet its RMSD is high compared with ESM3, PVQD, and ESMFold, achieving competitive performance only after fine-tuning (SFT). Additionally, clarifying the presentation of mean and median values could help with data interpretation.

- Mixed Results: The results on certain tasks are not fully convincing. In particular, while DPLM-2 exhibits high scTM in unconditional protein generation, it does not achieve lower scRMSD compared to other methods, indicating potential limitations in generation quality.

**Questions:**

- Sequence tokens use a smaller vocabulary than structure tokens. Are the corresponding structural embeddings in DPLM-2 trained from scratch during fine-tuning?
- While DPLM-2 enables flexible generation, what are the trade-offs in structural invariance when using structure tokens instead of 3D coordinates?
- In Table 2, what is meant by DPLM-2 (seq → structure) or (structure → seq)? If these indicate modality-by-modality generation, could you clarify how this is implemented?
- For the protein representation learning evaluation, it might be useful to include a broader range of baselines, such as GNN-based models, for a more comprehensive comparison.
- In Section 3.3, you state, “Recent efforts have applied this approach to protein structure coordinates (Van Kempen et al., 2024; Liu et al., 2023; Gao et al., 2024; Lu et al., 2024). This allows language models to better learn the composition of local structural elements. However, how to learn an effective structure tokenizer remains an active research question.” Given the similarity of your method to prior approaches, particularly with the use of LFQ as in Gao et al., 2024, could you elaborate on how your method contributes to addressing this active research question?

---

> ### Author Response · Authors · 2024-11-24
> **Rebuttal by Authors**
>
> Thank you so much for your constructive suggestions. We understand your concerns about the clarity issues and missing technical details, potential inconsisency in some result intepretation, and requiring analysis and discussion on structure tokenizer. To address these issues, we have made our major efforts in (1) elaborating on technical details and accordingly updating the manuscript for improved clarity; (2) providing discussion on the implications of in silico designablity metrics; and (3) providing more analysis and discussion on structure tokenization. Your insightful comments and suggestions have greatly improved our manuscript. We sincerely thank you once again and welcome any further feedback!

---

> ### Author Response · Authors · 2024-11-24
> **Rebuttal by Authors**
>
> > **`Q1`**: Missing Details: Key details are missing in certain sections. For instance:
>
> > **`Q1.1`**: A core contribution is the DPLM-2 model itself, but essential details, such as how sequence and structure tokens are combined and how the structure tokenizer is trained, are not included in the main text. Moving these details from the appendix to the primary text, with clear explanations including the frozen pretrained structure encoder and the sequence prediction head on top of the structure decoder, would significantly improve clarity.
>
> **`A1.1`**: Thanks for your valuable suggestions on clarity issues! We have updated our paper by moving the essential details into the main text, and alternating ablation study (section 4.1.3) to the appendix.
>
> > **`Q1.2`**: The distinct noise level for each modality could be better explained, as this aspect is currently underdeveloped.
>
> **`A1.2`**: Thank you for pointing this out. We introduce a distinct scheduler to control the noise level of structure and sequence flexibly during training (section 3.1 in our paper). Different combinations of structures and sequence schedulers (denoted as $t_z$ and $t_s$, respectively) imply training for different applications. Specifically, we mainly focus on: (1) sequence-conditioned structure generation (e.g., folding),  (2) structure-conditioned sequence generation (e.g., inverse-folding),  (3) sequence generation,  (4) structure generation,  (5) structure-sequence co-generation.
>
> For conditional generation tasks (e.g., folding and inverse-folding), we set the noise scheduler of the conditioned modality to 0, e.g., no noise in the conditioned modality. Specifically, in the folding task, the $t_s$ is always set to $0$, while in the inverse-folding tasks the $t_z$ is always set to $0$.
>
> In the structure-sequence co-generation task, we keep the $t_z$ and $t_s$ for the same, enhancing the structure-sequence consistency in co-generation. The structure or sequence generation tasks do not depend on another modality, so we set the noise scheduler of another modality to $T$, which is the maximum timestep and means 100% noise in another modality. For example, in structure generation task, the $t_s$ is always set to $T$.
>
> During training, we jointly train the above 5 tasks simultaneously. We divide the training data in a batch into 5 parts according to a preset proportion, and each part is used for a specific task training. In our experiment, the proportion for each task is the same, which is 20%.
>
> > **`Q1.3`**: In Section 4.2, the authors mention that performance can improve with supervised fine-tuning using a folding objective; however, the paper lacks details on this fine-tuning process.
>
> **`A1.3`**: According to the above question, we can further enhance a specific generation task by supervised finetuning (SFT). This involves continuing training for the specific task with a proportion of 100%, while the proportion for other tasks is set to 0%. For example, in Tab. 4, the folding supervised finetuning is performed by continuous training from a pre-trained DPLM-2 with 100% proportion of folding objective using the same training data for additional 50K steps with a constant and smaller learning rate (5e-5 vs 1e-5 for joint pre-training).

---

> ### Author Response · Authors · 2024-11-24
> **Rebuttal by Authors**
>
> > **`Q2`**: Clarity and Consistency Issues: Minor inconsistencies reduce clarity, such as inconsistent bolding of best results in Tables 2 and 6. In Table 4, DPLM-2 is presented as performing well in zero-shot folding, yet its RMSD is high compared with ESM3, PVQD, and ESMFold, achieving competitive performance only after fine-tuning (SFT). Additionally, clarifying the presentation of mean and median values could help with data interpretation.
>
> **`A2`**: Thank you for pointing out the misleading and we have changed the way of highlighting in our paper: the highest number is bold, and the second highest number is underlined. The current results of conditional generation (Table 4, Table 5) are presented as mean/median values, and we have updated our paper for clearer clarification.
>
> **About SFT**: The results of PVQD and ESMFold are actually obtained by supervised finetuning on folding-only objective, so it is meaningful and fair to compare to DPLM-2 with folding SFT too.

---

> ### Author Response · Authors · 2024-11-24
> **Rebuttal by Authors**
>
> > **`Q3`**: Mixed Results: The results on certain tasks are not fully convincing. In particular, while DPLM-2 exhibits high scTM in unconditional protein generation, it does not achieve lower scRMSD compared to other methods, indicating potential limitations in generation quality.
>
> **`A3`**: Thank you for pointing this out. This is a great question about the evaluation metrics and their interpretation for the in silico designability of protein generation or design.  We would like to address your question as follows:
> |                                            | scTM           | scRMSD        | helix ratio | strand ratio | coil ratio |
> |--------------------------------------------|----------------|---------------|-------------|--------------|------------|
> | native PDB samples                         | 0.904 ± 0.129  | 4.623 ± 5.688 | 0.36        | 0.22         | 0.42       |
> | Multiflow (w/ distilation)                 | 0.930 ± 0.098  | 3.208 ± 4.741 | 0.75        | 0.10         | 0.15       |
> | Multiflow (w/o distillation)               | 0.750 ± 0.163  | 9.306 ± 8.499 | 0.71        | 0.06         | 0.23       |
> | Multiflow (retrained on our training data) | 0.871 ± 0.934  | 6.580 ± 6.258 | 0.57        | 0.16         | 0.26       |
> | DPLM-2                                     | 0.925 ± 0.085  | 3.899 ± 3.723 | 0.48        | 0.17         | 0.35       |
>
> We first need to highlight that the generated samples from DPLM-2 share similar scTM (0.925) and scRMSD (3.9) as native PDB samples, which also exhibit good scTM (0.904) with a little bit higher scRMSD (4.623). Moreover, Additionally, DPLM-2 maintains a balanced structural composition (helix: 0.4, strand: 0.2, coil: 0.45), closely resembling natural distributions. In contrast, for MultiFlow, the officially released model with distillation attains much lower scRMSD (3.2), while the performance of our retrained version (on the same DPLM-2 training set) degrades in both scTM (0.871) and scRMSD (6.58). Lower scRMSD in MultiFlow with distillation, appears to be driven by overrepresentation of structured elements (Figure 4A), i.e., significantly biasing towards proteins with more helices, with less strands and loops (also see Figure 4C). This overrepresentation drives the observed scRMSD improvement but deviates from natural protein diversity.
>
> TM-score emphasizes global topology, while RMSD is sensitive to local structural errors. As such, although scTM and scRMSD are generally correlated, discrepancies can arise. The purpose of TM-score is to solve this sensitivity of RMSD, because RMSD is an average distance of all residue pairs in two structures, a local error (e.g. a misorientation of the tail) will raise a big RMSD value although the global topology is correct. In TM-score, however, the small distance is weighted stronger than the big distance, which makes the score insensitive to the local modeling error.
>
> As shown in Fig 4B, some samples from DPLM-2 with higher loop proportion are more conformationally flexible, hence may show high scTM (>0.9) but worse scRMSD (>2.0), similar to natural protein. However, this does not necessarily indicate a limitation in generation quality but reflects differences in metric sensitivity.
>
> As a result, the in-silico designability of protein generation should be evaluated comprehensively using both scTM and scRMSD, as each metric offers distinct insights and serves different purposes. For users aiming to generate samples with accurate global topology, scTM serves as a reliable indicator, whereas scRMSD may occasionally exclude reasonable structures. Conversely, for applications requiring structurally rigid and stable proteins, such as functional designs (e.g., binder design), scRMSD has been shown to correlate more strongly with in vitro success rates, as suggested by RFDiffusion.

---

> ### Author Response · Authors · 2024-11-24
> **Rebuttal by Authors**
>
> > **`Q4`**: Sequence tokens use a smaller vocabulary than structure tokens. Are the corresponding structural embeddings in DPLM-2 trained from scratch during fine-tuning?
>
> **`A4`**: That's correct. The structural token embeddings in DPLM-2 are trained from scratch. To efficiently utilize the evolutionary information from pre-trained sequence-based DPLM, DPLM-2 uses a warm-up strategy (outlined in Section 3.2 of our paper). This approach initializes DPLM-2 with the weights of the sequence-trained DPLM.
>
> More specifically, since DPLM's vocabulary consists only of amino acids, DPLM-2 expands this with discrete structure tokens. The embeddings for these new tokens are initialized using the mean and standard deviation of the learned amino acid embeddings. This embedding initialization keeps the distributional statistics of the embedding space consistent with the pre-trained DPLM, ensuring stable early-stage training (for learning structure-sequence alignment) and reducing the risk of extreme gradients that could cause training instability.

---

> ### Author Response · Authors · 2024-11-24
> **Rebuttal by Authors**
>
> > **`Q5`**: While DPLM-2 enables flexible generation, what are the trade-offs in structural invariance when using structure tokens instead of 3D coordinates?
>
> **`A5`**:
> Thank you for your insightful question.
>
> Our approach can be seen as decoupling the learning of the structurally invariant topology to the tokenizer and the language model, and the geometric reasoning on 3D coordinates to the structure decoder parameterized by triangular modules and IPAs from AlphaFold2 with FAPE loss. This shares the similarity to AF2 and ESMfold, where the sequence encoding modules like Evoformer (co-evolution encoding) and ESM (amino-acid encoding) provide invariant features (in the form of single and pair embeddings) to the structure decoder that learns to convert invariant features into 3D coordinates. The AF2-style structure decoder does not enforce strict equivariance to rigid transformations. Instead, it relies on the FAPE loss to ensure structural consistency, which minimizes coordinate errors in a manner that is invariant to global rotations and translations.
>
> As such, we suggest that the primary trade-off when using invariant structure tokens instead of 3D coordinates mainly lies in the potential loss of fine-grained structural details. Structure tokens cluster similar local structures into discrete representations, which inherently introduce quantization errors. This trade-off, on one hand, enables efficient multimodal learning and generation by simplifying the representation space, on the other hand, represents an important challenge in the current field of multimodal protein language models as suggested by the Reviewer FqCq (Q9) and in our discussion section. Future efforts should be made towards better structure modeling to mitigate this trade-off. Some potential directions include introducing separate structure encoders for structure encoding and generation purposes [1], or hybrid tokenization for recovering the lost fine-grain structural variations [2].
>
> [1] Janus: Decoupling visual encoding for unified multimodal understanding and generation. Arxiv 2024.
>
> [2] HART: Efficient Visual Generation with Hybrid Autoregressive Transformer.

---

> ### Author Response · Authors · 2024-11-24
> **Rebuttal by Authors**
>
> > **`Q6`**: In Table 2, what is meant by DPLM-2 (seq → structure) or (structure → seq)? If these indicate modality-by-modality generation, could you clarify how this is implemented?
>
> **`A6`**:
> Thanks for pointing this out, we have updated our paper to improve clarity. The co-generation can be performed in simultaneous generation (co-generation) and cascaded workflow: first generating the structure then the sequence conditioned on generated structure (struct → seq), and the reverse way (seq → struct), without the need of other folding or inverse folding models.
>
> > **`Q7`**: For the protein representation learning evaluation, it might be useful to include a broader range of baselines, such as GNN-based models, for a more comprehensive comparison.
>
> **`A7`**:
> Thanks for your valuable suggestion! We have accordingly added the results of GearNet [1], as the GNN-based baseline, as you suggested. We have updated the results in Table 6 of our paper.
>
> [1] Protein representation learning by geometric structure pretraining. ICLR 2023

---

> ### Author Response · Authors · 2024-11-24
> **Rebuttal by Authors**
>
> > **`Q8`**: In Section 3.3, you state, “Recent efforts have applied this approach to protein structure coordinates (Van Kempen et al., 2024; Liu et al., 2023; Gao et al., 2024; Lu et al., 2024). This allows language models to better learn the composition of local structural elements. However, how to learn an effective structure tokenizer remains an active research question.” Given the similarity of your method to prior approaches, particularly with the use of LFQ as in Gao et al., 2024, could you elaborate on how your method contributes to addressing this active research question?
>
> **`A8`**:
> Thank you for your question. Our approach addresses the active research question of structure tokenization through a simpler and more practical design compared to Gao et al., 2024:
>
> **Simplicity of design and training**: We use a strong pretrained structure encoder (from ESM-IF) with an AF2-style decoder (triangular modules + IPA) trained using FAPE loss. There are also no unnecessary modifications to LFQ. In contrast, Gao et al introduced very complicated modifications to LFQ (SoftLFQ). This makes our method fairly easy and straightforward to implement, as well as effective and efficient for training.
>
> **Performance**: Despite its simplicity, our tokenizer performs competitively in structure reconstruction (recRMSD), demonstrating its ability to effectively capture structural features. Higher reconstruction does not necessarily lead to better generation quality, which is widely observed in VQ-VAE literature while the LFQ paper [2] also clearly elaborated on this. We in this paper have shown that our tokenizer can actually support various generation tasks with decent generation performance, while Gao et al., (2024) mainly focused on evaluation of reconstruction. For generations, they only assessed antibody CDR infilling, which is fairly easy, and general protein generation ability of their tokenizer remians unclear.
>
> [1] Gao et al.: FoldToken: Learning Protein Language via Vector Quantization and Beyond. Arxiv 2024
>
> [2] Language Model Beats Diffusion -- Tokenizer is Key to Visual Generation. ICLR 2024

---

> > ### Comment · Reviewer_4A34 · 2024-11-24
> >
> > I have carefully reviewed all the reviews and responses. I believe the responses have addressed all my concerns, and the authors have conducted extensive experiments and presented their work effectively. Therefore, I have raised my score to 8.

---

> > > ### Author Response · Authors · 2024-11-25
> > > **Thank you so much!**
> > >
> > > Dear Reviewer 4A34,
> > >
> > > We are thrilled that our responses have addressed all of your concerns, and we sincerely appreciate your supportive feedback and the increased rating!
> > >
> > > Your thoughtful suggestions and encouraging words mean a lot to us. We will surely make more efforts to include suggestions, discussions, and new results in a better form of presentation in the future revision. Your feedback has been invaluable in helping us improve, and we cannot thank you enough for your support!
> > >
> > > Best,
> > >
> > > Authors

---

### Official Review · Reviewer_f8By · 2024-11-04

**Soundness:** 3
**Presentation:** 3
**Contribution:** 2
**Rating:** 3
**Confidence:** 5

**Summary:**

This paper introduces DPLM-2, a multimodal discrete diffusion protein language model that can simultaneously generate both protein sequences and their 3D structures. The model extends the DPLM protein language model by incorporating structural information through a lookup-free quantization-based tokenizer that converts 3D coordinates into discrete tokens. DPLM-2 is trained on both experimental structures from PDB and synthetic structures from AFDB-SwissProt.

**Strengths:**

1. The paper is clearly written.

2. The model achieves simultaneous structure-sequence generation without requiring a two-stage training approach.

3. The experiments on learning structure tokenization are valuable for the community.

**Weaknesses:**

1. The DPLM-2 model is trained starting from the weights of DPLM model. As evident from e.g. Table 2 (poor diversity) and Table 5 (high AAR), DPLM suffers from mode collapse. Training DPLM-2 starting from such a checkpoint  leads to severe mode collapse in DPLM-2. Generating the same result over and over again makes a generative model useless. Training both models from scratch rather than finetuning might help and provide the insight on how adding structural information to the DPLM architecture affects the generation ability. Even better would be a thourough ablation study of the training procedure.

2. Authors compare DPLM and DPLM-2 only with EvoDiff on unconditional sequence generation. Adding more baselines of different architectures (AR transformers, CARP, continuous diffusion, flow-matching, etc.) would greatly improve the work.

3. There are some issues with model evaluation. First, the model is evaluated only structurally, but it is langugae model after all. Using sequence-based evaluation metrics, including sequence clustering for the diversity should benefit the soundness of the work. Second, it is not clear, why the Authors use pdb-TM (claculate TM-score against PDB), if the model is trained also on the synthetic data. Third, the designability metric used in this work evaluates the consistensy between the generated structure and the prediction of ESMFold on the generated sequence. It does not measure protein "quality".

4. Authors compare DPLM-2 on inverse folding task with weak baselines. Adding recognized IF models would greatly benefit the work. The same goes to other tasks, e.g. representation learning.

5. The dataset preparation included random cropping. It is not clear if it has a detrimental effect on the model behaviour.

6. The paper lacks analysis on the trained structural tokens. Additional exploration of the interpratability of the tokens themselves and untilization of the codebooks would greatly benefit the paper. Do the tokens correspond to some local environments in structure, or are they just abstract entities?

**Questions:**

1. The model does not learn the distribution of protein lengths. Have you tried to overcome this limitation?

2. The ablation results presented in 4.1.3 and table 3 are controversial. Could you please clarify the procedure, how many samples was used for evaluation and so on?

3. On page 9, line 442 is stated that DPLM-2 adopts argmax decoding. In the original DPLM paper argmax did not work. Can you elaborate on this?

4. The experiment on DeepLoc is not described in sufficient detail. Why you chose to use only one tiny dataset? Could you provide experiments that show the described catastrophic forgetting issue, which is of high importance?

5. One of the main claims of the paper states that the co-generation guarantees consistency between structure and sequence. This is a strong statement that requires strong evidence. However on line 223 the assumption of conditional independence is made. Can you provide a rigorous mathematical proof that garantees such consistensy?

---

> ### Author Response · Authors · 2024-11-24
>
> Thank you so much for your constructive suggestions. We understand your concerns about the potential risks of reduced sampling diversity and mode collapse, the need for more evaluation metrics, results, and explanations for a comprehensive assessment, and the theoretical guarantee of structure-sequence consistency under the conditional independence assumption in diffusion models. To address these issues, we have made our major efforts in (1) providing a discussion on diversity metrics with updated results and an ablation study on how the training strategies influence generation diversity; (2) including sequence-based metrics for unconditional generation, conducting sampling from length distribution of natural proteins, analysis on structure tokens and discussion of sampling strategies for conditional generation (3) providing discussions in theoretical aspect on the conditional independence assumption in diffusion models, including DPLM-2 and beyond. Your insightful comments and suggestions have greatly improved our manuscript. We sincerely thank you once again and welcome any further feedback!
>
> > `Q1:` The DPLM-2 model is trained starting from the weights of DPLM model. As evident from e.g. Table 2 (poor diversity) and Table 5 (high AAR), DPLM suffers from mode collapse. Training DPLM-2 starting from such a checkpoint leads to severe mode collapse in DPLM-2. Generating the same result over and over again makes a generative model useless. Training both models from scratch rather than finetuning might help and provide the insight on how adding structural information to the DPLM architecture affects the generation ability. Even better would be a thourough ablation study of the training procedure. The dataset preparation included random cropping. It is not clear if it has a detrimental effect on the model behaviour.
>
> `A1`: Thanks for this valuable question. We understand you are concerned about the sampling diversity and the risk of mode collapse of DPLM-2, and wondering how specific training strategies (e.g., multimodal finetuning from pre-trained DPLM) relate to the model behaviors.
>
> **Regarding sampling diversity and risk of mode collapse.**
>
> We thank the reviewer for bringing this to our attention. We notice that there are inconsistent results of diversity in Table 2 between `avg inner-TM` (averaged pairwise TM-score among samples) and `MaxCluster` (number of distinct structure clusters), which do not make sense. For example, DPLM-2 (co-generation), attains a `MaxCluster=0.545` (i.e., there are 54.5 distinct fold clusters out of 100 samples determined by Foldseek with TM-score threshold = 0.5), which is better than the `MaxCluster=0.500` for the official Multiflow (w/ distillation), whereas the `avg inner-TM` is unreasonably high (0.703 vs 0.468 for MultiFlow), which contradicts with the value of MaxCluster and the visualized observation we perceive from the predicted structures.
>
> We have accordingly checked our code and found out that our implementation of avg inner-TM was mistakenly incorrect. Here we provide a updated results in the Table below with corrected avg inner-TM, where DPLM-2 gets reasonable avg inner-TM similar to native PDB samples (with the same length intervals) and lower than MuliFlow and RFDiffusion. These results empirically indicate that DPLM-2 does not struggle with mode collapse issue during sampling. Plus, we can further improve sampling diversity by using temperature-annealed sampling with higher initial temperature.
>
>
> | Model                                              | scTM          | scRMSD        | avg. inner-TM | MaxCluster | helix ratio | strand ratio |
> |----------------------------------------------------|---------------|---------------|---------------|------------|-------------|--------------|
> | Native PDB                                         | 0.904 ± 0.129 | 4.623 ± 5.688 | 0.271 ± 0.020 | 0.776      | 0.36        | 0.22         |
> | MultiFlow (official ckpt)                          | 0.930 ± 0.098 | 3.208 ± 4.741 | 0.356 ± 0.013 | 0.500      | 0.75        | 0.10         |
> | RFDiffusion                                        | 0.914 ± 0.155 | 1.969 ± 4.073 | 0.352 ± 0.025 | 0.598      | 0.62        | 0.18         |
> | DPLM-2                                             | 0.925 ± 0.085 | 3.899 ± 3.723 | 0.270 ± 0.018 | 0.545      | 0.46        | 0.16         |
> | DPLM-2 (temperature-annealed sampling: 2.2 -> 0.1) | 0.883 ± 0.120 | 5.447 ± 5.477 | 0.275 ± 0.031 | 0.584      | 0.43        | 0.18         |

---

> > ### Author Response · Authors · 2024-11-24
> >
> > **Regarding ablation study of training strategies.**
> >
> > As you suggested, here we provide a comprehensive ablation study extending the Table 3 of our original manuscript to examine the effects of four training strategies on DPLM-2-650M: sequence pre-training (finetuning from seq.-pretrained DPLM), data augmentation with predicted structures, random length cropping (cutting off long proteins with a 50% probability per sample to increase diversity), and self-mixup for mitigating exposure bias in discrete diffusion.
> >
> > As seen in the following table, predicted structure from swissprot benefit designability (exp 2 vs exp 1); (2) finetuning from pre-trained DPLM leads to improved designability and sampling diversity (exp 5 vs exp 2, exp 3 vs exp 1); (3) absence of length cropping results in considerable degradation of diversity (exp 4 vs exp 5) ; and (4) self-mixup training significantly boosts sampling diversity while preserving decent designability (exp 6 vs exp 5). Remarkably, training a multimodal PLM from scratch is challenging when only limited structural data is available and the extensive sequence data is underutilized. To address this, the four proposed training strategies have proven highly effective, enabling DPLM-2 to be established efficiently (1.5 days on 16 A100 GPUs for the DPLM-2 650M model). Moreover, we believe that a well-designed data mixture policy incorporating all available multimodal protein data can further improve multimodal PLM training, as demonstrated by the success of ESM-3.
> > | Exp id    | finetuning from seq.-pretrained DPLM | predicted structures | random length cropping | self-mixup training | scTM          | MaxCluster |
> > |-----------|--------------------------------------|----------------------|------------------------|---------------------|---------------|------------|
> > | 1         | no                                   | no                   | yes                    | no                  | 0.702 ± 0.259 | 0.274      |
> > | 2         | no                                   | yes                  | yes                    | no                  | 0.886 ± 0.137 | 0.214      |
> > | 3         | yes                                  | no                   | yes                    | no                  | 0.889 ± 0.158 | 0.396      |
> > | 4         | yes                                  | yes                  | no                     | no                  | 0.937 ± 0.061 | 0.168      |
> > | 5         | yes                                  | yes                  | yes                    | no                  | 0.916 ± 0.099 | 0.440      |
> > | 6 (paper) | yes                                  | yes                  | yes                    | yes                 | 0.925 ± 0.085 | 0.545      |
> >
> > `--end of Q1--`

---

> ### Author Response · Authors · 2024-11-24
>
> > `Q2:` Authors compare DPLM and DPLM-2 only with EvoDiff on unconditional sequence generation. Adding more baselines of different architectures (AR transformers, CARP, continuous diffusion, flow-matching, etc.) would greatly improve the work. The model is evaluated only structurally, but it is langugae model after all. Using sequence-based evaluation metrics, including sequence clustering for the diversity should benefit the soundness of the work.
>
>
> `A2:` Thanks for your valuable suggestion on adding more baselines and sequence clustering for sequence diversity. We have added more baselines with various architectures on unconditional sequence generation, including autoregressive language models (Progen2), CNNs (CARP), continuous diffusion on ESM2 embedding space (DiMA) and flow matching (MultiFlow). We also include sequence diversity using mmseq2 clustering at sequence identy = 0.5 for good quality samples with pLDDT > 70. This quality threshold for diversity is inspired by Multiflow, which is more informative by avoiding diverse but messy sequences.  We follow the experimental setting in DiMA [2], generating 2048 sequences with length sampled from the length distribution of PDB + SwissProt. The results highlight that DPLM-2 is able to generate structurally plausible and diverse sequences for protein generation. We also find that training data distillation greatly helps Multiflow's sequence quality in terms of pLDDT and diversity.
>
>
> |                                                          | ProGen2 [1] | CARP [2] | DiMA [3] (result from their paper) | EvoDiff | MultiFlow (official w/ distillation) | MultiFlow (retrained on DPLM-2 data) | DPLM  | DPLM2 |
> |----------------------------------------------------------|-------------|----------|------------------------------------|---------|--------------------------------------|--------------------------------------|-------|-------|
> | pLDDT                                                    | 57.2        | 30.0     | 83.3                               | 35.846  | 79.4                                 | 62.6                                 | 84.0  | 83.7  |
> | diversity (↑) / mmseq cluster at seq-id=0.5 & plddt > 70 | -           | 0.0      | -                                  | 0.020   | 0.860                                | 0.294                                | 0.745 | 0.755 |
>
> [1] Progen2: Exploring the Boundaries of Protein Language Models. 2022
>
> [2] CARP: Convolutions are competitive with transformers for protein sequence pretraining. Cell Systems 2024
>
> [3] DiMA: Diffusion on language model embeddings for protein sequence generation. Arxiv 2024
>
>
>
> > `Q3:` There are some issues with model evaluation. It is not clear, why the Authors use pdb-TM (claculate TM-score against PDB), if the model is trained also on the synthetic data. Third, the designability metric used in this work evaluates the consistensy between the generated structure and the prediction of ESMFold on the generated sequence. It does not measure protein "quality".
>
> `A3:` Thank for your suggestions!
>
> **About "pdb-TM"**. We use "pdb-TM" to assess the novelty of proteins generated by DPLM-2 against natural proteins. As you suggested, it is indeed more reasonable to have TM-scores against swissprot and pdb structures (swissprot & pdb TM). As shown in the table, the "SwissProt & PDB TM" scores for DPLM-2 (650M) align closely with the "pdb-TM" results, consistent with the findings presented in our paper. These results from unconditional sampling indicate that DPLM-2, as a generative model of protein structure and sequence, learns the training data distribution and serves as a foundation for various potential conditional purposes.  Finally, we will change to "avg. max-TM" in our revised version as used in Flowflow-2 [1] for a more accurate terminology.
>
> |               | pdb-TM        | swissprot & pdb TM |
> |---------------|---------------|--------------------|
> | DPLM-2 (650M) | 0.640 ± 0.204 | 0.744 ± 0.170      |
>
> **About "quality"**. Thank you for pointing this out! We initially followed FrameDiff and Multiflow to term "quality" for self-consistency based designability for generated structures. To avoid any ambiguity, we will rephrase our wording by rephrasing "quality" to "designability".
>
> [1] Sequence-Augmented SE(3)-Flow Matching For Conditional Protein Backbone Generation. NeurIPS 2024

---

> > ### Author Response · Authors · 2024-11-24
> >
> > > `Q4:` On page 9, line 442 is stated that DPLM-2 adopts argmax decoding. In the original DPLM paper argmax did not work. Can you elaborate on this?
> >
> > `A4`: What you mention here is about the decoding strategy for inverse folding and forward folding, where strong and clear conditioning information is given. In the original DPLM paper, their inverse-folding results were actually also obtained by argmax decoding so as to maximize generation accuracy, while stochastic sampling was used for unconditional generation or scaffolding. In DPLM-2, we follow their settings for a fair comparison.
> >
> > We understand that you are also concerned about if high AAR indicates DPLM-2 just overfit or collapse to native sequence and less generalizable for "de novo design". This is not the case. Here, we provide ablation study on the sampling strategy in inverse folding. The argmax decoding strategy picks the token with highest probability at each timestep, yielding sequence with high probability and resulting in high amino acid recovery (AAR). On the other hand, we employ a sampling strategy with annealing temperature from 2.2 to 0.1 to improve diversity, and the generated sequence has a lower AAR while maintaining the same scTM as argmax decoding. This demonstrates that temperature annealing sampling strategy is capable of generating more diverse sequences that, while not similar to the ground truth, still meet the given structural conditions.
> >
> > |                               | AAR         | scTM      |
> > |-------------------------------|-------------|-----------|
> > | argmax decoding               | 49.01/50.10 | 0.88/0.93 |
> > | Temperature-annealed sampling | 43.15/42.24 | 0.88/0.93 |
> >
> >
> >
> > > `Q5:` Authors compare DPLM-2 on inverse folding task with weak baselines. Adding recognized IF models would greatly benefit the work. The same goes to other tasks, e.g. representation learning
> >
> > `A5:` Thanks for your valuable suggestion. **For inverse folding task**, we mainly focus on the comparison with other multimodal generative models (MultiFlow, ESM3) in our paper. We have also added more recognized baseline methods in inverse folding evaluation (ProteinMPNN & LM-Design [1]). We conduct experiments on CAMEO 2022 testset. We find that DPLM-2 is able to achieve close results with the strong baselines despite slightly lower scTM. To further improve scTM to bridge the last gap, there are several potential directions: (1) Inverse folding SFT: DPLM-2 conducts this task in a zero-shot manner while other systems are purpose-built models, thus task-oriented SFT could help as we have observed in folding ; (2) better structure modeling includes introducing separate structure encoders for structure encoding and generation purposes [3], or hybrid tokenization for recovering the lost fine-grain structural variations [4].
> >
> > | Model                                       | AAR         | scTM      |
> > |---------------------------------------------|-------------|-----------|
> > | ProteinMPNN                                 | 44.45/45.93 | 0.91/0.97 |
> > | LM-Design                                   | 56.40/58.24 | 0.88/0.96 |
> > | DPLM-2 650M (argmax decoding)               | 49.01/50.10 | 0.88/0.93 |
> > | DPLM-2 650M (temperature-annealed sampling) | 43.15/42.24 | 0.88/0.93 |
> >
> >
> > *For representation learning evaluation*, as you suggested, we have added GearNet [2], as the GNN-based baseline, for a broader range of baselines. We have updated the results in Table 6 of our paper.
> >
> > ---
> >
> > [1] Structure-informed language models are protein designers. ICML 2023
> >
> > [2] Protein representation learning by geometric structure pretraining. ICLR 2023
> >
> > [3] Janus: Decoupling visual encoding for unified multimodal understanding and generation. Arxiv 2024.
> >
> > [4] HART: Efficient Visual Generation with Hybrid Autoregressive Transformer.

---

> ### Author Response · Authors · 2024-11-24
>
> > `Q6:` The paper lacks analysis on the trained structural tokens. Additional exploration of the interpratability of the tokens themselves and untilization of the codebooks would greatly benefit the paper. Do the tokens correspond to some local environments in structure, or are they just abstract entities?
>
> `A6:` Thanks for your suggestion. In the Figure 2 of the original submission, we have shown the reconstruction accuracy and an interpretation analysis on the correspondence of structure tokens and local structural elements in terms of secondary structures. As you suggested, we calculate the codebook utilization in the following table. We find that LFQ-based tokenizers always achieve nearly 100% codebook utilization with more evenly distributed code usage, while vanilla VQ-VAE struggles with codebook collapse.
>
> For interpretability, we also update a more informative simplex plot of struct token vs second struct in Fig. 2B. We can observe a strong correlation between a vast majority of the structure tokens and structured local environments, where a lot of structured tokens concentrate on the alpha helix and beta sheet vertices, while some tokens lie between regions or the loop vertice. There are also a subset of struct tokens having less clear clues to specific secondary structures. This suggests that structure tokens mostly capture clear secondary elements, some may correspond to structured local environments (in bewteen helic and sheet), while others could be high-level abstract entities.
>
> | tokenizer | codebbok size | codebook untilization | train | cameo 2022 |  |  |
> |---|---|---|---|---|---|---|
> |  |  |  | lddt_ca | lddt_ca | tm-score | rmsd |
> | VQ-VAE-1k | 1024 | 63.50% | 0.76 | 0.71 | 0.8 | 6.14 |
> | LFQ-1k | 1024 | 100% | 0.82 | 0.77 | 0.86 | 4.35 |
> | LFQ-2k | 2048 | 100% | 0.84 | 0.79 | 0.88 | 3.62 |
> | LFQ-4k | 4096 | 100% | 0.86 | 0.82 | 0.91 | 3.31 |
> | LFQ-8k | 8192 | 99.50% | 0.92 | 0.86 | 0.93 | 2.58 |
> | LFQ-16k | 16384 | 98.60% | 0.92 | 0.87 | 0.94 | 2.32 |
>
>
> > `Q7:` The model does not learn the distribution of protein lengths. Have you tried to overcome this limitation?
>
> `A7`: Thanks for your question. In our paper, our primary purpose is to conduct fair comparisons with previous models under the similar settings to better assess the strengths and limitations of our models, hence follow MultiFlow in sampling within length intervals. Meanwhile, in many protein design applications users have prior knowledge of the target lengths or the ranges of lengths, or indeed require explicit length control and manipulation. As such, we may not need to directly learn the length distribution.
>
> DPLM-2 is capable of generating proteins from the empirical length distribution. Specifically, we sample 2048 sequences with length sampled from the length distribution of PDB + SwissProt. The table below demonstrates that DPLM-2 can generate highly plausible proteins, which is consistent with sampling with length intervals.
>
> |  | scTM | scRMSD | pLDDT |
> |---|---|---|---|
> | Original: [100, 200, ..., 500]  | 0.925 ± 0.085 | 3.899 ± 3.723 | 82.686 |
> | Training set (PDB+Swissprot) length dist. | 0.929 ± 0.086 | 3.967 ± 3.257 | 83.698 |
>
> > `Q8:` The ablation results presented in 4.1.3 and table 3 are controversial. Could you please clarify the procedure, how many samples was used for evaluation and so on?
>
> `A8`: Thanks for pointing this out. In ablation study, we evaluate the effects of sequence pre-training and data augmentation on unconditional protein generation. Specifically, We investigate the effect of sequence pre-training by randomly initializing DPLM-2 instead of using DPLM parameters, while for effect of predicted structures we leverage only PDB structures for training. We conduct experiments on a 150M parameter DPLM-2. For each DPLM-2 variant, we sample 100 examples for each length in 100, 200, 300, 400 and 500. We evaluate designability by scTM and diversity by the number of difference clusters for each length. The results of Table 3 in our paper demonstrate the effectiveness of sequence pre-training and data augmentation.

---

> ### Author Response · Authors · 2024-11-24
>
> > `Q9:` The experiment on DeepLoc is not described in sufficient detail. Why you chose to use only one tiny dataset? Could you provide experiments that show the described catastrophic forgetting issue, which is of high importance?
>
> `A9`: Thank you for your question. In our study, we selected the DeepLoc subcellular dataset because it highlights a pronounced performance gap between DPLM-2 and DPLM, which serves as the initialization model for DPLM-2. This dataset provides an ideal testbed for investigating the factors contributing to this performance drop.
>
> We hypothesize two potential causes for the observed degradation:
> 1. DPLM-2 needs to accommodate additional structural representations given the same model capacity (parameters), which could negatively impact the representation learning performance.
> 2. As continuous training on smaller magnitude of structure data, DPLM-2 may experience catastrophic forgetting of the representation power gained during DPLM's large-scale sequence pretraining.
>
> To explore (1), we eliminated pretraining factors by retraining both DPLM and DPLM-2 with random initialization on the SwissProt and PDB datasets for 100K training steps. Additionally, we evaluated performance across all three tasks (HumanPPI, MetalIonBinding & DeepLoc) where DPLM-2 underperformed compared to DPLM. As shown in the table below, when large-scale sequence pretraining is removed, DPLM-2 significantly outperforms DPLM (exp 2 vs exp 1). This indicates that incorporating structural information enhances performance rather than harming it, which rejects the hypothesis (1).
>
> However, when DPLM undergoes large-scale pretraining and DPLM-2 is subsequently trained from the pretrained DPLM, the performance of DPLM-2 on certain tasks diminishes (exp 4 vs exp 3). Given the relatively smaller structure data for DPLM-2 training, this suggests that catastrophic forgetting occurs during DPLM-2's multimodal training, reducing the advantages of large-scale pretraining. To verify and mitigate this, during the course of rebuttal of last week, we have curated additional 1.3M predicted structures from AFDB_rep [1], and trained DPLM-2 on this larger data. The experimental results show that the amount of structure data is indeed a key factor for better multimodal protein representations, leading to significantly improved performance over the original data (exp 5 vs exp 4). In particular, on HumanPPI, enlarging data from 200K to 1.5M helps DPLM-2 attain 2.3% improvement, and also outperforms SaProt, a strong multimodal PLM trained with 40M foldseek tokenized AFDB data.
>
> | exp id |  | HumanPPI (Acc%) | MetalIonBinding (Acc%) | DeepLoc (Acc%) |
> |---|---|---|---|---|
> | 0 | SaProt | 86.41 | 75.75 | 85.57 |
> | 1 | DPLM (PDB + swissprot only) | 73.33 | 62.25 | 63.49 |
> | 2 | DPLM-2 (PDB + swissprot only) | 77.22 | 69.47 | 66.77 |
> | 3 | DPLM w/ fully pretraining on UniRef50 | 86.41 | 75.15 | 84.56 |
> | 4 | DPLM-2 w/ seq-pretraining (finetuned from DPLM) data: swissprot + pdb (200K) | 84.44 | 74.28 | 82.98 |
> | 5 | DPLM-2 w/ seq-pretraining (finetuned from DPLM) data: AFDB_rep + Swissprot + pdb (1.3M + 200K) | 87.78 | - | 83.42 |
>
>
> [1] Clustering predicted structures at the scale of the known protein universe. Nature 2023

---

> > ### Author Response · Authors · 2024-11-24
> >
> > > `Q10:` One of the main claims of the paper states that the co-generation guarantees consistency between structure and sequence. This is a strong statement that requires strong evidence. However on line 223 the assumption of conditional independence is made. Can you provide a rigorous mathematical proof that garantees such consistensy?
> >
> > `A10`: Thanks for this valuable question. Conditional independence is not a special assumption made by DPLM-2, it is a fundamental assumption made by diffusion models in general and their multimodal extensions, derived from the nature of their forward and backward processes. Previous theoretical studies on diffusion models have shown the convergence between generated samples distribution and data distribution is guaranteed under such conditional independence. In this paper, we have empirical evidence showing the consistency/compatibility between co-generated structures and sequences (e.g., scTM for co-generation), and we believe a mathematical proof of this is beyond the scope of this paper and can refer to the established theoretical results on diffusion. Nevertheless, we do love to elaborate on our thoughts and understanding of this as follows.
> >
> > **Conditional independence in diffusion models in general.**
> >
> >
> > Conditional independence over the elements of high-dimensional data, i.e., $p\_\theta(\mathbf{x}\_{t-1} | \mathbf{x}\_t) = \textstyle\prod\_{i=1}^d p\_\theta(x_{t-1, [i]} | \mathbf{x}\_t)$, is a prevailing assumption in diffusion probablistic models, both continuous and discrete variants, thanks to their iterative nature of probabilistic modeling. For example, in continuous diffusion models for vision generation, the denoising networks learn to reconstruct a denoised image at each timestep $t-1$ by simultaneously and independently operating over all pixels conditioned on the previous noisier pixels of the image and the current timestep (or equivalently noise level) $t$. So as for discrete diffusion, where discrete diffusion for text or protein sequence treats tokens of a sequence of $\mathbf{x\_{t-1}}$ independently given $\mathbf{x\_t}$. Several recent works have established the theoretical foundations on the convergence analysis of both continuous diffusion [1] and discrete diffusion [2,3], showing that there are theoretical guarantees of the convergence of the generated sample distribution of the diffusion models and the data distribution, which means that a well-learned diffusion models can preserve the statistical structure of the data (in other words, the consistency between the elements $\mathbf{x} = \\{ x\_1, ..., x\_d \\}$ of the generated samples).

---

> ### Author Response · Authors · 2024-11-24
>
> **Conditional independence in multimodal diffusion models.**
>
> Multimodal diffusion models aim to accommodate two or more modalities using a unified models. In this case, conditional independence between modalities is generally made $p\_\theta(\mathbf{x}\_{t-1}, \mathbf{y}\_{t-1} | \mathbf{x}\_t, \mathbf{y}\_{t}) = \textstyle\prod\_i p\_\theta(x\_{t-1, [i]} | \mathbf{x}\_t, \mathbf{y}\_{t-1}) \textstyle\prod\_j p\_\theta(y\_{t-1, [j]} | \mathbf{x}\_t, \mathbf{y}\_{t})$. For instance, UniDiffuser [4] is a multimodal continuous diffusion model that handles text and image modalities independently at each timestep, conditioned on the predictions from the previous timestep. Multiflow [3], on the other hand, factorizes protein data into three modalities—translation, orientation, and amino acid type—assuming conditional independence. It establishes a multimodal diffusion/flow-based model by combining three types of stochastic processes over Euclidean, SO(3), and categorical spaces for these modalities. In DPLM-2, we adopt a unified discrete diffusion approach where structure tokens and amino acid tokens are treated as conditionally independent. While theoretical guarantees for the convergence of mixture diffusion processes are still under-explored, existing discrete diffusion theory [2,3] ensures that a well-trained DPLM-2 can converge to the tokenized structure-sequence data distribution, supporting consistency between structure and sequence tokens.
>
> Additionally, theoretical studies on non-autoregressive Transformers (NATs) for text generation, which are akin to masked discrete diffusion, indicate that the learning difficulty of such models can be evaluated through conditional total correlation, a dataset-dependent and model-free measure captures the discrepancy between a joint data distribution and a fully factorized distribution under conditional independence [6]. These studies suggest that simplifying the original complex target data (e.g., by using instance-level knowledge distillation from other models as in NATs for text generation or in MulitFlow for sequence generation, or by using tokenized structure instead of 3D coordinates as in DPLM-2/ESM3), reduces conditional total correlation, thereby enhancing both learning and generation quality.
>
> Given the consistency of structure tokens and amino acid can be ensured to learn in DPLM-2 by previous theoretical results [2,3,6], the overall structure and sequence consistency can be achieved with a decent structure tokenizer, such as the one proposed in this paper, which can map structure tokens to their atomic coordinates with a sufficient accuracy.
>
> ----
>
> [1] Sampling is as easy as learning the score: theory for diffusion models with minimal data assumptions. ICLR 2023
>
> [2] Convergence analysis of discrete diffusion model: Exact implementation through uniformization. Arxiv 2024
>
> [3] Convergence of Score-Based Discrete Diffusion Models: A Discrete-Time Analysis. Arixv 2024
>
> [4] UniDiffuser: One Transformer Fits All Distributions in Multi-Modal Diffusion at Scale. ICML 2023
>
> [5] Generative Flows on Discrete State-Spaces: Enabling Multimodal Flows with Applications to Protein Co-Design. ICML 2024.
>
> [6] On the Learning of Non-Autoregressive Transformers. ICML 2021
>
>
> `---end of Q10---`

---

> > ### Author Response · Authors · 2024-11-26
> >
> > Dear Reviewer f8By,
> >
> > Hi, thank you for taking the time to provide such thoughtful and valuable feedback on our manuscript. We deeply appreciate your insights, and we have tried our best to address your concerns in our response. Please kindly check it out.
> >
> > As the final deadline for manuscript revisions is approaching (Nov 26 AoE), we would be very grateful for any further feedback you might have. Please don’t hesitate to reach out if you have additional questions—we’d be happy to provide further clarifications!
> >
> > Looking forward to hearing from you, and many many thanks!
> >
> > Best,
> >
> > Authors

---

> ### Comment · Reviewer_f8By · 2024-11-30
> **Official Comment by Reviewer f8By**
>
> Dear Authors,
>
> Thank you for the detailed response to my initial review. Some of my concerns have been addressed. However, several important points require further attention:
>
> 1. I appreciate the ablations. It would be valuable for the community if the analysis of the results of the ablation study end up in the final version of the manuscript.
>
> 2. DPLM-2 is a finetuned protein language model. So its evaluation should contain a great deal of sequence analysis. The authors provide only pLDDT, which is primarily a structural quality measure, and MMseqs2 clustering. MMseqs2 clustering (and also structural clustering for that matter) should be performed at different thresholds to capture different aspects of the diversity. Adding perplexity as a measure of naturalness/quality and novelty through sequence identity to nearest neighbor in the dataset to the evaluation toolkit would make analysis more sound.
>
> 3. In the presented setting, sc-TM and sc-RMSD evaluate the degree of consistency between the predictions in two modalities, sequence and structure. Both "quality" and "designability" are misnomers.
>
> 4. I appreciate the results on the novelty evaluation using TM-score on the full training set, not only PDB, which is a subset thereof. The results show that the model generates more structures similar to those in the training data, however, in this case reporting mean and std is insufficient as it may mask issues with memorization. It is advised to plot the distribution.
>
> 5. Regarding argmax vs. stochastic sampling. This important topic requires a thorough discussion, since different strategies are used for different tasks. In DPLM, which DPLM-2 is based upon, Gumbel-Max trick was used (at least) in unconditional generation to alleviate mode collapse. I have not found any mentions of this in the manuscript. Here, argmax sampling is used for inverse folding, while stochastic sampling is used for unconditional generation and scaffolding. Why different tasks use different sampling approaches? What stochastic approach is used, please provide the details? What is the reasoning behind this and how is it supported experimentally?
>
> 6. Even with added baselines, important comparisons are missing. There is a lot of active research on inverse folding, so there is no lack of good baselines and good methodology ([1-4] to name a few). At the end of the day, it is not about beating SOTA result, but about providing objective comparison for the benefit of the protein design community.
>
> 7. Thank you for adding the codebook utilization metrics. However, they are purely quantitative and don't provide insight into the semantic meaning of tokens. Since the structure tokenization is an important part of the work, it would very much benefit from a deep analysis of the used tokens. Since the codebook sizes far exceed the number of secondary structure features, it would be great to map tokens to known structural motifs beyond secondary structure and provide visualizations like in ESM-3 paper.
>
> 8. Regarding the representation learning comparison, there is a huge body of work out there for comparison on these tasks. Why not to compare against strong baselines? Again, it is not about beating SOTA, but about correctly reflecting the state of affairs in the experiments. Also, there is a more recent and more strongly performing Gear-net version (https://arxiv.org/abs/2303.06275).
>
> 9. Regarding one of the main claims of the paper, you do not answer the question. The following is outlined in the main contributions: "DPLM-2 enables unconditional co-generation of designable and diverse proteins that **guarantees** consistency between structure and sequence". It seems like a gross overstatement without a formal rigorous proof. If there is no guarantees of consistency, then only empirical evidence should be declared.
>
> 10. Adding a discussion of the used self-mixup training strategy would strengthen the work. Also the experiment and data in table 8 are not discussed. If the absolute numbers of clusters are reported, the number of samples that underwent clustering should be stated in the caption.
>
> References:
>
> [1] https://arxiv.org/abs/2305.15151
>
> [2] https://arxiv.org/abs/2306.16819
>
> [3] https://arxiv.org/abs/2312.06297v2
>
> [4] https://arxiv.org/abs/2310.11802

---

> > ### Author Response · Authors · 2024-12-02
> >
> > Dear Reviewer f8By,
> >
> > Thank you so much for your detailed and constructive feedback! We truly value your input and are currently working hard on additional experiments and analyses to address each of your points. Please rest assured that we take your comments very seriously and will provide a thorough response soon. Thanks!
> >
> > Best regards,
> >
> > Authors

---

> > > ### Author Response · Authors · 2024-12-03
> > > **Further Responses**
> > >
> > > Dear Reviewer f8By,
> > >
> > > Thank you for your thoughtful and encouraging feedback, which we truly appreciate. Your detailed and insightful comments are invaluable to us, and we are deeply grateful for the time and effort you have invested in reviewing our work.
> > >
> > > We have addressed your concerns and questions point by point below and kindly invite you to review our responses. Since there is an additional day for authors to reply, we welcome any further feedback you may have and are happy to provide further clarifications if needed.
> > >
> > > Finally, we will carefully incorporate all results and discussions into the revised version of the manuscript to ensure it meets the highest standards. Thanks for helping us make our work a better one!

---

> > > > ### Author Response · Authors · 2024-12-03
> > > >
> > > > > `Q2:` DPLM-2 is a finetuned protein language model. So its evaluation should contain a great deal of sequence analysis. The authors provide only pLDDT, which is primarily a structural quality measure, and MMseqs2 clustering. MMseqs2 clustering (and also structural clustering for that matter) should be performed at different thresholds to capture different aspects of the diversity. Adding perplexity as a measure of naturalness/quality and novelty through sequence identity to nearest neighbor in the dataset to the evaluation toolkit would make analysis more sound.
> > > >
> > > > Thanks for your suggestions. We have conducted more comprehensive evaluations as you advised, including:
> > > > 1. sequence and structural diversity: We conduct MMseqs2 clustering and foldseek structural clustering at different thresholds. We calculate diversity for high-quality samples, following the practice of multiflow. For MMseqs2 clustering, we select samples with pLDDT > 70, while for foldseek clustering we choose samples with scTM > 0.5.
> > > > 2. sequence naturalness: We calculate perplexity as a measure of naturalness with ProGen2-large [1],
> > > > 3. sequence novelty: We calculate novelty through sequence identity to the nearest neighbor in the training set.
> > > > 4. we plan to also provide evaluation of conditional perplexity derived from  $p(\text{seq} | \text{struct})$ using invfold model like ESM-IF [2] in the future manuscript.
> > > >
> > > > All models generate 100 samples per length in the range of [100, 200, 300, 400, 500] for evaluation, with the results summarized in the table below.
> > > >
> > > > One particularly insightful observation is the distinct behavior of MultiFlow (w/ distillation) and DPLM-2 regarding structural diversity. Specifically, DPLM-2 exhibits greater diversity under strict TM-score thresholds (≤0.5), while MultiFlow achieves better diversity at higher TM-score thresholds (≥0.7). Combined with the average inner-TM scores (DPLM-2: 0.275, MultiFlow: 0.356) presented in Q1 of our initial response, this suggests that DPLM-2 excels at generating diverse structures in terms of global topologies but exhibits limited structural variation within each cluster. This finding highlights a key limitation of the current structural tokenization approach: the loss of fine-grained structural variations, emphasizing the need for future improvements in this area.
> > > >
> > > > Additionally, DPLM-2 achieves the lowest ProGen2 perplexity, while its sequence identity to training data (0.475) is higher than that of DPLM and MultiFlow. This indicates that the sequences generated by DPLM-2 align more closely with the natural distribution.
> > > >
> > > >
> > > > |  | MultiFlow (official w/ distillation) | MultiFlow (retrained on DPLM-2 data) | DPLM | DPLM2 |
> > > > |---|---|---|---|---|
> > > > | pLDDT | 79.4 | 62.6 | 84.0 | 83.7 |
> > > > | seq-diversity (↑) / mmseq cluster at seq-id=0.3 & plddt > 70 | 0.804 | 0.204 | 0.740 | 0.745 |
> > > > | seq-diversity (↑) / mmseq cluster at seq-id=0.5 & plddt > 70 | 0.860 | 0.294 | 0.745 | 0.755 |
> > > > | seq-diversity (↑) / mmseq cluster at seq-id=0.7 & plddt > 70 | 0.862 | 0.294 | 0.815 | 0.795 |
> > > > | seq-diversity (↑) / mmseq cluster at seq-id=0.9 & plddt > 70 | 0.862 | 0.294 | 0.885 | 0.895 |
> > > > | struct-diversity (↑) / foldseek at tmscore=0.3 & scTM > 0.5 | 0.030 | 0.080 | -- | 0.198 |
> > > > | struct-diversity (↑) / foldseek at tmscore=0.5 & scTM > 0.5 | 0.500 | 0.440 | -- | 0.545 |
> > > > | struct-diversity (↑) / foldseek at tmscore=0.7 & scTM > 0.5 | 0.962 | 0.830 | -- | 0.646 |
> > > > | struct-diversity (↑) / foldseek at tmscore=0.9 & scTM > 0.5 | 0.990 | 0.910 | -- | 0.746 |
> > > > | seq-naturalness / progen2 ppl | 8.11 ± 2.08 | 9.15 ± 2.77 | 4.33 ± 2.51 | 4.08 ± 2.00 |
> > > > | seq-novelty / mmseq search against PDB+swissprot | 0.306 | 0.312 | 0.304 | 0.475 |
> > > >
> > > > ----
> > > >
> > > > [1] Progen2: Exploring the Boundaries of Protein Language Models. Cell system 2022
> > > >
> > > > [2] Learning inverse folding from millions of predicted structures. ICML 2022

---

> > > > > ### Author Response · Authors · 2024-12-03
> > > > >
> > > > > > `Q3:` In the presented setting, sc-TM and sc-RMSD evaluate the degree of consistency between the predictions in two modalities, sequence and structure. Both "quality" and "designability" are misnomers.
> > > > >
> > > > > Thanks for your suggestions. Considering that scTM and scRMSD represent the consistency between structure and sequence, we will rename the evaluation metric to "structure-sequence compatibility/consistency". If you have a more appropriate way of expressing it, we would be glad to hear your suggestions!
> > > > >
> > > > > > `Q4:` I appreciate the results on the novelty evaluation using TM-score on the full training set, not only PDB, which is a subset thereof. The results show that the model generates more structures similar to those in the training data, however, in this case reporting mean and std is insufficient as it may mask issues with memorization. It is advised to plot the distribution.
> > > > >
> > > > > Thanks for your suggestions. To more comprehensively demonstrate the similarity between the generated proteins and training data, we have plotted the distribution as you suggested. Specifically, for each generated protein, we search the structure against the training set with foldseek and calculate the TMscore and sequence identity with the most similar protein in the training set.
> > > > >
> > > > > As seen in [this figure](https://anonymous.4open.science/r/supple_dplm2-2342/novelty_tm_id.pdf), a considerable number of points are distributed in the bottom right area, indicating a high TMscore but a relatively low sequence identity. This suggests that DPLM-2 is likely to generate proteins with structures similar to those in the training set but often with novel sequences.
> > > > >
> > > > >
> > > > > > `Q5:` Regarding argmax vs. stochastic sampling. This important topic requires a thorough discussion, since different strategies are used for different tasks. In DPLM, which DPLM-2 is based upon, Gumbel-Max trick was used (at least) in unconditional generation to alleviate mode collapse. I have not found any mentions of this in the manuscript. Here, argmax sampling is used for inverse folding, while stochastic sampling is used for unconditional generation and scaffolding. Why different tasks use different sampling approaches? What stochastic approach is used, please provide the details? What is the reasoning behind this and how is it supported experimentally?
> > > > >
> > > > >
> > > > > We apologize for the confusion and the lack of details about the sampling strategies in the manuscript. We would like to make further clarifications as follows. We will include these missing details in the next version. Thanks again for your suggestions.
> > > > >
> > > > > **Why different tasks use different sampling approaches?**
> > > > > In our original manuscript, we utilize argmax decoding for conditional generation tasks (e.g., folding and inverse folding) to maximize generation accuracy and ensure a fair comparison with DPLM. On the other hand, stochastic sampling was employed for unconditional generation or motif-scaffolding tasks to encourage generation diversity while maintaining good generation quality. In the original rebuttal Q4, we also demonstrate that stochastic sampling with temperature annealing can be used in the inverse folding task to sample more diverse sequences while ensuring structural plausibility.
> > > > >
> > > > > **What stochastic approach is used, please provide the details?**
> > > > > We utilize a temperature-based stochastic approach. With a normally fixed temperature $\tau$ (we used  $\tau=0.7$ in the original manuscript), the full process can be referred to DPLM paper Appendix A, Algorithm 1.
> > > > >
> > > > > Here we mainly focus on the temperature-annealed version used in our initial rebuttal for better sampling diversity. The details are shown below:
> > > > > 1. Determine the minimum temperature $\tau\_{\min}$, the maximum temperature $\tau\_{\max}$ and the total sampling steps $T$. Initialize the start timestep $t = 0$.
> > > > > 2. For each timestep $t$, calculate the temperature $\tau \leftarrow \tau\_{\min} + (1-\frac{t}{T})(\tau\_{\max} - \tau\_{\min})$.
> > > > > 3. DPLM-2 performs a sampling iteration based on the $\tau$. Increment the timestep: $t \leftarrow t + 1$.
> > > > > 4. Repeat (2) if $t \leq T$, otherwise proceed to (5).
> > > > > 5. End sampling.
> > > > >
> > > > > We will include a more formal presentation of the technical details of this temp-annealed sampling in next version of manuscript.
> > > > >
> > > > > **What is the reasoning behind this and how is it supported experimentally?**
> > > > > The gumbel-argmax trick used in the original DPLM is akin to sampling with fixed temperature 1.0 at every timestep, and we found the unconditional generation performance with gumbel-argmax trick is sub-optimal in terms of diversity.
> > > > > To this end, the temperature annealing sampling approach introduces more randomness during the initial stage of sampling by using a large temperature, and more fidelity during the final stage of sampling by using a small temperature. This method improves generation diversity while maintaining generation quality.

---

> > > > > > ### Author Response · Authors · 2024-12-03
> > > > > >
> > > > > > > `Q10:` Adding a discussion of the used self-mixup training strategy would strengthen the work. Also the experiment and data in table 8 are not discussed. If the absolute numbers of clusters are reported, the number of samples that underwent clustering should be stated in the caption.
> > > > > >
> > > > > > Thanks for your suggestions. We would like to make further clarification of the rationale behind the self-mixup training strategy and the details of it as follows.
> > > > > >
> > > > > > **Clarification on self-mixup**
> > > > > >
> > > > > > The exposure bias problem, which is described as the input mismatch between training and sampling, has already garnered attention in the research of continuous diffusion [1,2,3] and NLP [4,5]. We find that the discrete diffusion model also encounters this issue.  According to Eq.4 in the manuscript, the model is trained to be tasked with $p\_{\theta}(\mathbf{x}^{(0)}\_i|\mathbf{x}^{(t)})$, essentially doing masked-prediction. During training, the model makes prediction conditioned on the $\mathbf{x}^{(t)}$, which is a mixup of ground-truth tokens and mask tokens as noise:  ${\mathbf{x}}^{(t)} = \alpha\_t {\mathbf{x}^{(0)}} + (1-\alpha\_t)\mathbf{q}\_{\text{noise}}$; However, during inference, the model predicts $p\_{\theta}(\mathbf{x}^{(0)}\_i|\hat{\mathbf{x}}^{(t)})$ conditioned on previously generated sample $\hat{\mathbf{x}}^{(t)}$, which is a mixup of model prediction and masks, essentially requiring denoising and masked-prediction. The difference between $\mathbf{x}^{(t)}$ and $\hat{\mathbf{x}}^{(t)}$ causes a discrepancy between $p\_{\theta}(\mathbf{x}^{(0)}\_i|\mathbf{x}^{(t)})$ and $p\_{\theta}(\mathbf{x}^{(0)}\_i|\hat{\mathbf{x}}^{(t)})$, potentially leading to error accumulation since the model trend to be over-confident on its predictions (as in training the model is always exposed to ground-truth, hence the name exposure bias), and negatively impacting the generation performance.
> > > > > >
> > > > > > To mitigate this, we propose to bridge this gap by training model to make predictions conditioned on its own predicted results:
> > > > > >
> > > > > > 1. Predict $\hat{\mathbf{x}}^{(0)}$ conditioned on the ground truth training sample $\mathbf{x}^{(t)}$
> > > > > > 2. Construct the generated sample: $\hat{\mathbf{x}}^{(t)} \leftarrow \hat{\mathbf{x}}^{(0)} + (1-\alpha\_t)\mathbf{q}\_{\text{noise}}$
> > > > > > 3. Compute self-mixup loss according to Eq4:
> > > > > > $\hat{\mathcal{J}}\_t = \mathbf{E}\_{q(\mathbf{x}^{(0)})} \left[\lambda^{(t)}  \sum\_{1 \leq i \leq L} b_i(t) \cdot \log p\_{\theta}(\mathbf{x}^{(0)}\_i|
> > > > > > \hat{\mathbf{x}}^{(t)})\right]$
> > > > > >
> > > > > > We can illustrate this more clearly with a break-down example:
> > > > > > Let the ground truth $\mathbf{x}^{(0)}$ be `A B C D E` and the $\mathbf{x}^{(t)}$ be `[m] [m] [m] D E` as in masked discrete diffusion, where `[m]` represents the mask token.
> > > > > > 1. call a model forward to obtain model prediction $\hat{\mathbf{x}}^{(0)}$, which is `a b c D E` (with the ground truth token `D E` preserved for masked positions), where `a b c` represent model prediction by argmax.
> > > > > > 2. construct self-mixup $\hat{\mathbf{x}}^{(t)}$. In our experiments, we always replace the ground truth token in $\mathbf{x}^{(t)}$ (`D E` in this case) with the mask token. Therefore $\hat{\mathbf{x}}^{(t)}$ becomes `a b c [m] [m]`,
> > > > > > 3. compute self-mixup loss. Now we can calculate cross entropy between $\hat{\mathbf{x}}^{(t)}$ (`a b c [m] [m]`) and  $\mathbf{x}^{(0)}$ (`A B C D E`) at all positions. More specifically, this can be seen as mask positions are applied masked language modeling loss while non-masked positions are applied denoising autoencoder loss. Moreover, this also improves sample-efficiency compared to typical masked discrete diffusion where training loss is applied to mask positions.
> > > > > >
> > > > > > In our experiments, we first train DPLM-2 with the original loss $\mathcal{J}\_t$ in Eq.4 for 50K steps to ensure the prediction quality. This step is crucial; otherwise, the model's predictions might be poor, leading to an excessively large self-mixup loss and causing training instability. After this initial phase, we continue training with self-mixup loss $\hat{\mathcal{J}}\_t$ to mitigate the exposure bias issue.
> > > > > >
> > > > > > **Regarding the experimental details of Table 8**
> > > > > >
> > > > > > For results presented in Table 8, we conduct experiments with the DPLM-2 650M model on the unconditional generation task. We sample 100 proteins within each length interval and calculate scTM for structure-sequence compatibility and the number of clusters for diversity. We will supplement the necessary experimental details in the captions of all diagrams, tables, and figures in our paper in the next version, as you suggested.
> > > > > >
> > > > > > -----
> > > > > > [1] Elucidating the Exposure Bias in Diffusion Models. ICLR 2024
> > > > > >
> > > > > > [2] Input Perturbation Reduces Exposure Bias in Diffusion Models. ICML 2023
> > > > > >
> > > > > > [3] Alleviating Exposure Bias in Diffusion Models through Sampling with Shifted Time Steps. ICLR 2024
> > > > > >
> > > > > > [4] Sequence level training with recurrent neural networks. ICLR 2016
> > > > > >
> > > > > > [5] Scheduled sampling for sequence prediction with recurrent neural networks. NIPS 2015

---

> > > > > > > ### Author Response · Authors · 2024-12-03
> > > > > > >
> > > > > > > > `Q7:` Thank you for adding the codebook utilization metrics. However, they are purely quantitative and don't provide insight into the semantic meaning of tokens. Since the structure tokenization is an important part of the work, it would very much benefit from a deep analysis of the used tokens. Since the codebook sizes far exceed the number of secondary structure features, it would be great to map tokens to known structural motifs beyond secondary structure and provide visualizations like in ESM-3 paper.
> > > > > > >
> > > > > > > Thank you for the valuable suggestions. We now fully understand your point and completely agree that mapping structure tokens to structural motifs can provide more fine-grained insights into what structure tokens learn.
> > > > > > >
> > > > > > > Since mapping each structure token to a dataset of "known structural motifs" is quite challenging for us within this short period, we propose an alternative approach. Specifically, as structure tokens are residue-wise representations, we aim to map each structure token to structural motifs defined as the nearest-neighbor local structural environment of a residue in the training dataset (for efficiency, we used only the PDB dataset). The process is as follows:
> > > > > > >
> > > > > > > 1. For each structure in the PDB dataset (~20K in total), we first tokenize the structure into structure tokens and save the (structure token, 30-nearest-neighbors structural motif) pair for each residue. We use 30 nearest neighbors because the pre-trained GVPTransformerEncoder, which we used as the structure encoder, employs 30 nearest neighbors as the hyperparameter for geometric features.
> > > > > > > 2. After processing all structures, we obtain a table where each row corresponds to a structure token and its associated structural motifs (i.e., num_structural_motifs).
> > > > > > > 3. To analyze whether a structure token tends to occur in a similar local structural environment, we use Foldseek (TM-threshold = 0.5) to cluster the structural motifs for each structure token (i.e., motif_clusters). Although Foldseek may not be entirely accurate in clustering such short and discontinuous structural regions, it provides a reasonable comparative sense of the similarity/difference among all structural motifs associated with each structure token.
> > > > > > >
> > > > > > > In [this figure](https://anonymous.4open.science/r/supple_dplm2-2342/struct_token_hist.pdf), we plot the histogram of num_structural_motifs vs. motif_clusters for each structure token (randomly sampling 500 out of 8,192 structure tokens to ensure readability). From the visualization, we observe that many structure tokens correspond to highly similar structural motifs (evidenced by a small ratio of motif_clusters to num_structural_motifs), while others exhibit a high degree of ambiguity.
> > > > > > >
> > > > > > > Additionally, leveraging this mapping between structure tokens and structural motifs, we can create visualizations similar to those in ESM-3, as you suggested. In [this figure](https://anonymous.4open.science/r/supple_dplm2-2342/struct_tokens_vis.pdf), we showcase two structure tokens and their corresponding similar structural motifs across four different PDB structures, illustrating the diversity or consistency in the mapped local structural environments.

---

> ### Author Response · Authors · 2024-12-03
>
> > `Q8:` Regarding the representation learning comparison, there is a huge body of work out there for comparison on these tasks. Why not to compare against strong baselines? Again, it is not about beating SOTA, but about correctly reflecting the state of affairs in the experiments. Also, there is a more recent and more strongly performing Gear-net version (https://arxiv.org/abs/2303.06275).
>
>
> Thanks for your valuable suggestion. We have accordingly added more recent strong baselines and updated the stronger GearNet version as you suggested. The results of newly supplemented baseline models are mainly focused on the EC and GO downstream tasks. Meanwhile, we discovered discrepancies between the new results and previous results in the EC and GO tasks. We utilized old version of SaProt codebase as the evaluation pipeline. After consulting with the authors of SaProt, it is because there was an issue in the early SaProt codebase in calculating the metrics for these tasks, which had been fixed later. As such, we have updated the EC and GO results for both SaProt and DPLM-2.
>
> | Models                 | Thermostability | HumanPPI | Metal Ion Binding | EC    | GO    |       |       | DeepLoc     |        |
> |------------------------|-----------------|----------|-------------------|-------|-------|-------|-------|-------------|--------|
> |                        |                 |          |                   |       | MF    | BP    | CC    | Subcellular | Binary |
> | SaProt                 | 0.724           | 86.41    | 75.75             | 0.882 | 0.682 | 0.486 | 0.479 | 85.57       | 93.55  |
> | SaProt-GearNet         | 0.660           | 85.80    | 74.44             | 0.889 | 0.678 | 0.522 | 0.508 | 84.16       | 93.63  |
> | MIF-ST                 | 0.694           | 75.54    | 75.08             | 0.807 | 0.633 | 0.375 | 0.322 | 78.96       | 91.76  |
> | GearNet                | 0.571           | 73.86    | 71.26             | 0.874 | 0.644 | 0.481 | 0.476 | 69.45       | 89.18  |
> | GearNet updated        | --              | --       | --                | 0.890 | 0.681 | 0.488 | 0.464 | --          | --     |
> | CoupleNet [1]          | --              | --       | --                | 0.866 | 0.669 | 0.467 | 0.494 | --          | --     |
> | CDConv [2]             | --              | --       | --                | 0.820 | 0.654 | 0.453 | 0.479 | --          | --     |
> | ESM2-650M-S [3]        | 0.668           | --       | --                | 0.823 | 0.649 | 0.463 | 0.519 | --          | --     |
> | VABS-NET [4]           | --              | --       | --                | 0.900 | 0.695 | 0.531 | 0.579 | --          | --     |
> | ESM-GearNet-INR-MC [5] | --              | --       | --                | 0.896 | 0.683 | 0.518 | 0.504 | --          | --     |
> | ESM2-650M              | 0.691           | 84.78    | 71.88             | 0.868 | 0.670 | 0.473 | 0.470 | 83.68       | 92.28  |
> | DPLM                   | 0.695           | 86.41    | 75.15             | 0.875 | 0.680 | 0.357 | 0.409 | 84.56       | 93.09  |
> | DPLM-2                 | 0.714           | 84.44    | 74.28             | 0.881 | 0.682 | 0.493 | 0.481 | 82.98       | 93.64  |
>
> ----
>
>
> [1] Learning Complete Protein Representation by Dynamically Coupling of Sequence and Structure. NIPS 2024
>
> [2] Continuous-Discrete Convolution for Geometry-Sequence Modeling in Proteins. ICLR 2023
>
> [3] Structure-informed Protein Language Model. Arxiv 2024
>
> [4] Pre-Training Protein Bi-level Representation Through Span Mask Strategy On 3D Protein Chains. ICML 2024
>
> [5] Pre-training Sequence, Structure, and Surface Features for Comprehensive Protein Representation Learning. ICLR 2024
>
>
>
>
> > `Q9:` Regarding one of the main claims of the paper, you do not answer the question. The following is outlined in the main contributions: "DPLM-2 enables unconditional co-generation of designable and diverse proteins that guarantees consistency between structure and sequence". It seems like a gross overstatement without a formal rigorous proof. If there is no guarantees of consistency, then only empirical evidence should be declared.
>
> We apologize for the confusion due to our inaccurate use of "guarantees". We never aimed to over-claim that the structure-sequence consistency is mathematically and rigorously guaranteed. Instead, we mainly claim our empirical findings. To make the statements more precise, we will rephrase our wording to "DPLM-2 enables unconditional protein co-generation of both structure and sequence, which demonstrates good structure-sequence consistency."
>
> Besides, your initial question on this greatly encouraged us to delve deep into this hence we provided a discussion with literature review in our initial response. We are also truly grateful for your inspirations!

---

> ### Author Response · Authors · 2024-12-03
>
> > `Q6:` Even with added baselines, important comparisons are missing. There is a lot of active research on inverse folding, so there is no lack of good baselines and good methodology ([1-4] to name a few). At the end of the day, it is not about beating SOTA result, but about providing objective comparison for the benefit of the protein design community.
>
> Thanks for your valuable suggestion. We have added more good inverse folding baselines as you suggested to provide objective comparison for the benefit of the protein design community.
> We conduct experiments on the CATH 4.2 testset, and * means results are quoted from their respective paper. We will include these results in the next version of our manuscript.
> | Model | AAR | scTM |
> |---|---|---|
> | Knowledge-Design* [1] | 60.77 | -- |
> | GraDe-IF* [2] | 52.21 | -- |
> | MMDesign* [3] | 54.88 | -- |
> | VFN-IFE* [4] | 62.67 | -- |
> | PiFold* [5] | 51.66 | -- |
> | Bridge-IF* [6] | 58.59 | -- |
> | ProteinMPNN | 45.96 | 0.87 |
> | LM-Design | 54.41 | 0.88 |
> | DPLM-2  Argmax decoding | 42.7 | 0.84 |
> | DPLM-2  temp-annealed sampling | 36.3 | 0.84 |
>
> ----
> [1] Knowledge-Design: Pushing the Limit of Protein Design via Knowledge Refinement. ICLR 2024
>
> [2] Graph Denoising Diffusion for Inverse Protein Folding. NIPS 2023
>
> [3] Progressive Multi-Modality Learning for Inverse Protein Folding. ICME 2024
>
> [4] De novo protein design using geometric vector field networks. ICLR 2024
>
> [5] PiFold: Toward effective and efficient protein inverse folding. ICLR 2023
>
> [6] Bridge-IF: Learning Inverse Protein Folding with Markov Bridges. NIPS 2024

---

> > ### Author Response · Authors · 2024-12-03
> >
> > This wraps up our response to your follow-up questions from `Q2` to `Q10` (not in a sequential order though)! We hope we have adequately addressed your questions and concerns. Thank you once again for your invaluable feedback, which has significantly helped improve the quality of our work!
> >
> > Cheers!
> >
> > Authors

---

### Author Response · Authors · 2024-11-24
**General Response & Summary of Rebuttal**

Dear Reviewers, ACs and SACs,

We want to express sincere appreciation for all reviewers' efforts in reviewing and providing valuable suggestions! We have tried our best to address reviewers' concerns respectively, mainly including:

- Added comprehensive ablation study of training strategies and analysis of structure tokens, as suggested by Reviewer f8BY.
- Added recognized baselines and explored temperature-annealed sampling strategy for  improved sequence novelty in inverse folding as suggested, by Reviewer f8BY and Reviewer FqCq.
- Added representation learning experiments for investigating the factors contributing to the performance gap between DPLM-2 and DPLM, as suggested by Reviewer f8BY and Reviewer FqCq.
- Added discussion about theoretical convergence guarantees on the consistency between structure and sequence under the conditional independence assumption in diffusion model, as suggested by Reviewer f8BY.
- Added essential technical details and improved clarity in our paper as suggested by Reviewer 4A34.
- Added comprehensive analysis of scTM, scRMSD and secondary structure for unconditional protein generation, as suggested by Reviewer 4A34 and Reviewer FqCq.
- Added discussion on comparison of structure tokenization method between our method and Gao et al., 2024, as suggested by Reviewer 4A34.
- Added exploration on training on larger-scale structure dataset, as suggested by Reviewer FqCq.
- Added discussion on information that structure tokens learn and how to address the potential issue of lossy compression and the absence of fine-grained structural variation introduced by discrete structure tokens, as suggested by Reviewer FqCq.
- Added discussion on the repetition issues in DPLM and DPLM-2 as suggested by FqCq.
- Also fixed some abnormal evaluation results in unconditional protein generation caused by incorrect implementation or sampling configurations.

We again thank everyone's time and effort in discussing, providing valuable and inspiring feedback, and helping us improve our manuscript. Moreover, we have accordingly revised the paper to best include most of the insightful suggestions and comments from the reviewers. We do sincerely appreciate you!

We are very happy to address any further feedback during the discussion phase!

Many thanks, and cheers!

Authors

---

### Meta-Review · Area_Chair_FtBn · 2024-12-19

**Metareview:**

This paper proposes a joint generative model for protein structure (represented by discrete tokens) and sequence. The paper details a comprehensive study. The authors and two of three reviewers engaged in a thorough discussion. Two referees are strongly supporting acceptance and the non-engaged referee the opposite. Based upon the discussion and browsing through the paper, acceptance is recommended.

**Additional Comments On Reviewer Discussion:**

None.

---

### Decision · Program_Chairs · 2025-01-22

Accept (Poster)